# Liberating confinement from Lagrangians:
# 1-form symmetries and lines in 4D $\mathcal{N} = 1$ from 6D $\mathcal{N} = (2,0)$

**Lakshya Bhardwaj, Max Hübner and Sakura Schäfer-Nameki**

Mathematical Institute, University of Oxford,
Andrew-Wiles Building, Woodstock Road, Oxford, OX2 6GG, UK

## Abstract

We study confinement in 4d $\mathcal{N} = 1$ theories obtained by deforming 4d $\mathcal{N} = 2$ theories of Class S. We argue that confinement in a vacuum of the $\mathcal{N} = 1$ theory is encoded in the 1-cycles of the associated $\mathcal{N} = 1$ curve. This curve is the spectral cover associated to a generalized Hitchin system describing the profiles of two Higgs fields over the Riemann surface upon which the 6d $(2,0)$ theory is compactified. Using our method, we reproduce the expected properties of confinement in various classic examples, such as 4d $\mathcal{N} = 1$ pure Super-Yang-Mills theory and the Cachazo-Seiberg-Witten setup. More generally, this work can be viewed as providing tools for probing confinement in non-Lagrangian $\mathcal{N} = 1$ theories, which we illustrate by constructing an infinite class of non-Lagrangian $\mathcal{N} = 1$ theories that contain confining vacua. The simplest model in this class is an $\mathcal{N} = 1$ deformation of the $\mathcal{N} = 2$ theory obtained by gauging $SU(3)^3$ flavor symmetry of the $E_6$ Minahan-Nemeschansky theory.



# 1 Introduction

This paper is devoted to the study of confinement in a large class of 4d $\mathcal{N}=1$ theories. We begin by reviewing the definition of confinement that we use in this paper. A vacuum $r$ of a quantum field theory (QFT) $\mathfrak{T}$ is called confining if the vacuum expectation value (vev) of some genuine[1] line operator in $\mathfrak{T}$ exhibits area law in $r$. This is correlated with the existence of confining strings in the spectrum which can end on such line operators and are responsible for giving rise to the linear potential that gives rise to the area law. A classic example is provided by 4d $\mathcal{N}=1$ pure Super-Yang-Mills (SYM) theory with gauge group $SU(n)$. This theory has $n$ vacua and in each vacuum, the Wilson line operator in the fundamental representation of $SU(n)$ exhibits area law. Thus, each vacuum is confining.

Confinement can be characterized in terms of the 1-form symmetry group $\mathcal{O}$ of $\mathfrak{T}$ [1], which captures equivalence classes of line operators with two line operators $L_1$ and $L_2$ considered

---

[1]A non-genuine line operator is one which lives at the boundaries or corners of higher-dimensional extended operators. A genuine line operator exists independently of any higher-dimensional extended operator.

to be in the same class if there exists a local operator living at the junction of $L_1$ and $L_2$. For the theories that are studied in this paper, these equivalence classes form an abelian group $\Lambda$ under OPE and characterize different charges under the 1-form symmetry group $\mathcal{O} = \widehat{\Lambda}$, which is the Pontryagin dual of $\Lambda$. If a line operator $L_1$ shows area or perimeter law, then another line operator $L_2$ in the same equivalence class shows the same law. Thus confinement can be characterized by dividing $\Lambda$ into two subsets: those showing area law and those showing perimeter law. Furthermore, the classes exhibiting perimeter law form a subgroup $\Lambda_r$ of $\Lambda$ which depends on the vacuum $r$ under consideration [2].

Consider a line operator $L$ that exhibits perimeter law in vacuum $r$. Then, any element of the 1-form symmetry group $\mathcal{O}$ under which $L$ is non-trivially charged is spontaneously broken in the vacuum $r$, because we can set the vev of $L$ to a non-zero constant by introducing a counter-term along the location of $L$, which cancels the perimeter dependence [1,3]. Thus, the 1-form symmetry group $\mathcal{O}_r$ preserved in vacuum $r$ can be written as[2]

$$\mathcal{O}_r = \widehat{\left(\frac{\Lambda}{\Lambda_r}\right)} \subseteq \widehat{\Lambda} = \mathcal{O}. \tag{1}$$

In other words, the data of the preserved 1-form symmetry group $\mathcal{O}_r$ is equivalent to the data of the set $\Lambda - \Lambda_r$ of line operators that exhibit area law. The confining strings are charged under $\mathcal{O}_r$ and their charges take values in its Pontryagin dual $\Lambda/\Lambda_r$.

The goal of this paper is to study confinement in $\mathcal{N} = 1$ deformations of 4d $\mathcal{N} = 2$ Class S theories [4], i.e. those 4d $\mathcal{N} = 2$ theories that can be obtained via compactification of the 6d $(2,0)$ theories on Riemann surfaces with a partial topological twist. We only consider those Class S theories that can be obtained by compactifying 6d $\mathcal{N} = (2,0)$ theory of $A_{n-1}$ type on a Riemann surface with untwisted punctures and no closed twist lines [5]. It should be noted that, though in practice we will largely consider such Class S constructions with irregular punctures, our considerations apply to general setups involving both regular and irregular punctures.

Much like the 4d $\mathcal{N} = 2$ Class S theories have a description in terms of Higgs bundles that are solutions to a Hitchin system [6], their $\mathcal{N} = 1$ deformations are similarly related to a set of BPS equations, which form a generalized Hitchin-like system, involving two Higgs fields. Akin to the spectral curve (or the Seiberg-Witten curve) in the $\mathcal{N} = 2$ case, one can associate a spectral curve, known as the $\mathcal{N} = 1$ curve, to the generalized Hitchin system [7,8]. The $\mathcal{N} = 1$ curve has appeared in various forms in the literature [7–28]. The $\mathcal{N} = 1$ deformation is realized by turning on singular behaviors of the second Higgs field at the locations of the punctures on the Riemann surface, which we dub as a "rotation" of the involved punctures. The profile of the second Higgs field is solved in terms of its asymptotic behavior by the generalized Hitchin system, which also constrains the profile of the $\mathcal{N} = 2$ Higgs field. Ultimately, the different solutions for the two Higgs field capture $\mathcal{N} = 1$ vacua of the deformed theory.

One can also study a generalization of the above setup, where one starts with a topological twist that only preserves 4d $\mathcal{N} = 1$ supersymmetry. The BPS equations are a generalized Hitchin system, where the two Higgs fields are now on an equal footing and can have singularities at mutually distinct locations on the Riemann surface. One can again associate an $\mathcal{N} = 1$ curve to a vacuum of the resulting 4d $\mathcal{N} = 1$ theory, which now does not have an interpretation as a deformation of a 4d $\mathcal{N} = 2$ Class S theory. In M-theory, the two twists are distinguished as follows: the Class S construction is obtained by wrapping M5-branes on the Gaiotto curve embedded in a local K3-surface. The $\mathcal{N} = 1$ twists arise by instead embedding the curve into a local Calabi-Yau threefold. These setups are discussed in sections 3.3 and 3.4.

The 1-form symmetry group $\mathcal{O}$ of a Class S theory is encoded in the 1-cycles of the punctured Riemann surface as discussed in detail in the recent work [29] (which is based on [30],

---

[2]Hats denote Pontryagin duals, i.e. $\widehat{\Lambda} := \mathrm{Hom}(\Lambda, U(1))$.

also see [31, 32] and the study of line operators in [33]), which we review in our context in section 3.2. In a similar spirit, we argue in section 3.5 that the preserved 1-form symmetry group $\mathcal{O}_r$ in a vacuum $r$ is encoded in the 1-cycles of the $\mathcal{N} = 1$ curve $\Sigma_r$ associated to the vacuum $r$. Our work can thus be viewed as a part of the recent surge of activity in the study of generalized symmetries of QFTs via compactifications of string theory and higher-dimensional QFTs [29, 31, 32, 34–43].

To explain and test our framework we first consider pure $\mathcal{N} = 1$ SYM as well as an extension to the setup studied in Cachazo-Seiberg-Witten (CSW) [20–22, 44], which corresponds to turning on a superpotential for the adjoint chiral superfield that lives in the $\mathcal{N} = 2$ vector multiplet. Both instances have well-documented confining vacua and we use them to test our general framework and to showcase how to go from the $\mathcal{N} = 1$ curve to the area/perimeter law of line operators.

The most exciting application of this work is to the realm of non-Lagrangian theories – which are ubiquitous in Class S constructions. We identify, in section 6, a family of $\mathcal{N} = 1$ theories, and show that this class of theories exhibits confinement! We argue – based on the curve and associated line operators that these theories have confining vacua. Clearly numerous generalizations of this can be considered, opening up a vast arena for studying confinement in theories with no apparent Lagrangian.

The plan of this paper is as follows:

In section 2 we will whet the appetite of the reader by discussing in detail the $\mathfrak{su}(2)$ SYM theory and its confining vacua using the Class S and $\mathcal{N} = 1$ perspective.

The main conceptual background of the paper will be explained in section 3, which includes the $\mathcal{N} = 1$ curve and associated Hitchin system. We then apply this general approach to two well-known instances of theories with confining vacua: in section 4 we study the $\mathcal{N} = 1$ curves for 4d $\mathcal{N} = 1$ $\mathfrak{su}(n)$ SYM , and use it to recover the well-known properties of confinement in this model. In section 5, we discuss the CSW model, whose confinement properties are also well-known in the literature. These two models provide extensive consistency checks of our proposed method of computing confinement, and also for testing our methodology.

In section 6, it comes time to reap the rewards as we use our method to find an infinite class of non-Lagrangian 4d $\mathcal{N} = 1$ theories that contain confining vacua. The simplest theory in this class can be described as an $\mathcal{N} = 1$ deformation of the $\mathcal{N} = 2$ asymptotically conformal theory obtained by gauging $\mathfrak{su}(3)^3$ flavor symmetry subgroup of the famous $E_6$ Minahan-Nemeschansky theory (or the $T_3$ trinion theory) [45]. Other theories in this class can be described as $\mathcal{N} = 1$ deformations of $\mathcal{N} = 2$ asymptotically conformal theories obtained by gauging $\mathfrak{su}(n)^n$ flavor symmetry group of the 4d $\mathcal{N} = 2$ SCFT obtained by compactifying $A_{n-1}$ $\mathcal{N} = (2, 0)$ theory on a sphere with $n$ maximal regular untwisted punctures.

The appendices contain a summary of notation and nomenclature in appendix A, details on a rotation, involving a non-generic superpotential, that is only possible at Argyres-Douglas points (appendix B), and a comprehensive discussion of the $\mathfrak{su}(3)$ and $\mathfrak{su}(4)$ CSW setups (appendix C and D, which contains a `Mathematica` code for computing the monodromies explicitly). In appendix E we discuss the relation to the Dijkgraaf-Vafa curve.

## 2  Appetizer: Confinement in $\mathfrak{su}(2)$ $\mathcal{N} = 1$ SYM Theory

In this section, we discuss confinement in $\mathcal{N} = 1$ pure super-Yang-Mills (SYM) theories with gauge algebra $\mathfrak{su}(2)$. As we review in subsection 2.1, there are three such theories: one with gauge group $SU(2)$ and two with gauge group $SO(3)$. The two theories with gauge group $SO(3)$ are distinguished by a discrete theta parameter, and correspondingly are referred to as $SO(3)_+$ and $SO(3)_-$ theories. All these theories have two massive vacua. Both of these vacua

are confining for the $SU(2)$ theory, while only one of them is confining for the $SO(3)_{\pm}$ theories.

In subsection 2.2, we discuss a construction of these 4d $\mathcal{N} = 1$ theories involving compactification of the 6d $A_1$ $\mathcal{N} = (2,0)$ theory. The construction involves transitioning through 4d $\mathcal{N} = 2$ pure SYM theories with gauge algebra $\mathfrak{su}(2)$. Compactifying the 6d theory on a sphere with two irregular punctures provides a Class S construction for these 4d $\mathcal{N} = 2$ theories. One can then further "rotate" one of the punctures to softly break $\mathcal{N} = 2$ supersymmetry to $\mathcal{N} = 1$. Field theoretically, this corresponds to adding a small superpotential proportional to the Coulomb branch (CB) parameter $u$ to the 4d $\mathcal{N} = 2$ theories (viewed as $\mathcal{N} = 1$ theories). As is well-known, all the CB vacua are lifted under this deformation, except the monopole and dyon points, giving rise to two massive vacua. As the rotation parameter is taken to infinity, these theories reduce to the pure $\mathfrak{su}(2)$ $\mathcal{N} = 1$ SYM theories, with the above two vacua being identified as the vacua of the pure $\mathcal{N} = 1$ theories.

We then proceed in subsection 2.3 to explain the above field theory results about confinement from the point of view of this compactification and properties of the 6d $\mathcal{N} = (2,0)$ theory.

## 2.1 Result from Field Theory

In this subsection, we review the discussion in [2, 46] about confinement in pure $\mathfrak{su}(2)$ $\mathcal{N} = 1$ SYM theories. A massive vacuum is called confining if a non-trivial subgroup of the 1-form symmetry group of the theory is left *unbroken* in that vacuum [1]. Thus we need to study the 1-form symmetry group in the $SU(2), SO(3)_{\pm}$ versions of the theory, and its spontaneous breaking in each of the two vacua.

The 1-form symmetry group acts on the line operators in the theory. For pure $\mathfrak{su}(2)$ gauge theories, the set $\mathcal{L}$ of line operators modulo screenings forms a group

$$\mathcal{L} \simeq \mathbb{Z}_2 \times \mathbb{Z}_2 \,, \tag{2}$$

under fusion. The two $\mathbb{Z}_2$ factors are generated by a fundamental Wilson line $W$ and a 't Hooft line $H$, with their sum $W + H$ (the sum operation represents fusion) being a dyonic Wilson-'t Hooft line operator.

These line operators are not all mutually local with respect to the Dirac pairing. For example, if $W$ is taken around $H$, or if $H$ is taken around $W$, then the correlation function changes sign. This non-locality is captured in a $\mathbb{Z}_2$ valued pairing defined on $\mathcal{L}$ as follows

$$
\begin{aligned}
\langle W, W \rangle &= 0 \,, \\
\langle H, H \rangle &= 0 \,, \\
\langle W, H \rangle &= \frac{1}{2} \,.
\end{aligned}
\tag{3}
$$

The change in phase of a correlation function as an element $\alpha \in \mathcal{L}$ is taken around $\beta \in \mathcal{L}$ is then

$$\exp\left(2\pi i \langle \alpha, \beta \rangle \right). \tag{4}$$

This means that not all the line operators in $\mathcal{L}$ are genuine line operators. If $\alpha, \beta \in \mathcal{L}$ are such that $\langle \alpha, \beta \rangle \neq 0$, then either $\alpha$ or $\beta$ lives at the boundary of a topological surface operator which acts on the other line operator, and this action is responsible for producing the phase (4). Thus specifying a theory $\mathfrak{T}$ (also called as an "absolute" theory) specifies a maximal subgroup $\Lambda$ of mutually commuting line operators in $\mathcal{L}$ which are genuine in $\mathfrak{T}$. The line operators in $\mathcal{L} - \Lambda$ are non-genuine line operators of $\mathfrak{T}$.

Consequently, a theory can only contain one out of $W$, $H$ and $W + H$ in its spectrum of genuine line operators. Choosing $W$ to lie in the spectrum gives rise to a pure 4d gauge theory

with gauge group $SU(2)$. On the other hand, choosing $H$ or $W + H$ give rise to pure 4d gauge theories with gauge group $SO(3)$. The two $SO(3)$ theories are differentiated by a discrete theta parameter[3]. The theory containing $H$ is called the $SO(3)_+$ theory and the theory containing $W + H$ is called the $SO(3)_-$ theory.

The 1-form symmetry in each of these three theories is $\mathbb{Z}_2$. The charged operator is the non-trivial element of $\mathcal{L}$ lying in the spectrum of genuine line operators of the theory. Notice, for future purposes, that the above analysis regarding $\mathcal{L}$, different global forms of the gauge group, discrete theta parameters and 1-form symmetry is independent of the amount of supersymmetry. In particular, it applies equally well to pure 4d $\mathfrak{su}(2)$ SYM theories with $\mathcal{N} = 1$ and $\mathcal{N} = 2$ supersymmetry.

To probe confinement in 4d $\mathfrak{su}(2)$ pure $\mathcal{N} = 1$ SYM theories, we realize them as deformations of 4d $\mathfrak{su}(2)$ pure $\mathcal{N} = 2$ SYM theories. Let us deform the $\mathcal{N} = 2$ theory by a superpotential $\mu \text{Tr} \phi^2$ where $\mu$ is a mass parameter and $\phi$ is an $\mathcal{N} = 1$ adjoint chiral superfield inside the $\mathcal{N} = 2$ vector multiplet. For masses $\mu \ll \Lambda_{\mathcal{N}=2}$ much smaller than the $\mathcal{N} = 2$ scale the superpotential is represented as $\mu U$ where $U$ is an $\mathcal{N} = 1$ chiral superfield whose scalar component corresponds to the CB modulus $u$. It is well-known [47] that this superpotential lifts the entire CB except for the monopole and dyon points, and thus the theory has two massive vacua, which we refer to as the monopole vacuum and the dyon vacuum respectively. The superpotential leads to condensation of monopoles at the monopole point and the condensation of dyons (whose charges align with $W + H$) at the dyon point. This has the following consequence for confinement in the three theories:

- For the $SU(2)$ theory, since the charge of the chosen line operator $W$ does not align with the charges of condensing monopoles or dyons, $W$ exhibits area law in both vacua. Hence, both vacua are confining and preserve the $\mathbb{Z}_2$ 1-form symmetry.

- For the $SO(3)_+$ theory in the monopole vacuum, the charge of the chosen line operator $H$ aligns with the charge of condensing monopoles, and hence $H$ exhibits perimeter law in the monopole vacuum. On the other hand, the charge of $H$ does not align with the charge of condensing dyons in the dyon vacuum, and hence $H$ exhibits area law in the dyon vacuum. Thus, the monopole vacuum is not confining and spontaneously breaks the $\mathbb{Z}_2$ 1-form symmetry, while the dyon vacuum is confining and preserves the $\mathbb{Z}_2$ 1-form symmetry.

- For the $SO(3)_-$ theory in the dyon vacuum, the charge of the chosen line operator $W + H$ aligns with the charge of condensing dyons, and hence $W + H$ exhibits perimeter law in the dyon vacuum. On the other hand, the charge of $W + H$ does not align with the charge of condensing monopoles in the monopole vacuum, and hence $W + H$ exhibits area law in the monopole vacuum. Thus, the dyon vacuum is not confining and spontaneously breaks the $\mathbb{Z}_2$ 1-form symmetry, while the monopole vacuum is confining and preserves the $\mathbb{Z}_2$ 1-form symmetry.

In a confining vacuum there are massive confining strings which are charged under the 1-form symmetry, while in a non-confining vacuum there are no such strings.

For $\mu \gg \Lambda_{\mathcal{N}=2}$ we can rely on the UV description of the $\mathcal{N} = 2$ theory to integrate out $\Phi$ thus leading to the pure $\mathcal{N} = 1$ SYM theory at low-energies, which is expected to admit two

---

[3]The $SO(3)_+$ theory is obtained from the $SU(2)$ theory by gauging its $\mathbb{Z}_2$ 1-form symmetry, while the $SO(3)_-$ theory is obtained by gauging the diagonal $\mathbb{Z}_2$ 1-form symmetry of the $SU(2)$ theory stacked with an SPT phase for the $\mathbb{Z}_2$ 1-form symmetry. After gauging, the SPT phase is understood as a discrete theta-like parameter which when added to the Lagrangian of the $SO(3)_+$ theory leads to the $SO(3)_-$ theory and vice versa. The above construction in terms of the $SU(2)$ theory allows us to call the $SO(3)_+$ theory as the theory with discrete theta parameter "turned off", and the $SO(3)_-$ theory as the theory with discrete theta parameter "turned on" [2].

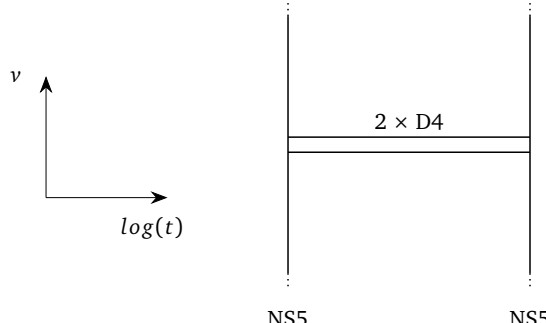

Figure 1: The Hanany-Witten setup realizing in Type IIA string theory the pure $\mathcal{N} = 2$ $\mathfrak{su}(2)$ SYM theories.

massive vacua. This ties in neatly with the two vacua found for $\mu \ll \Lambda_{\mathcal{N}=2}$ and suggests that there is no phase transition as we vary $\mu/\Lambda_{\mathcal{N}=2}$. Moreover, we are lead to the prediction that both the vacua of the $SU(2)$ pure $\mathcal{N} = 1$ SYM are confining, while for the $SO(3)_{\pm}$ pure $\mathcal{N} = 1$ SYM only one of the vacua is confining.

## 2.2 Construction from 6d $A_1$ $\mathcal{N} = (2,0)$ Theory

The pure $\mathcal{N} = 2$ theory admits a Hanany-Witten type brane construction in terms of NS5 and D4 branes in Type IIA superstring theory [48,49], which is shown in figure 1. This allows us to read off the Seiberg-Witten curve (SW curve) of the theory

$$v^2 = \frac{\Lambda^2}{t} + u + \Lambda^2 t \,, \tag{5}$$

where we have used the notation of [48] and we have shortened $\Lambda_{\mathcal{N}=2}$ used in the previous subsection to $\Lambda$.

Now, following [4,6], one can use the above SW curve to phrase the construction of the pure $\mathcal{N} = 2$ theory as a compactification of 6d $A_1$ $\mathcal{N} = (2,0)$ superconformal field theory (SCFT). The $(2,0)$ theory contains a chiral operator[4] that we denote as $\text{Tr}\Phi^2$ which transforms in an irreducible representation of the $\mathfrak{so}(5)_R$ symmetry. Vevs of this chiral operator parametrize the moduli space of supersymmetric vacua of the $(2,0)$ theory.

To construct the pure 4d $\mathcal{N} = 2$ theory, we need to compactify the $A_1$ $(2,0)$ theory on a sphere $\mathcal{C}$ with two punctures whose coordinate is $t$ and the punctures are located at $t = 0, \infty$. $\mathcal{C}$ is the Gaiotto curve for this Class S construction. We need to perform a topological twist along $\mathcal{C}$ which decomposes $\mathfrak{so}(5)_R \to \mathfrak{so}(3)_R \oplus \mathfrak{so}(2)_R$ and identifies $\mathfrak{so}(2)_R$ with the $\mathfrak{so}(2)$ holonomy group of $\mathcal{C}$, while identifying $\mathfrak{so}(3)_R$ as the R-symmetry of the descendant 4d $\mathcal{N} = 2$ theory. This decomposes $\text{Tr}\Phi^2$ into various operators, out of which we single out an operator $\text{Tr}\phi^2$ which is charged only under the $\mathfrak{so}(2)_R$ subalgebra of $\mathfrak{so}(5)_R$ and transforms as a quadratic differential on $\mathcal{C}$ due to the twist. The vevs

$$\phi_2 := \langle \text{Tr}\phi^2 \rangle \,, \tag{6}$$

are quadratic meromorphic differentials on $\mathcal{C}$ and parametrize the CB of the resulting 4d $\mathcal{N} = 2$ theory. To fully specify the space of $\phi_2(t)$, we need to specify their behavior at the punctures $t = 0, \infty$. This can be read off by identifying the SW curve as

$$\phi_2 = \frac{v^2}{t^2} dt^2 = \left( \frac{\Lambda^2}{t^3} + \frac{u}{t^2} + \frac{\Lambda^2}{t} \right) dt^2 \,, \tag{7}$$

---

[4]It should be noted that while $\text{Tr}\Phi^2$ is a genuine local operator in the 6d theory, $\Phi$ is not.

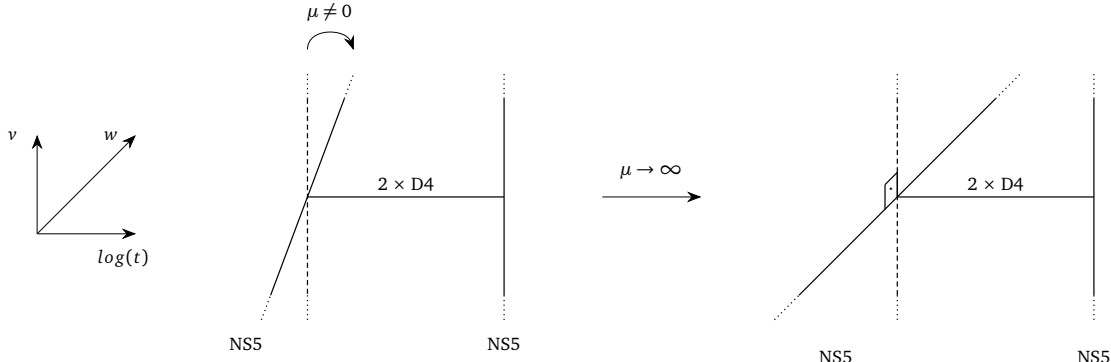

Figure 2: Rotation of the branes in the Hanany-Witten setup, which results in breaking $\mathcal{N} = 2$ to $\mathcal{N} = 1$ supersymmetry. Again we show the construction for $\mathfrak{su}(2)$ pure SYM.

from which follows that $\phi_2$ has poles of order 3 at both punctures. The corresponding Higgs field $\phi$ has a pole of order $\frac{3}{2}$ at each puncture. Since this is higher-order singularity than a simple pole of order 1, it is called as an *irregular* singularity of the Higgs field. Thus, the 4d pure $\mathfrak{su}(2)$ $\mathcal{N} = 2$ SYM is constructed by compactifying the 6d $A_1$ $(2,0)$ theory on a sphere with two irregular punctures where the Higgs field has a pole of order $\frac{3}{2}$ at each puncture.

Now, to reach the 4d $\mathcal{N} = 1$ SYM theory, we would like to deform the 4d $\mathcal{N} = 2$ SYM theory by the superpotential discussed in the previous subsection. In the Type IIA brane construction, this corresponds to rotating one of the NS5 branes in two of the transverse directions denoted by a complex coordinate $w$ [10, 11, 50]. The $\mu \to \infty$ limit which leads to the 4d $\mathcal{N} = 1$ SYM theory is obtained when the NS5 brane has been completely rotated. See figure 2.

To achieve this deformation from the point of view of the 6d $A_1$ $(2,0)$ theory, we start by choosing an $\mathfrak{so}(2)_w$ inside $\mathfrak{so}(3)_R$ which corresponds to choosing an operator $\text{Tr}\varphi^2$ inside $\text{Tr}\Phi^2$ which is charged only under $\mathfrak{so}(2)_w$ subalgebra of $\mathfrak{so}(5)_R$. Then we turn on vevs

$$\varphi_2 := \langle \text{Tr}\varphi^2 \rangle, \tag{8}$$

which are meromorphic functions on $\mathcal{C}$. Since $\text{Tr}\varphi^2$ is charged under $\mathfrak{so}(3)_R$, its vevs break the $\mathcal{N} = 2$ R-symmetry in 4d, and hence the resulting theory only has 4d $\mathcal{N} = 1$ supersymmetry. The asymptotic values of the Higgs field[5] $\varphi$ are

$$\begin{aligned} \varphi &\sim \mu V &\text{as } t \to \infty \\ \varphi &\sim c &\text{as } t \to 0 \end{aligned}, \tag{9}$$

where $c$ is a constant and $V$ is a Higgs field defined via

$$V \frac{dt}{t} = \phi. \tag{10}$$

Thus $\varphi$ has a singularity only at $t = \infty$ but not at $t = 0$, which encodes the fact that we are "rotating" the puncture at $t = \infty$ but not the puncture at $t = 0$.

Turning on vevs for $\text{Tr}\phi^2$ and $\text{Tr}\varphi^2$ also turns on vevs for operator $\text{Tr}\phi\varphi$ sitting inside $\text{Tr}\Phi^2$, which is charged under both $\mathfrak{so}(2)_R$ and $\mathfrak{so}(2)_w$. The vevs

$$(\phi\varphi)_2 := \langle \text{Tr}\phi\varphi \rangle, \tag{11}$$

are meromorphic 1-forms on $\mathcal{C}$ due to the topological twist.

---

[5]Notice that the Higgs field $\varphi$ is a function on $\mathcal{C}$, while the Higgs field $\phi$ is a 1-form.

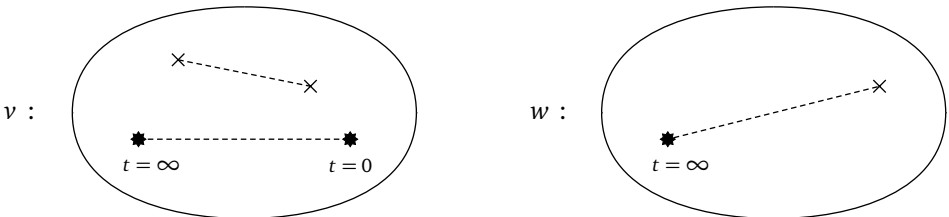

Figure 3: The figure displays the sheet structures of $v$-curve and $w$-curve over the Gaiotto curve $\mathcal{C}$ with the coordinate $t$. A star denotes a singular point where the corresponding curve escapes to infinity, $\times$ denotes a non-singular branch-point, and the dashed lines denote branch cuts.

The SW curve can now be transformed to an "$\mathcal{N} = 1$ curve" specified by

$$
\begin{aligned}
v^2 \frac{dt^2}{t^2} &= \phi_2 , \\
w^2 &= \varphi_2 , \\
vw \frac{dt}{t} &= (\phi\varphi)_2 ,
\end{aligned}
\tag{12}
$$

with $(t, v, w) \in \mathbb{C}^* \times \mathbb{C} \times \mathbb{C}$, which is a 2-fold cover of the Gaiotto curve $\mathcal{C}$. In particular, the last equation in the above set of equations combines the double covers associated to $v$ and $w$ into a single double cover. This equation can be imposed because $\phi$ and $\varphi$ are simultaneously diagonalizable and hence $\mathrm{Tr}(\phi\varphi)$ defines a consistent profile for $vw\frac{dt}{t}$.

It turns out that the set of equations (12) is consistent only for two values of the CB parameter $u$ [8]. To see this, we first write down the most general form of $w^2$ compatible with the boundary condition (9)

$$
w^2 = \mu^2 \Lambda^2 t + c .
\tag{13}
$$

Now the well-defined-ness of $vw$ requires us to impose that the product of (5) and (13) is a perfect square, that is

$$
v^2 w^2 = \frac{\left(\Lambda^2 t^2 + ut + \Lambda^2\right)\left(\mu^2 \Lambda^2 t + c\right)}{t} = \frac{P^2(t)}{Q^2(t)} .
\tag{14}
$$

Keeping $\Lambda$ fixed, the only way to achieve the perfect square condition is by requiring

$$
\begin{aligned}
c &= 0 , \\
u &= \pm 2\Lambda^2 .
\end{aligned}
\tag{15}
$$

Thus, we rediscover that after a rotation only two vacua survive, which are the monopole and the dyon points in the $\mathcal{N} = 2$ CB. The $\mathcal{N} = 1$ curves for these two vacua are

$$
\begin{aligned}
v^2 &= \Lambda^2 \left( \frac{1}{t} \pm 2 + t \right) , \\
w^2 &= \mu^2 \Lambda^2 t , \\
vw &= \mu\Lambda^2 . \qquad (t \pm 1)
\end{aligned}
\tag{16}
$$

The perfect square condition, which is equivalent to $vw$ being well-defined, can be understood in a more topological/group-theoretical way that will be very useful for further generalizations later in the paper. If we only consider (5) and (13), we find that in general there are 4 values of $(v, w)$ associated to a fixed value of $t$. That is, we are describing two separate double

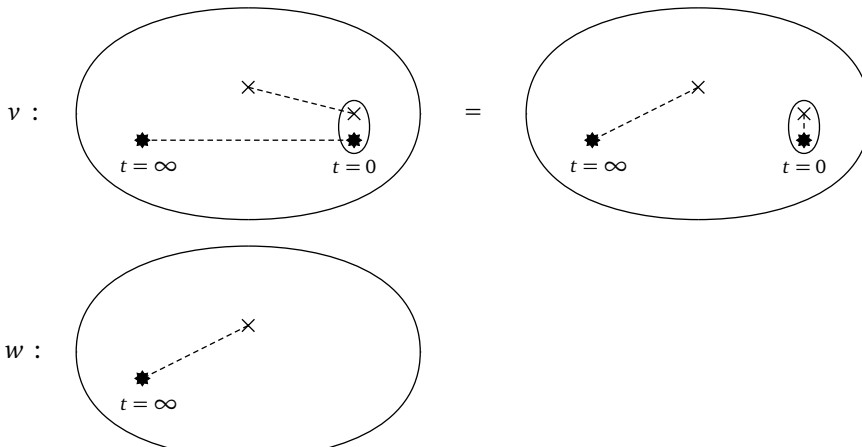

Figure 4: A possible movement and collision of branch points that ensures that the branch structures of $v$ and $w$ curves coincide. However, as explained in the text, this branch structure is not possible.

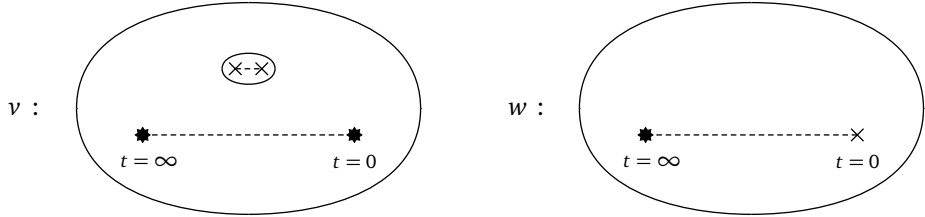

Figure 5: A possible movement and collision of branch points that ensures that the branch structures of $v$ and $w$ curves coincide. This configuration gives rise to a consistent $\mathcal{N} = 1$ curve.

covers $v, w$ of the Gaiotto curve $\mathcal{C}$ parametrized by $t$. Instead, we would like to combine these two double covers into a single double cover of $\mathcal{C}$, and the combined curve is the $\mathcal{N} = 1$ curve we are after. We can represent the sheet structures of the two double covers (5) and (13) in terms of branch points and the branch lines (i.e. branch cuts) joining them, as shown in figure 3.

To combine these two double covers, we need to move the branch points and potentially collide them such that the resulting branch structure of the two double covers is the same. One such possible movement of branch points is shown in figure 4. From the point of view of (5), this movement requires that $t = 0$ is a root of $\Lambda^2 t^2 + ut + \Lambda^2$ which is not possible for a fixed $\Lambda$, and requires us to change our starting $\mathcal{N} = 2$ theory. Thus, we reject this movement of branch points.

However, there is another possible movement of the branch points which results in the same branch structure for the two double covers. See figure 5. From the point of view of (5), this movement requires colliding the two branch points of $\Lambda^2 t^2 + ut + \Lambda^2$; and from the point of view of (13), this movement requires sending the branch point of (13) located at finite $t$ to be moved to $t = 0$. Thus, this movement enforces precisely the conditions (15) which lead to the $\mathcal{N} = 1$ curve (16).

Now, even though both the $\mathcal{N} = 1$ curves (16) have the same structure of branch points, they have different structure of branch lines. This can be seen easily by analyzing the behavior of $vw$ as $t \to 0$. In this limit we can write

$$vw = \pm \mu \Lambda^2 \,. \tag{17}$$

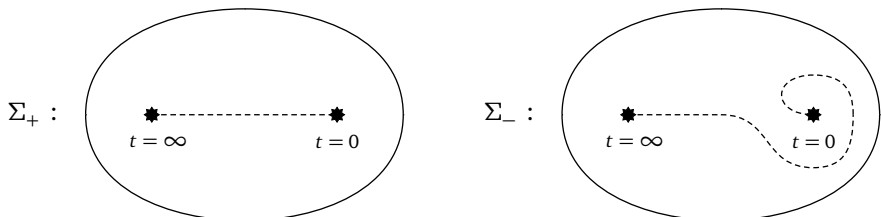

Figure 6: Branch-cut structure for the two $\mathcal{N} = 1$ curves for $\mathfrak{su}(2)$ SYM. The figures show the projections of the $\mathcal{N} = 1$ curves onto the Gaiotto curve (parametrized by $t$). The right hand figure is related to the left by a Dehn twist. These two curves characterize the two vacua of the theory.

So, for a fixed asymptotic value of $v$, the two curves have asymptotic values of $w$ having opposite signs. From the equation $w^2 = \mu^2 \Lambda^2 t$, we see that we can change the sign of asymptotic value of $w$ by encircling once the $t = 0$ point. Thus, the branch cuts of the two $\mathcal{N} = 1$ curves (seen as double covers of $\mathcal{C}$) are related by a Dehn twist around the puncture at $t = 0$. If we choose to represent the branch cut for the $\mathcal{N} = 1$ curve (16) with plus sign as in the left side of figure 6, then the branch cut for the $\mathcal{N} = 1$ curve (16) with minus sign is as shown on the right side of figure 6.

Taking an appropriate $\mu \to \infty$ limit of (16) leads to $\mathcal{N} = 1$ curves for the two vacua of pure $\mathfrak{su}(2)$ $\mathcal{N} = 1$ SYM [11]. Under this limit, the following quantities are kept fixed

$$
\begin{aligned}
\Lambda_{\mathcal{N}=1}^3 &:= \mu \Lambda^2, \\
\widetilde{t} &:= \mu t,
\end{aligned}
\tag{18}
$$

and after the limit we obtain the following $\mathcal{N} = 1$ curves

$$
\begin{aligned}
v^2 &= \frac{\Lambda_{\mathcal{N}=1}^3}{\widetilde{t}}, \\
w^2 &= \Lambda_{\mathcal{N}=1}^3 \widetilde{t}, \\
vw &= \pm \Lambda_{\mathcal{N}=1}^3.
\end{aligned}
\tag{19}
$$

Notice that the structure of branch points and cuts of the above two $\mathcal{N} = 1$ curves (viewed as double covers of the $\widetilde{t}$ plane) is exactly the same as in figure 6, i.e. the structure of branch points and cuts is left unchanged in the $\mu \to \infty$ limit.

## 2.3 Confinement from the 6d Construction

In this subsection, we apply our results from [29], and first discuss how the group of line operators $\mathcal{L}$ (2) is encoded in the cycles on the Gaiotto curve $\mathcal{C}$. Different theories $SU(2)$ and $SO(3)_{\pm}$ correspond to different subgroups $\Lambda$ of $\mathcal{L}$. The 1-form symmetry group is then identified with the Pontryagin dual[6] $\widehat{\Lambda}$ of $\Lambda$. Let $\mathcal{I}_r \subseteq \mathcal{L}$ be defined as

$$
\mathcal{I}_r = \{\text{projections of 1-cycles on the } \mathcal{N} = 1 \text{ curve } \Sigma_r \text{ for vacuum } r \text{ onto } \mathcal{C}\}. \tag{20}
$$

Then we propose that the 1-form symmetry group $\mathcal{O}_r$ preserved in the vacuum $r$ can be identified with

$$
\mathcal{O}_r = \widehat{\left(\frac{\Lambda}{\mathcal{I}_r|_\Lambda}\right)} \subseteq \widehat{\Lambda}, \tag{21}
$$

---

[6]Pontryagin dual group $\widehat{G}$ of an abelian group $G$ is the group formed by 1-dimensional representations of $G$ under tensor product operation.

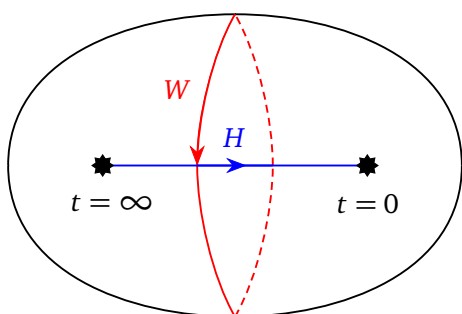

Figure 7: The Wilson ($W$, red) and 't Hooft ($H$, blue) lines in the Class S realization of the 4d $\mathcal{N} = 2$ $\mathfrak{su}(2)$ pure SYM theories.

where $\mathcal{I}_r|_\Lambda := \mathcal{I}_r \cap \Lambda$ and a hat on top of a group denotes the Pontryagin dual of that group. If $\mathcal{O}_r$ is trivial, then the vacuum $r$ is not confining. On the other hand, if $\mathcal{O}_r$ is non-trivial then the vacuum $r$ is confining.

The group $\mathcal{L}$ in the 4d $\mathcal{N} = 2$ $\mathfrak{su}(2)$ SYM theory descends from a similar group $\widehat{Z} \simeq \mathbb{Z}_2$ formed by dimension-2 surface operators (modulo screenings) in the 6d $A_1$ $(2,0)$ theory. Let us denote the non-trivial element of $\widehat{Z}$ by $f$, which is non-local with itself

$$\langle f, f \rangle = \frac{1}{2}. \tag{22}$$

As proposed in [29], after compactifying on $\mathcal{C}$, the set $\mathcal{L}$ (2) descends from $\widehat{Z}$ as shown in figure 7. Compactifying $f$ on the cycle $W$ (which encircles both punctures) leads to the element named $W \in \mathcal{L}$, and compactifying $f$ on the cycle $H$ (which extends between the two punctures) leads to the element named $H \in \mathcal{L}$. The pairing (3) on $\mathcal{L}$ is obtained by combining the pairing (22) on $\widehat{Z}$ with the intersection pairing on $\mathcal{C}$.

As we discussed earlier, the various global forms of the gauge group are distinguished as follows, where $[W]$ denotes the subgroup generated by $W \in \mathcal{L}$, etc:

- $SU(2)$ theory is obtained by choosing $\Lambda = [W] \subset \mathcal{L}$,

- $SO(3)_+$ theory is obtained by choosing $\Lambda = [H] \subset \mathcal{L}$,

- $SO(3)_-$ theory is obtained by choosing $\Lambda = [W + H] \subset \mathcal{L}$.

As we deform the $\mathcal{N} = 2$ theory, the sets $\Lambda$ and $\mathcal{L}$ remain same. That is, our encoding of $\Lambda$ and $\mathcal{L}$ into the Gaiotto curve $\mathcal{C}$ should hold even after rotating one of the punctures. This should continue to hold even as we take the limit $\mu \to \infty$, with the role of $\mathcal{C}$ played by the $\tilde{t}$-plane.

We can now study the subgroups $\mathcal{I}_r$ for the two vacua obtained after the deformation. For the vacuum $r = +$ with the plus sign in (16), this is encoded in the left side of figure 6 which depicts the projection of the corresponding $\mathcal{N} = 1$ curve $\Sigma_+$ onto $\mathcal{C}$. Because of the branch cut extending between the two punctures, one must go around a puncture twice to obtain a cycle on $\Sigma_+$. This cycle projects to $2W \equiv 0 \in \mathcal{L}$. Traveling from one puncture to the other along one side of the branch cut, we obtain another cycle on $\Sigma_+$ which projects to $H \in \mathcal{L}$. Thus, we find that

$$\mathcal{I}_+ = [H]. \tag{23}$$

For the vacuum $r = -$ with the minus sign in (16), $\mathcal{I}_-$ is encoded in the right side of figure 6 which depicts the projection of the corresponding $\mathcal{N} = 1$ curve $\Sigma_-$ onto $\mathcal{C}$. Again, because of the branch cut, one must go around a puncture twice to obtain a cycle on $\Sigma_-$ which projects to $2W \equiv 0 \in \mathcal{L}$. On the other hand, traveling from one puncture to the other along one side

of the branch cut, now we obtain a cycle on $\Sigma_-$ which projects to $W + H \in \mathcal{L}$. Thus, we find that

$$\mathcal{I}_- = [W + H]. \tag{24}$$

From these, we readily compute for the $SU(2)$ theory the 1-form symmetry that is preserved in each of the vacua is

$$\begin{aligned}
\mathcal{O}_+ &\simeq \mathbb{Z}_2, \\
\mathcal{O}_- &\simeq \mathbb{Z}_2.
\end{aligned} \tag{25}$$

That is, both vacua $r = \pm$ are confining for the $SU(2)$ theory. For the $SO(3)_+$ theory, we find

$$\begin{aligned}
\mathcal{O}_+ &\simeq 0, \\
\mathcal{O}_- &\simeq \mathbb{Z}_2.
\end{aligned} \tag{26}$$

That is, the monopole vacuum $r = +$ is not confining, while the dyon vacuum $r = -$ is confining. For the $SO(3)_-$ theory, we find

$$\begin{aligned}
\mathcal{O}_+ &\simeq \mathbb{Z}_2, \\
\mathcal{O}_- &\simeq 0.
\end{aligned} \tag{27}$$

That is, the monopole vacuum $r = +$ is confining, while the dyon vacuum $r = -$ is not confining. Thus, our proposal (21) recovers the field theory results discussed in subsection 2.1.

Since the branch structure (over the $\widetilde{t}$-plane) of the $\mathcal{N} = 1$ curves (19) in the $\mu \to \infty$ limit is also described by the figure 6, the above results about $\mathcal{O}_\pm$ remain the same even after the $\mu \to \infty$ limit.

# 3 $\mathcal{N} = 1$ Hitchin Systems and Confinement

## 3.1 Confinement, 1-form Symmetries, and Relative and Absolute Theories

In this paper we study 4d "relative" theories which are not genuine 4d theories as they live at the boundaries of some genuine 5d topological theories. These 4d theories admit a set $\mathcal{L}$ of line operators modulo screenings and flavor charges, with the key property that not all line operators in $\mathcal{L}$ are mutually local with each other[7]. We will only study theories for which $\mathcal{L}$ is an abelian group under OPE of line operators[8]. The group $\mathcal{L}$ carries a "pairing" which is a bihomomorphism

$$\langle \cdot, \cdot \rangle : \mathcal{L} \times \mathcal{L} \to \mathbb{R}/\mathbb{Z}. \tag{28}$$

The pairing $\langle \alpha, \beta \rangle$ captures the non-locality between two line operators $\alpha, \beta \in \mathcal{L}$. As we move $\alpha$ and $\beta$ around each other such that the starting and ending configurations are the same, the correlation function returns back to itself times the phase factor

$$\exp\left(2\pi i \langle \alpha, \beta \rangle\right). \tag{29}$$

An "absolute" 4d theory is a genuine 4d theory that chooses a maximal subgroup[9] $\Lambda \subset \mathcal{L}$ of line operators such that the pairing $\langle \cdot, \cdot \rangle$ restricted to $\Lambda$ vanishes[10] [46]. This is done by specifying a topological boundary condition of the 5d topological theory, and reducing the 5d theory on a segment with one end occupied by the relative 4d theory and the other end

---

[7]It is also possible for the relative theory to also have operators of other dimensions which are non-local to each other, but they are not the focus of this work.

[8]The group $\mathcal{L}$ of line defects modulo screenings may not form an abelian group, but an algebra like structure if the theory carries non-invertible 1-form symmetries. An example is provided by the $O(2)$ gauge theory [51].

[9]Such a subgroup is often referred to as "maximal isotropic subgroup" or as "polarization".

[10]An absolute theory also chooses maximal mutually local subsets of operators of other dimensions.

occupied by the chosen boundary condition. The resulting absolute 4d theory $\mathfrak{T}$ has a group of genuine line operators (modulo screenings) given by $\Lambda$ while the other line operators in $\mathcal{L}$ remain as non-genuine line operators constrained to live at the ends of topological surface operators generating the 1-form symmetry $\widehat{\Lambda}$ of $\mathfrak{T}$, where is the Pontryagin dual of $\Lambda$. The action of the 1-form symmetry is then responsible for the phase factor (29).

The vacua are independent of the choice of $\Lambda$ and so the vacua can be associated to the relative 4d theory. A vacuum $r$ of a relative theory divides line operators in $\mathcal{L}$ into two distinct sets: those showing perimeter law and those showing area law. The subset of line operators exhibiting perimeter law form a subgroup $\mathcal{I}_r \subseteq \mathcal{L}$. For an absolute QFT having (genuine) line operators specified by a polarization $\Lambda$, the subgroup $\Lambda_r := \mathcal{I}_r \cap \Lambda \subseteq \Lambda$ of line operators show perimeter law. Then, the vacuum $r$ preserves a subgroup

$$
\mathcal{O}_r := \widehat{\left(\frac{\Lambda}{\Lambda_r}\right)} \subseteq \widehat{\Lambda}, \tag{30}
$$

of the 1-form symmetry group $\widehat{\Lambda}$. If $\mathcal{O}_r$ is trivial, it is said that the vacuum $r$ is not confining. On the other hand, if $\mathcal{O}_r$ is non-trivial, the vacuum $r$ is said to be *confining*. Moreover, if $\mathcal{O}_r \simeq \mathbb{Z}_t$, then $t$ is called the *confinement index* of the vacuum $r$ [21,52].

## 3.2    1-form Symmetry for $A_{n-1}$ Class S Theories

The class of 4d theories we study in this paper are related to 4d $\mathcal{N} = 2$ theories of Class S obtained by compactifying 6d $A_{n-1}$ $(2,0)$ SCFT on a punctured Riemann surface $\mathcal{C}_g$ of genus $g$ with arbitrary (untwisted) punctures but without any outer-automorphism twists. The line operators $\mathcal{L}$ in this class of theories arise by wrapping dimension-2 surface operators along 1-cycles of $\mathcal{C}_g$. For $A_{n-1}$ $(2,0)$ theory, the dimension-2 surface operators modulo screenings form a group $\widehat{Z} \simeq \mathbb{Z}_n$. The group $\widehat{Z}$ carries a non-trivial pairing $\langle \cdot, \cdot \rangle$ capturing the non-locality between the dimension-2 surface operators. Choosing a generator $f \in \widehat{Z}$, this pairing can be written as

$$
\langle f, f \rangle = \frac{1}{n}. \tag{31}
$$

Thus the $(2,0)$ theory is a relative QFT in the language of section 3.1. Its compactification on $\mathcal{C}_g$ gives rise to a *relative* 4d $\mathcal{N} = 2$ Class S theory.

Apriori, the possible ways of wrapping dimension-2 operators along 1-cycles of $\mathcal{C}_g$ are described by $H_1(\mathcal{C}_g, \widehat{Z}, *)$, which is the homology group of 1-cycles (with coefficients in $\widehat{Z}$) that are allowed to end on punctures, indicated by $*$. See figure 8. On the other hand, let $H_1(\mathcal{C}_g, \widehat{Z})$ denote the homology group of 1-cycles (with coefficients in $\widehat{Z}$) that do not end on punctures. Clearly $H_1(\mathcal{C}_g, \widehat{Z}) \subseteq H_1(\mathcal{C}_g, \widehat{Z}, *)$. Now, it is not possible for all dimension-2 operators to end on every puncture. Given a specific puncture $p$ of type $\mathcal{P}$, only a subgroup $Z_{\mathcal{P}} \subseteq \widehat{Z}$ of dimension-2 surface operators can end at $p$. So, let $\mathcal{S}$ be the subgroup of $H_1(\mathcal{C}_g, \widehat{Z}, *)$ such that the coefficient $\alpha \in \widehat{Z}$ associated to a 1-cycle in $\mathcal{S}$ ending at a puncture of type $\mathcal{P}$ is such that $\alpha \in Z_{\mathcal{P}}$. See figure 9. The physical interpretation of $\mathcal{S}$ is the subgroup of 1-cycles that can be wrapped by the dimension-2 operators.

$\mathcal{S}$ is not straightforwardly identified with the group $\mathcal{L}$ of 4d line operators, as $\mathcal{S}$ also captures flavor charges of 4d line operators but the flavor charges are not part of the data tracked by $\mathcal{L}$. The data of flavor charges is modded out by identifying certain elements of $\mathcal{S}$ resulting in a projection map

$$
\pi : \quad \mathcal{S} \to \mathcal{L}. \tag{32}
$$

The elements of $\mathcal{S}$ that are modded are described as follows. Consider a 1-cycle $L_p$ encircling a puncture $p$ of type $\mathcal{P}$. Wrapping dimension-2 surface operators along $L_p$ we generate

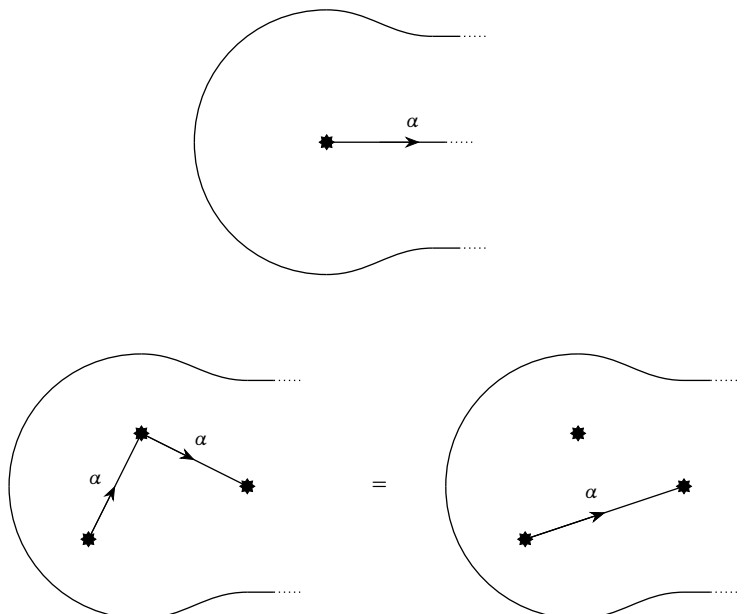

Figure 8: Top: The cycles in $H_1(\mathcal{C}_g, \widehat{Z}, *)$ are allowed to end on punctures. $\alpha$ denotes some element of $\widehat{Z}$. Bottom: Composition rule for cycles that end on punctures.

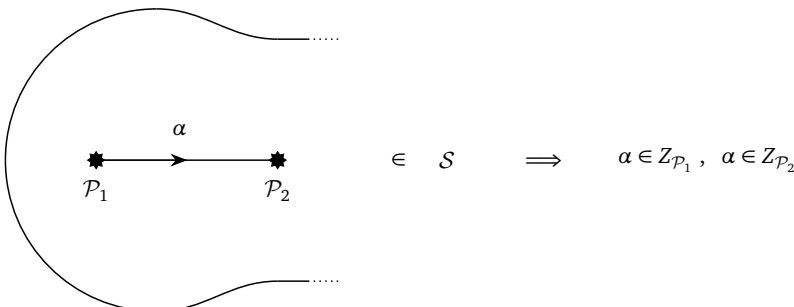

Figure 9: If a 1-cycle in $\mathcal{S}$ carrying $\alpha \in \widehat{Z}$ ends on punctures of types $\mathcal{P}_1$ and $\mathcal{P}_2$, then we must have $\alpha \in Z_{\mathcal{P}_1}$ and $\alpha \in Z_{\mathcal{P}_2}$.

a subgroup $\bar{Z}_p \simeq \widehat{Z}$ of $\mathcal{S}$. Then, depending on the type $\mathcal{P}$ of the puncture $p$, a subgroup $\bar{Z}_{\mathcal{P},p} \subseteq \bar{Z}_p \subseteq \mathcal{S}$ is modded out where $\bar{Z}_{\mathcal{P},p} \simeq \bar{Z}_{\mathcal{P}}$ and $\bar{Z}_{\mathcal{P}}$ is a subgroup of dimension-2 operators $\widehat{Z}$. See figure 10.

The pairing on $\mathcal{L}$ can be determined in terms of pairing on $\widehat{Z}$ and the intersection pairing of 1-cycles. First of all, combining these two pairings we obtain a pairing on $H_1(\mathcal{C}_g, \widehat{Z}, *) \simeq H_1(\mathcal{C}_g, \mathbb{Z}, *) \otimes \widehat{Z}$. For two elements $a \otimes \alpha, b \otimes \beta \in H_1(\mathcal{C}_g, \mathbb{Z}, *) \otimes \widehat{Z}$ the pairing is written as

$$\langle a \otimes \alpha, b \otimes \beta \rangle = \langle a, b \rangle \langle \alpha, \beta \rangle, \tag{33}$$

with $\langle a, b \rangle$ determined using the intersection pairing and $\langle \alpha, \beta \rangle$ determined using the pairing on $\widehat{Z}$. The above pairing is then extended by linearity to all of $H_1(\mathcal{C}_g, \widehat{Z}, *)$. Since $\mathcal{S} \subseteq H_1(\mathcal{C}_g, \widehat{Z}, *)$, we obtain a pairing on $\mathcal{S}$ by simply restricting the pairing on $H_1(\mathcal{C}_g, \widehat{Z}, *)$. Now we would like to push-forward the pairing on $\mathcal{S}$ to a pairing on $\mathcal{L}$ using the projection map (32). This can only be done consistently if

$$\langle \alpha, \beta \rangle = 0, \tag{34}$$

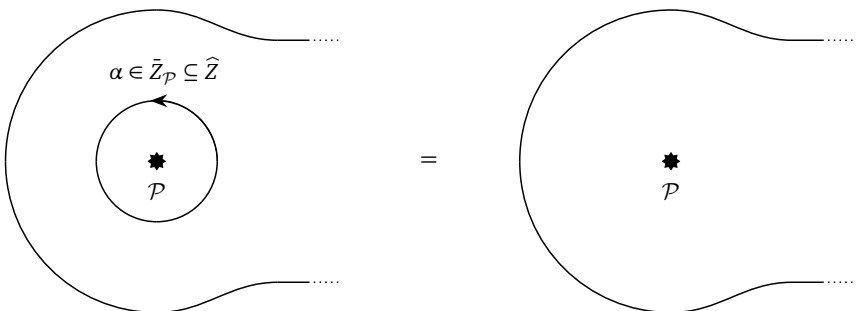

Figure 10: A 1-cycle surrounding a puncture of type $\mathcal{P}$ and carrying $\alpha \in \bar{Z}_{\mathcal{P}} \subseteq \widehat{Z}$ is trivial in $\mathcal{L}$.

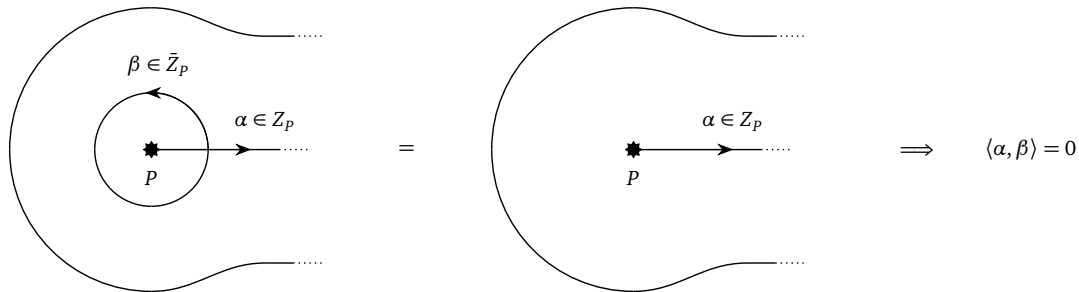

Figure 11: A consistent pairing on $\mathcal{L}$ exists only if the mutual pairing between elements of $\bar{Z}_{\mathcal{P}}$ and $Z_{\mathcal{P}}$ vanishes.

for all $\alpha \in Z_{\mathcal{P}} \subseteq \widehat{Z}$ and $\beta \in \bar{Z}_{\mathcal{P}} \subseteq \widehat{Z}$, and for all $\mathcal{P}$. See figure 11. Thus, the well-defined-ness of pairing on $\mathcal{L}$ imposes the above constraint on the subgroups $Z_{\mathcal{P}}, \bar{Z}_{\mathcal{P}}$ for all puncture types $\mathcal{P}$. Once this condition is satisfied, we obtain a pairing on $\mathcal{L}$ as a push-forward of (33).

This determines $\mathcal{L}$ and pairing $\langle \cdot, \cdot \rangle$ on $\mathcal{L}$ for the relative Class S theory arising from the above compactification. As discussed in section 3.1, an absolute Class S theory arising from this compactification chooses a maximal subgroup $\Lambda \subset \mathcal{L}$ such that the pairing restricted to $\Lambda$ is trivial. The 1-form symmetry of such an absolute Class S theory is $\widehat{\Lambda}$ which is the Pontryagin dual of $\Lambda$.

### 3.2.1 Data Associated to Various Punctures

Let us now collect information about $Z_{\mathcal{P}}, \bar{Z}_{\mathcal{P}}$ for various types of punctures that can arise in untwisted compactifications of 6d $A_{n-1}$ $(2,0)$ theory. The punctures can be divided into two types: the regular punctures for which the $\mathcal{N}=2$ Hitchin field $\phi$ has at most a simple pole, and the irregular punctures for which $\phi$ has higher-order poles.

For regular punctures, it was argued in [29] that in this case we have[11]

$$Z_{\mathcal{P}} = 0 \subset \widehat{Z} \,,$$
$$\bar{Z}_{\mathcal{P}} = \widehat{Z} \,. \tag{35}$$

The constraint (34) is trivially satisfied. In other words, no element of $\widehat{Z}$ can end on a regular

---

[11]A first-principles way to see that this must be the case is to realize the puncture as a boundary condition of 5d $\mathcal{N}=2$ $\mathfrak{su}(n)$ SYM theory. A surface operator of the 6d theory wrapping a loop encircling the puncture becomes a gauge Wilson line of the 5d theory. For a regular puncture, the associated boundary condition is such that the 5d dynamical gauge field becomes a background gauge field at the 4d boundary. Consequently, every gauge Wilson line of the 5d theory reduces to a flavor Wilson line at the 4d boundary, which does not contribute to $\mathcal{L}$ and hence $\bar{Z}_{\mathcal{P}} = \widehat{Z}$. $Z_{\mathcal{P}} = 0$ is now fixed by (34).

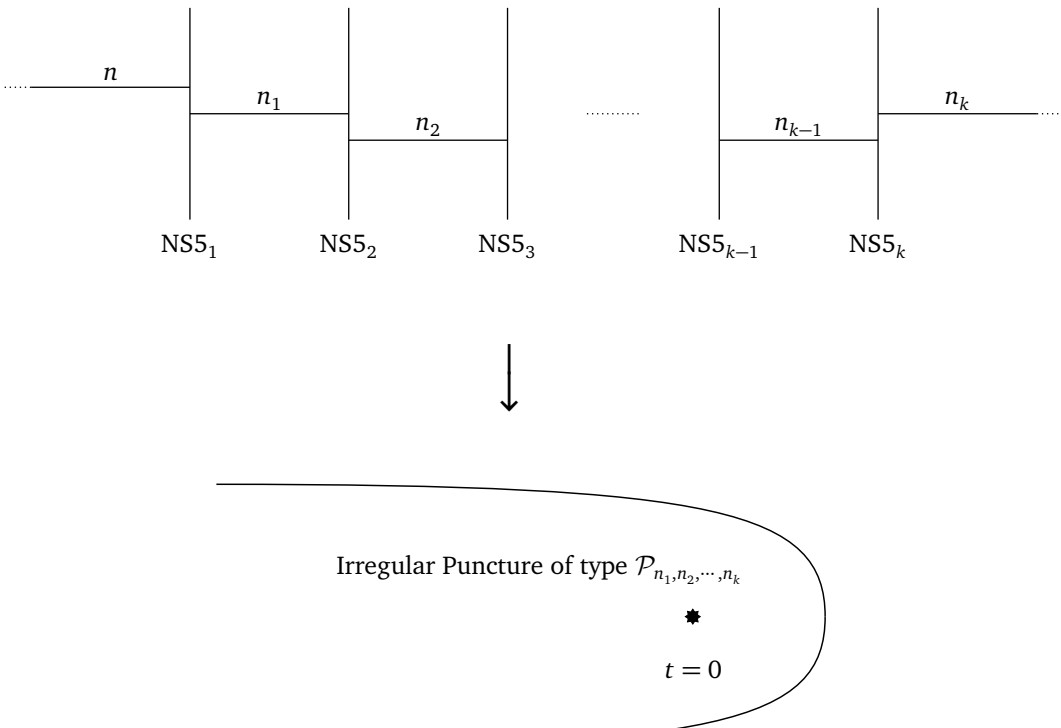

Figure 12: Top: $k \geq 1$ parallel NS5 branes in Type IIA superstring theory with D4 branes stretched between them. All the branes share a common 4-dimensional space-time, and the preserved supersymmetry is $\mathcal{N} = 2$. Here, $n, n_i$ denote numbers of D4 branes. The stacks of $n$ D4 branes on the left and $n_k$ D4 branes on the right are semi-infinite. Bottom: If $n_1 < n$, then the above brane construction can be associated to an irregular puncture of $A_{n-1}$ $(2, 0)$ theory compactified on a cigar parametrized by a complex coordinate $t$ with the puncture being located at $t = 0$. This puncture is referred to be of the type $\mathcal{P}_{n_1, n_2, \cdots, n_k}$.

puncture and all elements of $\widehat{Z}$ are trivial when inserted along a loop encircling a regular puncture.

More interesting values for $Z_{\mathcal{P}}$ and $\bar{Z}_{\mathcal{P}}$ occur for irregular punctures. For general irregular punctures, this information about $Z_{\mathcal{P}}, \bar{Z}_{\mathcal{P}}$ is not known. However, this information can be deduced using Lagrangian field theory for a special class of irregular punctures that can be constructed using Hanany-Witten type brane constructions in Type IIA superstring theory. Consider a brane configuration of the form shown in figure 12, where the following inequalities are satisfied

$$
\begin{aligned}
n_1 &\leq n - 1\,, \\
n + n_2 &\leq 2n_1\,, \\
n_{i-1} + n_{i+1} &\leq 2n_i\,, \qquad \text{for} \quad 2 \leq i \leq k - 1 \\
n_{k-1} &\geq 2\,.
\end{aligned}
\tag{36}
$$

At $t = 0$, this constructs an irregular puncture for $A_{n-1}$ $(2, 0)$ theory which we call to be of type $\mathcal{P}_{n_1, n_2, \cdots, n_k}$.

Let us first consider a sphere with two punctures of type $\mathcal{P}_0$. This corresponds to the brane configuration shown in figure 13. The resulting 4d $\mathcal{N} = 2$ theory can be read from the brane configuration to be pure SYM theory with gauge algebra $\mathfrak{su}(n)$. This theory has

$$
\mathcal{L} \simeq \mathbb{Z}_n^W \times \mathbb{Z}_n^H\,,
\tag{37}
$$

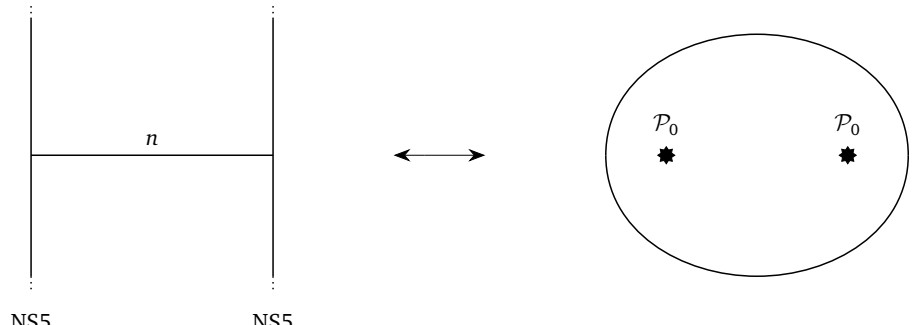

Figure 13: The Type IIA brane construction associated to 6d $A_{n-1}$ $(2,0)$ theory compactified on a sphere with 2 irregular punctures of type $\mathcal{P}_0$.

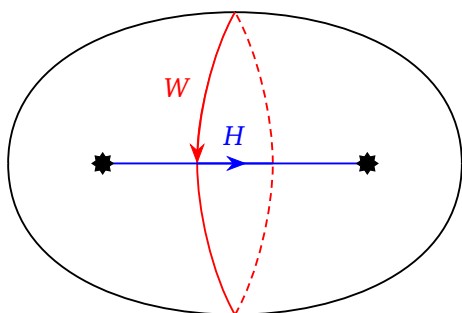

Figure 14: The generators $W$ and $H$ of 1-cycles on a 2-punctured sphere. The intersection number between them is $\langle W, H \rangle = 1$.

with the sub-factor $\mathbb{Z}_n^W$ arising from Wilson lines and the sub-factor $\mathbb{Z}_n^H$ arising from 't Hooft lines. The pairing on $\mathcal{L}$ is

$$\langle W_g, H_g \rangle = \frac{1}{n}, \tag{38}$$

where $W_g$ is a generator of $\mathbb{Z}_n^W$ and $H_g$ is a generator of $\mathbb{Z}_n^H$. On the other hand, as can be seen from figure 14, we also have

$$H_1(\mathcal{C}_g, \widehat{Z}, *) \simeq \mathbb{Z}_n^W \times \mathbb{Z}_n^H, \tag{39}$$

where $\mathbb{Z}_n^W$ sub-factor is generated by wrapping $f \in \widehat{Z} \simeq \mathbb{Z}_n$ along the cycle denoted $W$ in the figure 14, and $\mathbb{Z}_n^H$ sub-factor is generated by wrapping $f \in \widehat{Z} \simeq \mathbb{Z}_n$ along the cycle denoted $H$. We can read the pairing between the generators of $\mathbb{Z}_n^W$ and $\mathbb{Z}_n^H$ to be

$$\langle W \otimes f, H \otimes f \rangle = \frac{1}{n}. \tag{40}$$

Matching with the gauge theory results (37), (38) we find that

$$\mathcal{L} = \mathcal{S} = H_1(\mathcal{C}_g, \widehat{Z}, *), \tag{41}$$

which implies that for a puncture of type $\mathcal{P} = \mathcal{P}_0$ the associated data is

$$\begin{aligned} Z_{\mathcal{P}} &= \widehat{Z}, \\ \bar{Z}_{\mathcal{P}} &= 0, \end{aligned} \tag{42}$$

which also trivially satisfies (34). In other words, every element of $\widehat{Z}$ can end on a type $\mathcal{P}_0$ irregular puncture and no element of $\widehat{Z}$ is trivial, when inserted along a loop encircling a type $\mathcal{P}_0$ irregular puncture.

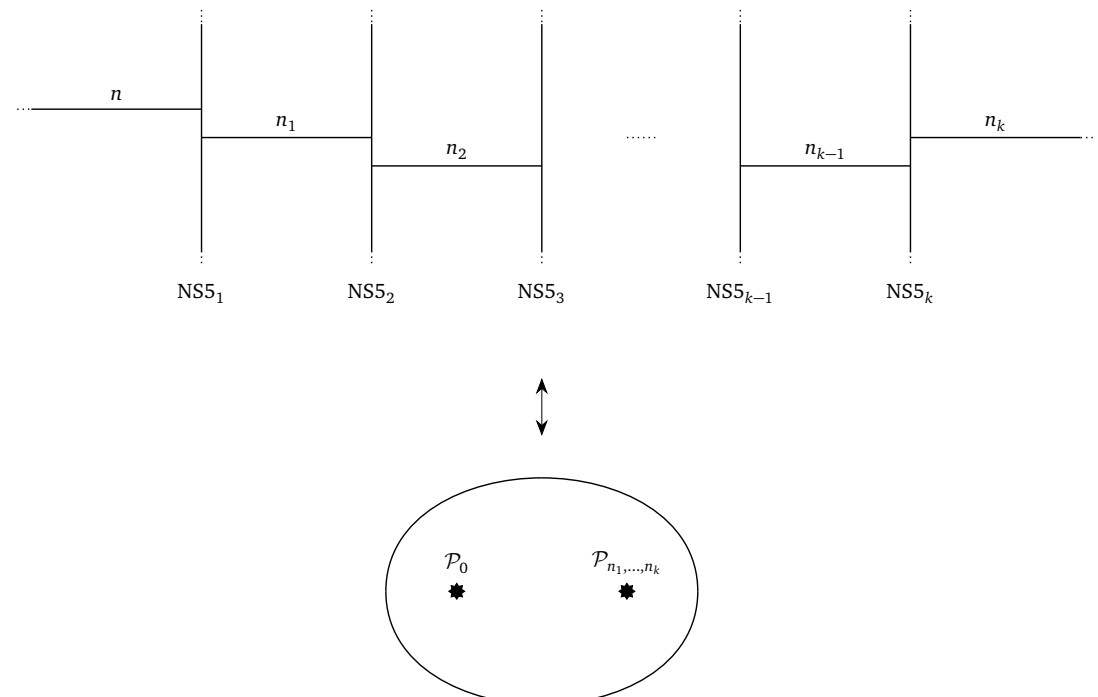

Figure 15: The Type IIA brane construction associated to 6d $A_{n-1}$ $(2,0)$ theory compactified on a sphere with 2 irregular punctures, one of type $\mathcal{P}_0$, and the other of type $\mathcal{P}_{n_1,n_2,\cdots,n_k}$.

Now let us consider a sphere with a puncture of type $\mathcal{P}_0$ and a puncture of general type $\mathcal{P}_{n_1,n_2,\cdots,n_k}$. This corresponds to the brane configuration shown in figure 15. The resulting 4d $\mathcal{N}=2$ theory is the following quiver

$$\mathfrak{su}(n) \relbar\joinrel\relbar \mathfrak{su}(n_1) \relbar\joinrel\relbar \mathfrak{su}(n_2) - \cdots - \mathfrak{su}(n_{k-1}) \relbar\joinrel\relbar n_k \mathsf{F} \ , \tag{43}$$

where there is a bifundamental hyper for any two adjacent gauge algebras, plus $n_k$ fundamental hypers for $\mathfrak{su}(n_{k-1})$. For $n_k > 0$, all electric charges for all $\mathfrak{su}(n_i)$ are screened by the fundamentals and bifundamentals. This implies that there are no magnetic charges that are simultaneously unscreened and mutually local with all the electric charges. Thus, using the Lagrangian description we find that

$$\mathcal{L} = 0 \tag{44}$$

for $n_k > 0$. This implies that any element of $\widehat{Z}$ wrapped along the 1-cycle $W$ must be trivial in $\mathcal{L}$. We know that this triviality does not arise at the location of $\mathcal{P}_0$ puncture. So it must be the case that for a puncture of type $\mathcal{P} = \mathcal{P}_{n_1,n_2,\cdots,n_k}$ with $n_k > 0$, we have

$$\bar{Z}_{\mathcal{P}} = \widehat{Z} \ . \tag{45}$$

Moreover, for (44) to hold, it should not be possible to insert any element of $\widehat{Z}$ along the 1-cycle $H$. Since there are no restrictions on the elements of $\widehat{Z}$ that can end on a puncture of type $\mathcal{P}_0$, we learn that for a puncture of type $\mathcal{P} = \mathcal{P}_{n_1,n_2,\cdots,n_k}$ with $n_k > 0$, we have

$$Z_{\mathcal{P}} = 0 \ . \tag{46}$$

Notice that this is just as for regular punctures, and the constraint (34) is trivially satisfied.

Now let us consider the $n_k = 0$ case. Before accounting for bifundamental matter the electric charges form $\mathbb{Z}_n \times \prod_{i=1}^{k-1} \mathbb{Z}_{n_i}$ group. Let $W$ be the generator of $\mathbb{Z}_n$ sub-factor and $W_i$ be

the generators for $\mathbb{Z}_{n_i}$ sub-factors. Accounting for the bifundamentals, we obtain the relations $W_i = W$ for all $i$. Thus, we have

$$\gcd(n, n_1, n_2, \cdots, n_{k-1})W = 0, \tag{47}$$

and the contribution of Wilson operators to $\mathcal{L}$ is $\mathbb{Z}_{\gcd(n,n_1,n_2,\cdots,n_{k-1})}$. Similarly, before accounting for the bifundamental matter, the magnetic charges also form $\mathbb{Z}_n \times \prod_{i=1}^{k-1} \mathbb{Z}_{n_i}$ group. Let $H$ be the generator of $\mathbb{Z}_n$ sub-factor and $H_i$ be the generators for $\mathbb{Z}_{n_i}$ sub-factors. The subgroup mutually local with the bifundamentals is spanned by

$$\frac{n}{\gcd(n, n_1, n_2, \cdots, n_{k-1})}H + \sum_{i=1}^{k-1} \frac{n_i}{\gcd(n, n_1, n_2, \cdots, n_{k-1})}H_i. \tag{48}$$

Thus, the contribution of 't Hooft operators to $\mathcal{L}$ is also $\mathbb{Z}_{\gcd(n,n_1,n_2,\cdots,n_{k-1})}$. In total, we have

$$\mathcal{L} \simeq \mathbb{Z}^W_{\gcd(n,n_1,n_2,\cdots,n_{k-1})} \times \mathbb{Z}^H_{\gcd(n,n_1,n_2,\cdots,n_{k-1})}. \tag{49}$$

From this, we read that for an irregular puncture of type $\mathcal{P} = \mathcal{P}_{n_1,n_2,\cdots,n_{k-1},0}$, we have

$$\begin{aligned}
Z_{\mathcal{P}} &= \left[ \frac{n}{\gcd(n, n_1, n_2, \cdots, n_{k-1})}f \right] \subseteq \widehat{Z}, \\
\bar{Z}_{\mathcal{P}} &= \left[ \gcd(n, n_1, n_2, \cdots, n_{k-1})f \right] \subseteq \widehat{Z},
\end{aligned} \tag{50}$$

where $[\alpha]$ denotes the subgroup of $\widehat{Z}$ generated by $\alpha \in \widehat{Z}$. The pairing between the generators of $Z_{\mathcal{P}}$ and $\bar{Z}_{\mathcal{P}}$ is

$$\left\langle \frac{n}{\gcd(n, n_1, n_2, \cdots, n_{k-1})}f, \; \gcd(n, n_1, n_2, \cdots, n_{k-1})f \right\rangle = 0, \tag{51}$$

as required by the constraint (34).

## 3.3 Rotation and $\mathcal{N} = 1$ Higgs Bundles

We next discuss constructions of $\mathcal{N} = 1$ theories by compactification of 6d $(2,0)$ theories on a Riemann surface with a partial topological twist. This includes both general $\mathcal{N} = 1$ theories, and the $\mathcal{N} = 1$ theories of interest that can be obtained by deforming $\mathcal{N} = 2$ theories of Class S. Along the way, we would encounter key notions of generalized Hitchin system and $\mathcal{N} = 1$ curve which characterize vacua of these $\mathcal{N} = 1$ theories.

### 3.3.1 Topological Twists for 4d $\mathcal{N} = 2$ and $\mathcal{N} = 1$

Start with 6d $(2,0)$ theory of type $\mathfrak{g} = A, D, E$ compactified on a Riemann surface $\mathcal{C}_{g,n}$ of genus $g$, with $n$ punctures. We want to perform a partial topological twist along $\mathcal{C}$ which preserves at least 4d $\mathcal{N} = 1$ supersymmetry. The global symmetries of the 6d theory are the local Lorentz and R-symmetry $\mathfrak{so}(6)_L \oplus \mathfrak{so}(5)_R$ which are broken to $\mathfrak{so}(4)_L \oplus \mathfrak{u}(1)_L \oplus \mathfrak{so}(5)_R$ by the background $M_4 \times \mathcal{C}$. To preserve $\mathcal{N} = 2$ supersymmetry, we would decompose the R-symmetry as $\mathfrak{so}(5)_R \to \mathfrak{su}(2)_R \oplus \mathfrak{u}(1)_1$ and twist $\mathfrak{u}(1)_L$ by $\mathfrak{u}(1)_1$. For a more general twist that in general preserves $\mathcal{N} = 1$ supersymmetry, we further reduce $\mathfrak{su}(2)_R$ to $\mathfrak{u}(1)_2$ and both $\mathfrak{u}(1)_1$ and $\mathfrak{u}(1)_2$ are used to twist $\mathfrak{u}(1)_L$. The supercharges $Q$ and scalars[12] $\Phi$ decompose as

$$\begin{aligned}
\mathfrak{so}(6)_L \oplus \mathfrak{so}(5)_R &\rightarrow (\mathfrak{so}(4)_L \oplus \mathfrak{u}(1)_L) \oplus (\mathfrak{u}(1)_1 \oplus \mathfrak{u}(1)_2) \\
Q: \quad (\mathbf{4}, \mathbf{4}) &\rightarrow ((\mathbf{2}, \mathbf{1})_{+1} \oplus (\mathbf{1}, \mathbf{2})_{-1}) \otimes (\mathbf{1}_{++} \oplus \mathbf{1}_{+-} \oplus \mathbf{1}_{-+} \oplus \mathbf{1}_{--}) \\
\Phi: \quad (\mathbf{1}, \mathbf{5}) &\rightarrow \mathbf{1}_0 \otimes \left( \mathbf{1}_{2,0} \oplus \mathbf{1}_{-2,0} \oplus \mathbf{1}_{0,2} \oplus \mathbf{1}_{0,0} \oplus \mathbf{1}_{0,-2} \right).
\end{aligned} \tag{52}$$

---

[12]These scalars are not genuine local operators in the 6d $(2,0)$ SCFT, but the Casimirs built out of these scalars are genuine local operators in the 6d theory.

The twists giving $\mathcal{N} = 1$ supersymmetry are parametrized by an integer parameter $\alpha$ which sets the charge $q_{\text{tw}}$ of the preserved diagonal combinination of the three $\mathfrak{u}(1)$ factors

$$q_{\text{tw}} = q_L + (1 - \alpha)q_1 + \alpha q_2 \,. \tag{53}$$

For $\alpha = 0, 1$, we recover the usual $\mathcal{N} = 2$ twist. For other values of $\alpha$, only 4 supercharges are preserved and hence the twist is $\mathcal{N} = 1$. It should be noted that one can obtain $\mathcal{N} = 1$ supersymmetry from the $\mathcal{N} = 2$ twist, as we discuss below.

The scalars $\Phi$ of the 6d theory transform with charges $q_{\text{tw}} = \pm 2(1-\alpha), \pm 2\alpha, 0$. The twisted scalars carrying charges $\pm 2\alpha$ and $\pm 2(1-\alpha)$ are sections of two line bundles $\mathcal{L}_1$ and $\mathcal{L}_2$ with

$$\deg(\mathcal{L}_1) + \deg(\mathcal{L}_2) = \deg(K_{\mathcal{C}}), \tag{54}$$

where $K_{\mathcal{C}}$ is the canonical line bundle on $\mathcal{C}$. We denote these by $\phi$ and $\varphi$ respectively.

The above setup preserves 4d $\mathcal{N} = 1$ supersymmetry for $\alpha \neq 0, 1$ since the twist only preserves a maximum of 4 supercharges. But for $\alpha = 0, 1$ one can have either 4d $\mathcal{N} = 1$ or 4d $\mathcal{N} = 2$ supersymmetry. To understand this, without loss of generality, consider the case $\alpha = 0$, for which $\phi$ must be singular, and hence non-zero, but $\varphi$ (which is a function on $\mathcal{C}$) can be zero or non-zero. For zero $\varphi$, we inherit $\mathfrak{su}(2)_R \subset \mathfrak{so}(5)_R$ R-symmetry in 4d and thus 4d $\mathcal{N} = 2$ supersymmetry. On the other hand, if $\varphi$ is non-zero, then the $\mathfrak{su}(2)_R$ R-symmetry is broken by the non-zero profile of $\varphi$, and we only obtain 4d $\mathcal{N} = 1$ supersymmetry.

### 3.3.2 Generalized Hitchin System and Rotation of Codim-2 Defects

The profiles of $\phi, \varphi$ over $\mathcal{C}$ satisfy a set of BPS equations that were determined in [7] and yield what is known as the $\mathcal{N} = 1$ Hitchin system

$$\begin{aligned}
\bar{D}\phi = \bar{D}\varphi &= 0, \\
[\phi, \varphi] &= 0, \\
F + [\varphi, \varphi^*] + [\phi, \phi^*] &= 0 \,.
\end{aligned} \tag{55}$$

Here the star denotes conjugation and $F$ abbreviates the field strength of a connection on $\mathcal{C}$. In this paper, we always restrict ourselves to the special case where the Higgs fields $\phi$ and $\varphi$ are diagonalizable (at each point in $\mathcal{C}$), and therefore the third BPS equation in (55) imposes $F = 0$.

The punctures on $\mathcal{C}$ are characterized by singularities of the two Higgs fields $\phi, \varphi$. From the point of view of the 6d $(2, 0)$ theory, punctures with different singularities are identified as different codimension-2 defects inserted along the locations of punctures. Such defects preserve 4d $\mathcal{N} = 1$ supersymmetry in general. In the standard $\mathcal{N} = 2$ Class S case, we have $\alpha = 0$ and $\varphi = 0$. The codimension-2 defects are then characterized by singularities of $\phi$ and preserve a mutual $\mathcal{N} = 2$ supersymmetry. Now, one can "rotate" such an $\mathcal{N} = 2$ codimension-2 defect to an $\mathcal{N} = 1$ codimension-2 defect[13] by turning on a singular $\varphi$ at the location of the corresponding puncture. As we will discuss in section 3.4, the second equation in (55) allows us to write the behavior of $\varphi$ near the defect, placed at $t = 0$, as

$$\varphi \sim \sum_{k=0}^{m} \frac{r_k}{t^{b_k}} \phi_\zeta^k \,, \tag{56}$$

---

[13]It is also possible that the resulting defect actually preserves an $\mathcal{N} = 2$ supersymmetry but it would be a different $\mathcal{N} = 2$ supersymmetry than the $\mathcal{N} = 2$ supersymmetry preserved by the unrotated defect. That is, in such a situation, inserting both the rotated and unrotated defects would only preserve $\mathcal{N} = 1$ supersymmetry.

where $\phi_\zeta$ is the contraction of $\phi$ with a holomorphic vector field $\zeta$ that is non-singular at $t = 0$, and $b_k \in \mathbb{Z}$. Let the most singular piece of $\phi_\zeta$ be of order $t^{-q}$ for some $q > 0$. Let

$$S = \{k \in \{0, 1, 2, \cdots, m\} : b_k + kq > 0\}. \tag{57}$$

The terms in the sum (56) that correspond to $k \in S$ capture the singular pieces of $\varphi$ near $p$. Consequently, $r_k$ for $k \in S$ are interpreted as deformation parameters rotating the codimension-2 defect. We refer to a puncture associated to a rotated codimension-2 defect as a rotated puncture.

### 3.3.3 Rotation of a 4d $\mathcal{N} = 2$ to a 4d $\mathcal{N} = 1$ Theory

Consider a situation in which all the punctures on $\mathcal{C}$ are rotated punctures. If we replace all the rotated punctures by their unrotated versions, and take a zero area limit of $\mathcal{C}$, then we obtain a (not necessarily conformal) 4d $\mathcal{N} = 2$ Class S theory which is UV complete in 4d. In a similar way, from the original situation having rotated punctures, one would want to obtain a UV complete 4d $\mathcal{N} = 1$ theory by taking the zero area limit of $\mathcal{C}$. For small non-zero area $A$, the compactified 6d system can be described at energy scales $E \ll 1/A$ by 4d $\mathcal{N} = 2$ Class S theory deformed by rotation parameters $r_k \in S$ coming from each puncture, defined at a cutoff scale $1/A$. In general, these parameters may contain relevant, marginal and irrelevant deformation parameters of the 4d $\mathcal{N} = 2$ Class S theory. If all the deformation parameters are relevant or marginal, then one can consistently take a zero area limit, thus lifting the cutoff and obtaining a UV complete 4d $\mathcal{N} = 1$ theory which is defined as relevant and marginal deformation of the initial 4d $\mathcal{N} = 2$ Class S theory. However, on the other hand, if any of the rotation parameters is irrelevant, then one runs into the usual issues of non-renormalizability and it is not clear if the zero area limit can be consistently taken, and if it can be taken then what the resulting 4d $\mathcal{N} = 1$ theory is. Irrespective of these subtleties, we can still study the confinement properties of the 4d $\mathcal{N} = 1$ theory with a cutoff imposed by the area of $\mathcal{C}$, as it is not impacted by the cutoff and the precise details of 4d UV completion (if it exists). We will study examples of both kinds of situations later in this paper.

In either case, different profiles on $\mathcal{C}$ of $\phi, \varphi$ satisfying the generalized Hitchin equations (55) (for a fixed structure of singularities) characterize different 4d vacua. For a 4d $\mathcal{N} = 2$ Class S theory, we have $\varphi = 0$ and the various profiles of $\phi$ form a moduli space which can be identified with the Coulomb branch (CB) of vacua of the 4d $\mathcal{N} = 2$ theory. After an $\mathcal{N} = 1$ rotation which switches on a non-zero $\varphi$, only a subset of profiles of $\phi$ satisfy the second condition in (55). This means that the CB vacua corresponding to other profiles of $\phi$ are lifted by the $\mathcal{N} = 1$ deformation, and the CB vacua corresponding to the profiles of $\phi$ that satisfy (55) remain as vacua of the resulting 4d $\mathcal{N} = 1$ theory (which may have a UV cutoff as discussed above). It should be noted that there can also be other vacua arising from the Higgs branch of the unrotated 4d $\mathcal{N} = 2$ theory, which we do not study in this paper.

## 3.4 The $\mathcal{N} = 1$ Curve

### 3.4.1 Spectral Curve

To a generic-enough profile of diagonalizable $\phi, \varphi$ satisfying (55), one can associate an $\mathcal{N} = 1$ curve which lives in $\mathcal{L}_1 \oplus \mathcal{L}_2$ and is an $N$-sheeted cover of $\mathcal{C}$. We start by picking two generic meromorphic sections $\zeta$ and $\eta$ of $\mathcal{L}_1^{-1}$ and $\mathcal{L}_2^{-1}$ respectively. The contractions

$$\phi_\zeta \equiv \phi(\zeta), \qquad \varphi_\eta \equiv \varphi(\eta), \tag{58}$$

are meromorphic functions on $\mathcal{C}$. By the second equation in (55), the Higgs fields $\varphi_\eta$ and $\phi_\zeta$ commute. If at generic points on the curve $\mathcal{C}$ the eigenvalue spectrum of $\phi_\zeta$ and $\varphi_\eta$ are

distinct, then each Higgs field can be expressed as a polynomial of the other Higgs field

$$\varphi_\eta = \mathcal{R}(\phi_\zeta) = \sum_k r_k(t)\phi_\zeta^k,$$
$$\phi_\zeta = \mathcal{S}(\varphi_\eta) = \sum_k s_k(t)\varphi_\eta^k. \tag{59}$$

By the first BPS equation in (55) the coefficient functions $r_k, s_k$ are meromorphic functions on the compact curve $\mathcal{C}$. We diagonalize (59) and find

$$w = \mathcal{R}(v) = \sum_k r_k(t)v^k,$$
$$v = \mathcal{S}(w) = \sum_k s_k(t)w^k, \tag{60}$$

solved by pairs of eigenvalues $(w, v)$ of $(\varphi_\eta, \phi_\zeta)$.

The above pairing of eigenvalues allows us to combine the spectral covers associated to $\varphi_\eta, \phi_\zeta$ into a single $N$-sheeted cover $\Sigma \subset \mathcal{L}_1 \oplus \mathcal{L}_2$ of $\mathcal{C}$ known as the $\mathcal{N} = 1$ curve. In more detail, the two characteristic equations[14]

$$\det(v - \phi_\zeta) = 0,$$
$$\det(w - \varphi_\eta) = 0, \tag{61}$$

for $\varphi_\eta$ and $\phi_\zeta$ define two spectral covers

$$\mathcal{P}(v) = \sum_l p_l(t)v^l = 0, \qquad \mathcal{Q}(w) = \sum_l q_l(t)w^l = 0. \tag{62}$$

The coefficients $p_l, q_l$ are again meromorphic functions. Each spectral curve (62) is an $N$-sheeted covering of $\mathcal{C}$ with the number of sheets $N$ depending on the type $\mathfrak{g} = A, D, E$ of the $(2,0)$ theory. Consider the monodromy of the $N$ sheets of $\mathcal{P}(v)$ around a branch point $p \in \mathcal{C}$ or around a cycle $C \in H_1(\mathcal{C})$, which is given by some element of the permutation group $S_N$ permuting the $N$ sheets. Now, the monodromy of $\mathcal{Q}(w)$ around $p$ or $C$ must be the same as the sheets (which are described by the eigenvalues) of $\mathcal{P}(v)$ and $\mathcal{Q}(w)$ are paired. Thus, the monodromies of $\mathcal{P}(v)$ must match the monodromies of $\mathcal{Q}(w)$. Furthermore, the pairs $(v, w)$ define a *combined $N$-fold cover* $\widetilde{\Sigma} \subset \mathcal{O}_{\mathcal{C}} \oplus \mathcal{O}_{\mathcal{C}}$ of $\mathcal{C}$ whose monodromies match the monodromies of $\mathcal{P}(v)$ and $\mathcal{Q}(w)$. The $\mathcal{N} = 1$ curve $\Sigma \subset \mathcal{L}_1 \oplus \mathcal{L}_2$ is then identified as the $N$-sheeted cover of $\mathcal{C}$ spanned by pairs $(v\zeta^{-1}, w\eta^{-1})$. The $\mathcal{N} = 1$ curve $\Sigma$ can also be thought of as being cut out by

$$\det(\lambda - \phi) = 0, \qquad \det(\sigma - \varphi) = 0, \qquad \det(\lambda\sigma - \phi\varphi) = 0, \tag{63}$$

where $\lambda \in \mathcal{L}_1$ and $\sigma \in \mathcal{L}_2$.

In conclusion, for each vacuum $r$ of a 4d $\mathcal{N} = 1$ theory that can be characterized by profiles $\phi_r, \varphi_r$ of $\mathcal{N} = 1$ Higgs fields satisfying at least one of the equations[15] (60), we can associate a curve $\Sigma_r \subset \mathcal{L}_1 \oplus \mathcal{L}_2$ which is an $N$-fold cover of $\mathcal{C}$.

### 3.4.2 Algorithm for Determining the $\mathcal{N} = 1$ Curve

In this paper, we focus on the study of the $\mathcal{N} = 1$ curves with $\alpha = 0$ with $\varphi \neq 0$, and in particular those cases which can be understood as "rotations" of standard $\mathcal{N} = 2$ Class S setups. For these cases, we provide an algorithm to determine the $\mathcal{N} = 1$ curves for those vacua of the $\mathcal{N} = 1$ theory (obtained after rotation) that arise from the $\mathcal{N} = 2$ Coulomb branch:

---

[14]To write down the characteristic equations, we represent $\phi_\zeta, \varphi_\eta$ as matrices acting in the fundamental representation, vector representation, **27**, **56** and **248** for the Lie algebras $A_n, D_n, E_6, E_7$ and $E_8$ respectively.

[15]Later, we will see an example where only one of the two equations in (60) is satisfied, but not both.

1. Unrotated Theory: Choose an $\mathcal{N} = 2$ Class S theory (which need not be conformal) by specifying the singularities of $\phi$ at the locations of punctures. We can determine the profile of $\phi$ away from the punctures by using holomorphicity. Different profiles of $\phi$ parameterize Coulomb branch of $\mathcal{N} = 2$ vacua, and the Seiberg-Witten (SW) curve associated to each vacuum is obtained by inserting the corresponding profile of $\phi$ into the characteristic equation $\det(\lambda - \phi) = 0$. The SW curve is an $N$-sheeted cover $\mathcal{P}(\lambda) = 0$ of the Gaiotto curve $\mathcal{C}_{g,n}$.

2. Rotation: Fix the singularities of $\varphi$ at each puncture $p$. This can be done by specifying $r_k^{(p)}$ and $b_k^{(p)}$ arising in (56) for $k \in S$. This determines the deformation parameters used to deform the Class S $\mathcal{N} = 2$ theory chosen above. Write down the generic meromorphic profile of $\varphi$ having singularities determined by the above imposed boundary conditions. This generic profile determines another $N$-sheeted cover $\mathcal{Q}(w) = 0$ of $\mathcal{C}_{g,n}$ via the characteristic equation $\det(w - \varphi) = 0$.

3. Topological Factorization: Determine all possible topological degenerations by moving and colliding the branch points of both the $N$-sheeted covers $\mathcal{P}(\lambda)$ and $\mathcal{Q}(w)$ of $\mathcal{C}$, such that the monodromies for $\mathcal{P}(\lambda), \mathcal{Q}(w)$ match after the degeneration. Each such topological degeneration determines a potential factorization of the discriminants of $\mathcal{P}(\lambda), \mathcal{Q}(w)$ as $N$-sheeted covers of $\mathcal{C}$.

4. "Holomorphic" Factorization: After determining all possible topological degenerations for which monodromies of $\mathcal{P}(\lambda), \mathcal{Q}(w)$ match, one needs to check that the corresponding potential factorizations of the discriminants of $\mathcal{P}(\lambda), \mathcal{Q}(w)$ are realizable without changing the singularities of $\phi, \varphi$ which defined the parent $\mathcal{N} = 2$ theory and its $\mathcal{N} = 1$ rotation. If this is possible, then one also needs to check that all the monodromies are as determined by the topological degeneration. If the monodromies also match, then the CB moduli for which the factorization is possible determine a vacuum of the descendant $\mathcal{N} = 1$ theory. It is possible that one finds multiple possible choices of CB parameters for a fixed topological degeneration leading to multiple $\mathcal{N} = 1$ vacua whose corresponding $\mathcal{N} = 1$ curves have the same set of branch points and monodromies. However, the branch lines connecting the branch points for these different vacua might be topologically distinct from each other. This difference reflects in the difference of the images of the map (65) for these different vacua, that we discuss in the next subsection.

### 3.5 Confinement from the $\mathcal{N} = 1$ Curve

Consider a relative 4d $\mathcal{N} = 1$ theory that has been obtained as a rotation of a relative 4d $\mathcal{N} = 2$ Class S theory of above type. We propose that the defect group $\mathcal{L}$ of line operators remains invariant under the rotation. Thus, $\mathcal{L}$ for the resulting 4d $\mathcal{N} = 1$ theory is the same as for the 4d $\mathcal{N} = 2$ theory determined in section 3.2.

Consider a vacuum $r$ of the 4d $\mathcal{N} = 1$ theory that descends from a Coulomb branch vacuum of the parent 4d $\mathcal{N} = 2$ theory. As we discussed in the previous subsection, if certain conditions are met, we can associate to this vacuum $r$ an $\mathcal{N} = 1$ curve $\Sigma_r \subset T^*\mathcal{C} \times \mathbb{C}$ which is an $N$-fold cover of $\mathcal{C}$ characterized by a projection map

$$\pi^r : \qquad \Sigma_r \to \mathcal{C}. \tag{64}$$

We can use this map to define a pushforward map

$$\pi^r_* : \qquad H_1(\Sigma_r, \widehat{Z}, *) \to H_1(\mathcal{C}, \widehat{Z}, *), \tag{65}$$

from 1-cycles on $\Sigma_r$ (that are allowed to end on punctures) to 1-cycles on $\mathcal{C}$ (that are allowed to end on punctures). We further argue (see below) that the line operators $\mathcal{I}_r \subseteq \mathcal{L}$ that exhibit perimeter law can be identified with

$$\mathcal{I}_r = \pi_* \left( \mathcal{S} \cap \pi_*^r \left( H_1(\Sigma_r, \widehat{Z}, *) \right) \right), \tag{66}$$

where $\pi$ is the projection map from $\mathcal{S}$ to $\mathcal{L}$ defined in (32), and $\pi_*$ is the associated pushforward.

As discussed in section 3.1, an absolute 4d $\mathcal{N} = 1$ theory specifies a polarization $\Lambda \subset \mathcal{L}$, and then the preserved 1-form symmetry group $\mathcal{O}_r$ for this absolute theory in the vacuum $r$ is determined to be

$$\mathcal{O}_r = \widehat{\left( \frac{\Lambda}{\Lambda_r} \right)}, \tag{67}$$

where $\Lambda_r := \mathcal{I}_r \cap \Lambda$.

Our argument for (66) is a generalization of the argument appearing in [10] where confinement for $\mathcal{N} = 1$ SYM was studied in this setup, which can be understood as a compactification of M-theory as follows. We can realize a CB vacuum of the 4d $\mathcal{N} = 2$ Class S theory before rotation as M-theory compactified on $T^*\mathcal{C}$ with M5 branes wrapping a curve $\Sigma \subset T^*\mathcal{C}$, which is identified as the SW curve for that vacuum. $\Sigma$ is an $n$-fold cover of $\mathcal{C}$ under the projection map $T^*\mathcal{C} \to \mathcal{C}$ and is cut out by the characteristic equation $\det(\lambda - \phi) = 0$, where $\phi$ is the $\mathcal{N} = 2$ Higgs field discussed above and $\lambda$ is a coordinate along the fiber of $T^*\mathcal{C}$. Once we rotate the $\mathcal{N} = 2$ theory to $\mathcal{N} = 1$, then a vacuum $r$ of the resulting $\mathcal{N} = 1$ theory is realized by M5 branes wrapping a curve $\Sigma_r \subset T^*\mathcal{C} \times \mathbb{C}$, which is identified as the $\mathcal{N} = 1$ curve associated to that vacuum. The projection of $\Sigma_r$ to $T^*\mathcal{C}$ is cut out by the characteristic equation $\det(\lambda - \phi) = 0$ and the projection of $\Sigma_r$ to $\mathbb{C}$ is cut out by the characteristic equation $\det(w - \varphi) = 0$, where $\varphi$ is the other Higgs field discussed above and $w$ is a coordinate along $\mathbb{C}$.

To study confinement, we study the charges of confining strings in this setup. The confining strings are realized by compactifying M2 branes along 1-cycles of $Y = T^*\mathcal{C} \times \mathbb{C}$. A crucial addition to the argument of [10] is that we need to also include 1-cycles in $Y$ whose projections onto $\mathcal{C}$ contain 1-cycles included in $H_1(\mathcal{C}, \mathbb{Z}, *)$ (see section 3.2) which end at punctures of $\mathcal{C}$. After including such 1-cycles we obtain a homology group $H_1(Y, \mathbb{Z}, *)$, which is isomorphic to $H_1(\mathcal{C}, \mathbb{Z}, *)$ under the projection map. Thus, as a first step, the possible charges of confining strings are characterized by $H_1(Y, \mathbb{Z}, *) \simeq H_1(\mathcal{C}, \mathbb{Z}, *)$. Now, we need to account for the fact that M2 branes can end on M5 branes, which implies that M2 branes characterized by different elements of $H_1(Y, \mathbb{Z}, *)$ can be related by topological moves, which split and join the M2 branes along the M5 brane locus $\Sigma_r$. All such identifications are captured by the fact that a confining string arising from an element $i_* H_1(\Sigma_r, \mathbb{Z}, *) \subseteq H_1(Y, \mathbb{Z}, *)$ is topologically equivalent to a trivial string and hence must have trivial charge. Here $i : \Sigma_r \hookrightarrow Y$ denotes the inclusion map and $i_*$ denotes the associated pushforward map which embeds the cycles of $\Sigma_r$ into $Y$. See figures 16 and 17.

We can use this fact to constrain the possible charges of confining strings to lie in

$$\frac{H_1(Y, \mathbb{Z}_n, *)}{i_* H_1(\Sigma_r, \mathbb{Z}_n, *)}, \tag{68}$$

since any cycle of the form $nC$ with $C \in H_1(Y, \mathbb{Z}, *) \simeq H_1(\mathcal{C}, \mathbb{Z}, *)$ lies in $i_* H_1(\Sigma_r, \mathbb{Z}, *)$ because $\Sigma$ is an $n$-fold cover of $\mathcal{C}$. In other words, at this step, the possible charges of confining strings are characterized by modding out $H_1(\mathcal{C}, \widehat{Z}, *)$ by the image of $H_1(\Sigma_r, \widehat{Z}, *)$ under the projection map (64)

$$\frac{H_1(\mathcal{C}, \widehat{Z}, *)}{\pi_*^r H_1(\Sigma_r, \widehat{Z}, *)}, \tag{69}$$

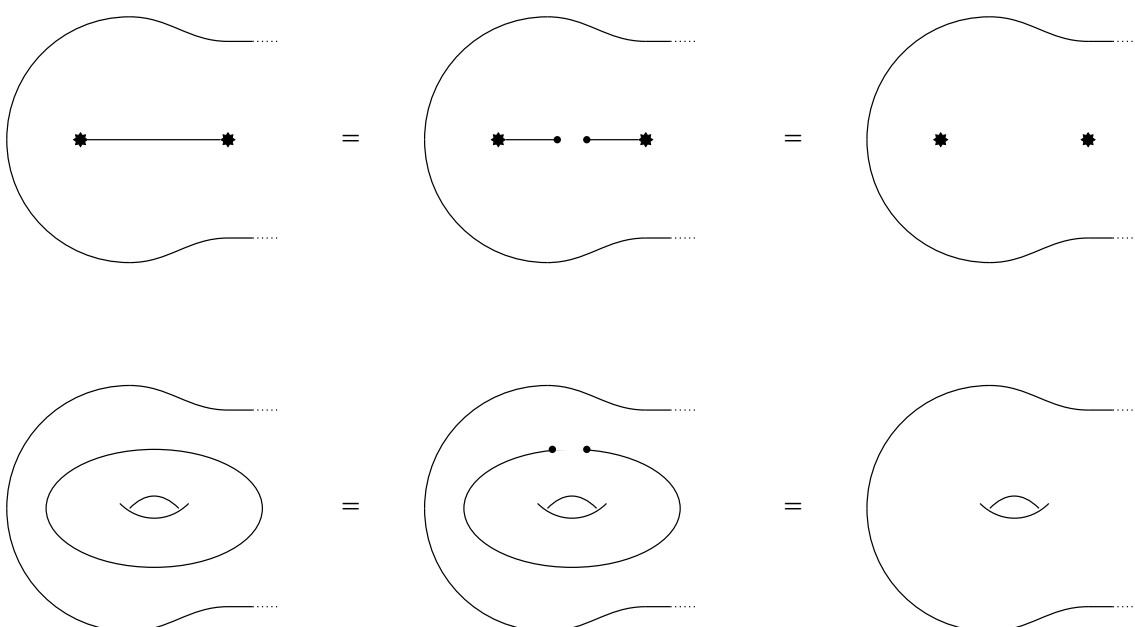

Figure 16: The figure shows various configurations of M2 branes wrapping 1-chains inside M5 branes. An M5 branes is depicted as a surface, which shows a local region of $\Sigma_r$. In the top-left configuration, we consider an M2 brane stretched between two punctures on $\Sigma_r$. The subsequent configurations depict that we can get rid of this M2 brane by splitting it into two by creating its end-points (shown with a black dot) on the M5 brane. Similarly, in the bottom-left configuration, we consider an M2 brane wrapping a compact 1-cycle on $\Sigma_r$. As shown in the subsequent configurations, we can get rid of this M2 brane again by creating its end-points on the M5 brane.

where $\widehat{Z} \simeq \mathbb{Z}_n$.

We can further reduce the set of possible charges by recalling that different punctures allow different subgroups of $\widehat{Z}$ to end on them. As we discussed in section 3.2, this means that the allowed charges of line operators take values in a subgroup $\mathcal{S}$ of $H_1(\mathcal{C}, \widehat{Z}, *)$. Consequently, at this step, the possible charges of confining strings are characterized by

$$\frac{\mathcal{S}}{\mathcal{S} \cap \pi_*^r \left( H_1(\Sigma_r, \widehat{Z}, *) \right)}. \tag{70}$$

Finally, we take into account the fact that not all charges in $\mathcal{S}$ are independent charges of line operators. As discussed in section 3.2, we need to mod out a subgroup of $\mathcal{S}$ to obtain the true group of charges $\mathcal{L}$ of line operators, resulting in a projection map $\pi : \mathcal{S} \to \mathcal{L}$. Thus, finally the possible charges of confining strings are characterized by

$$\frac{\mathcal{L}}{\mathcal{I}_r}, \tag{71}$$

with $\mathcal{I}_r$ given in (66). Once we choose an absolute theory with charges of line operators specified by subgroup $\Lambda \subset \mathcal{L}$ fixing the 1-form symmetry group $\mathcal{O} = \widehat{\Lambda}$, the confining strings are chosen to have charges

$$\frac{\Lambda}{\mathcal{I}_r \cap \Lambda}, \tag{72}$$

and the preserved 1-form symmetry group is $\mathcal{O}_r \subseteq \mathcal{O}$ where $\mathcal{O}_r$ is the Pontryagin dual of the above group formed by charges of confining strings.

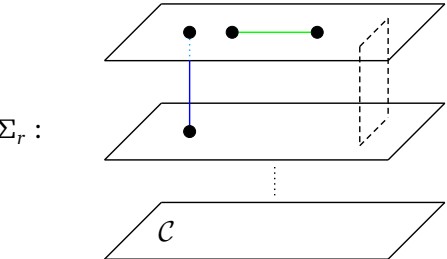

Figure 17: The $\mathcal{N} = 1$ curve $\Sigma_r \subset Y$ as a covering of the Gaiotto curve $\mathcal{C}$. The dashed sheet denotes a branch cut connecting two sheets. The green and blue line segments denote 1-cycles wrapped by M2 branes and give rise to potential confining strings. The green cycle is contained in the M5 brane locus and is of the type depicted in figure 16. The confining strings associated to it are trivial and uncharged under the 1-form symmetry. The blue cycle stretches between the sheets, is contained in $Y$, and is an element of the relative homology group $H_1(Y, \Sigma_r)$. These are *charged* under the 1-form symmetry.

# 4 Confinement in 4d $\mathcal{N} = 1$ SYM

In this section we consider the well studied case of 4d $\mathcal{N} = 1$ $\mathfrak{su}(n)$ SYM and determine the 1-form symmetry groups and their spontaneous breakings in various vacua for various global forms of the gauge group and discrete theta parameters. We do so using the machinery developed in section 3, and in particular using our main proposal in section 3.5 about reading off confinement from the $\mathcal{N} = 1$ curve. This problem was previously studied field-theoretically in [2] without using the modern language of 1-form symmetry and its spontaneous breaking. We enhance their description at various points by providing explicit results for the UV 1-form symmetry groups for various polarizations and the preserved 1-form symmetry groups in various vacua for various polarizations. Our main purpose is to use this example as a test ground to verify and demonstrate our more general prescription.

We begin with a Class S construction of 4d $\mathcal{N} = 2$ $\mathfrak{su}(n)$ SYM discussed in section 3.2, where the defect group $\mathcal{L}$ of the theory was also discussed. We then perform a rotation $\mu$ of the 4d $\mathcal{N} = 2$ theory such that 4d $\mathcal{N} = 1$ SYM is obtained as the $\mu \to \infty$ limit. Following the algorithm of section 3.4.2, we determine various $\mathcal{N} = 1$ vacua and their corresponding $\mathcal{N} = 1$ curves for finite $\mu$. Then we use the topological structure of $\mathcal{N} = 1$ curves to determine the group $\mathcal{I}_r$ of line operators showing perimeter law in each vacuum $r$. This allows us to present our main result, i.e. the computation of the preserved 1-form symmetry group $\mathcal{O}_r$ in the vacuum $r$ (for various choices of polarization $\Lambda$). These results remain unchanged as we take the limit $\mu \to \infty$ and recover pure 4d $\mathcal{N} = 1$ $\mathfrak{su}(n)$ SYM theory.

## 4.1 $\mathcal{N} = 2$ Curve and Line Operators

A Class S construction of 4d $\mathcal{N} = 2$ $\mathfrak{su}(n)$ SYM was discussed in section 3.2. It involves compactifying 6d $A_{n-1}$ $(2,0)$ theory on a compactification manifold $\mathcal{C}$ which is a sphere with two punctures, both of type $\mathcal{P}_0$ (see figure 13). As discussed there, the defect group $\mathcal{L}$ is identified with the group of 1-cycles $H_1(\mathcal{C}, \widehat{Z}, *)$

$$\mathcal{L} = H_1(\mathcal{C}, \widehat{Z}, *) \cong \mathbb{Z}_n^W \times \mathbb{Z}_n^H, \tag{73}$$

with coefficients in $\widehat{Z} \cong \mathbb{Z}_n$. The factors labelled $W, H$ are associated with the Wilson and 't Hooft lines of the field theory and geometrically with the 1-cycles encircling a puncture and running between the punctures respectively, as depicted in figure 14 in section 3.2.

The SW curve is [48]

$$\mathcal{P}(v) = P_n(v) - \Lambda^n\left(t + \frac{1}{t}\right) = 0, \tag{74}$$

where $(v, t) \in \mathbb{C} \times \mathbb{C}^*$ with $P_n(v) = v^n + u_2 v^{n-2} + \cdots + u_n$ where $u_k$ are combinations of CB parameters. The SW differential is $\lambda = v\, dt/t$. The dynamically generated scale is denoted $\Lambda_{\mathcal{N}=2} \equiv \Lambda$. The asymptotics of the Higgs field approaching the punctures at $t = 0, \infty$ can be derived from (74) and are taken to define the Higgs field $\phi$ profile characterizing punctures of type $\mathcal{P}_0$. At the two $\mathcal{P}_0$ punctures $t = 0, \infty$ the Higgs field $\phi = \phi_\zeta(dt/t)$ therefore diverges as

$$
\begin{aligned}
t \to 0: && \phi_\zeta &\sim \frac{\Lambda}{t^{1/n}}\mathrm{diag}\left(1, \omega, \omega^2, \cdots, \omega^{n-1}\right) + \cdots, \\
t \to \infty: && \phi_\zeta &\sim \Lambda t^{1/n}\mathrm{diag}\left(1, \omega, \omega^2, \cdots, \omega^{n-1}\right) + \cdots,
\end{aligned}
\tag{75}
$$

with the $n$-th root of unity $\omega = \exp(2\pi i/n)$. We have made the choice $\zeta = t\partial_t$ for which the coordinate $v = \lambda(\zeta) = x^8 + ix^9$ has the interpretation of two flat space-time coordinates in the weakly coupled IIA brane picture (see figure 13).

## 4.2 Constraints from Rotation

We rotate to $\mathcal{N} = 1$ by turning on the Higgs field $\varphi$ subject to the boundary conditions

$$t \to \infty: \qquad \varphi \to \mu\phi_\zeta. \tag{76}$$

The puncture at $t = 0$ is not rotated and $\varphi$ is required to be regular everywhere except $t = \infty$. At $t = \infty$ we therefore prescribe the asymptotics

$$t \to \infty: \qquad \varphi \sim \mu\Lambda t^{1/n}\mathrm{diag}\left(1, \omega, \omega^2, \cdots, \omega^{n-1}\right) + \cdots. \tag{77}$$

This constitutes a boundary value problem with bulk equations given by the BPS equations (55). Field theoretically, we are turning on a superpotential $W(\Phi) = \frac{\mu}{2}\mathrm{Tr}\,\Phi^2$ in the $\mathcal{N} = 2\ \mathfrak{su}(n)$ SYM theory, where $\Phi$ is an $\mathcal{N} = 1$ chiral multiplet living inside the $\mathcal{N} = 2$ vector multiplet. This deformation has been studied before and one expects only those points in $\mathcal{N} = 2$ Coulomb branch to survive, where all $A$-cycles of the SW curve (74) pinch to develop a nodal singularity. It is known that there are $n$ such points. Thus, we expect the existence of $n$ solutions to (55) corresponding to the $n$ points of the $\mathcal{N} = 2$ CB that are not lifted by the deformation (76).

Using (77), we deduce that

$$
\begin{aligned}
\mathrm{Tr}\,\varphi^k &= c_k && 2 \le k \le n-1 \\
\mathrm{Tr}\,\varphi^n &= n\mu^n\Lambda^n t + c_n
\end{aligned}
\tag{78}
$$

The above form of the Casimirs is valid over the whole sphere $\mathcal{C}$ for some constants $c_i$. Thus we can write the spectral equation $\det(w - \varphi) = 0$ as

$$\mathcal{Q}(w) = w^n - \sum_{k=2}^{n} c_k w^{n-k} - \mu^n\Lambda^n t = 0, \tag{79}$$

for some constants $c_i$.

## 4.3 Topological Factorization

As discussed in section 3.4, the solutions to the $\mathcal{N} = 1$ BPS equations (55) are curves $\Sigma_r \subset K_\mathcal{C} \oplus \mathcal{O}_\mathcal{C}(0)$ constituting $n$-fold coverings of $\mathcal{C}$. They are parametrized by $\lambda, w$ and combine the spectral curves of the Higgs fields $\phi, \varphi$ into a single covering. Contracting with $\zeta = t\partial_t$

we equivalently study the $n$-fold coverings parametrized by $v, w$ for the Higgs fields $\phi_\zeta, \varphi$. Crucially, the branch cut structures of the coordinates $v, w$ are required to match as otherwise the sheets of the spectral curves for $\phi_\zeta, \varphi$ can not be consistently combined into an $n$-fold covering. This allows us to describe a set of curves, which are topologically consistent, thereby improving on the constraints in section 4.2. Which of these also holomorphically satisfy the BPS-equations is then determined by computation with an ansatz derived from the branch cut structure of the candidate curves.

We begin by deriving the generic branch cut structures for the coordinate $v$ from the SW curve (74) and for the coordinate $w$ from the curve (79). The coordinate $v$ has in total two $\mathbb{Z}_n$ branch cuts emanating from the punctures at $t = 0, \infty$. The coordinate $w$ has a single $\mathbb{Z}_n$ branch cut emanating from the rotated puncture at $t = \infty$. The number of branch points for each cover is given by the degree in $t$ of the respective discriminants

$$\deg \Delta(\mathcal{P}, v) = 2n - 2, \qquad \deg \Delta(\mathcal{Q}, w) = n - 1. \tag{80}$$

Here $\Delta(\mathcal{P}, v)$ denotes the discriminant of the polynomial $\mathcal{P}$ with respect to the variable $v$. The branch points are given by the roots of (80). In the generic case, the discriminants (80) have isolated zeros and are associated with monodromy actions of order 2.

The SW curve is symmetric with respect to $t \to 1/t$ and the $2n - 2$ branch points come in pairs with identical monodromy action. We denote the cyclic permutation of the $n$ sheets as $a \in S_n$ and the transposition of the $i$-th and $j$-th sheet by $b_{i,j} \in S_n$. The generic branch cut structure for the coordinates $(v, w)$ can be described as[16]:

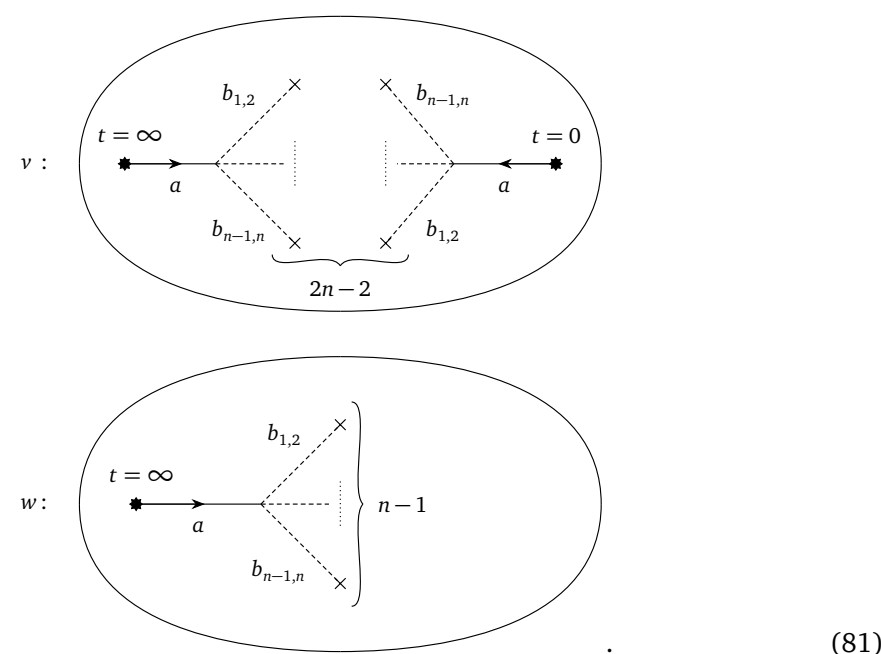

$$\tag{81}$$

Now we implement the topological factorization condition. The CB moduli $u_k$ and constants $c_k$ must be tuned such that the branch cut structures of $v, w$ coincide. The $n-1$ $\mathbb{Z}_2$-valued branch points of $w$ must collide at $t = 0$ to match the $\mathbb{Z}_n$ branch point of $\mathcal{P}(v)$ at $t = 0$. After implementing this, the branch cut structures of $v$ and $w$ can be described as:

---

[16]A summary of our notation for curves is given in appendix A.

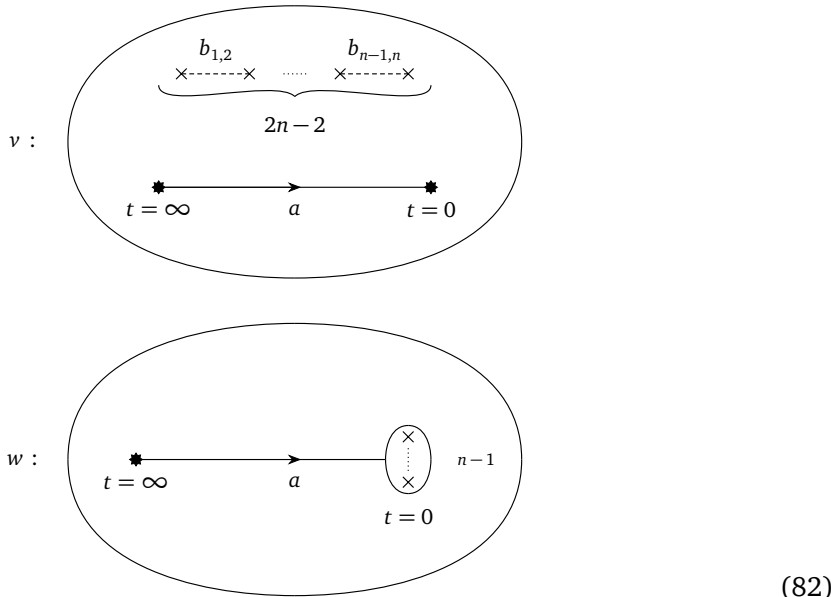

$$(82)$$

where we have rearranged the branch cuts for $v$ to match with branch cuts of $w$ at $t = 0$ and $t = \infty$, and we have denoted the collision of $n-1$ branch points by a circle surrounding the points. Since there are no more branch points for $w$, the $v$ curve cannot have any monodromy at $t \neq 0, \infty$ either. Thus, we find that the $2n-2$ $\mathbb{Z}_2$-valued branch points of $v$ must collide in pairs, eliminating all branch cuts not terminating at punctures. The final configurations for $v$ and $w$ are as follows:

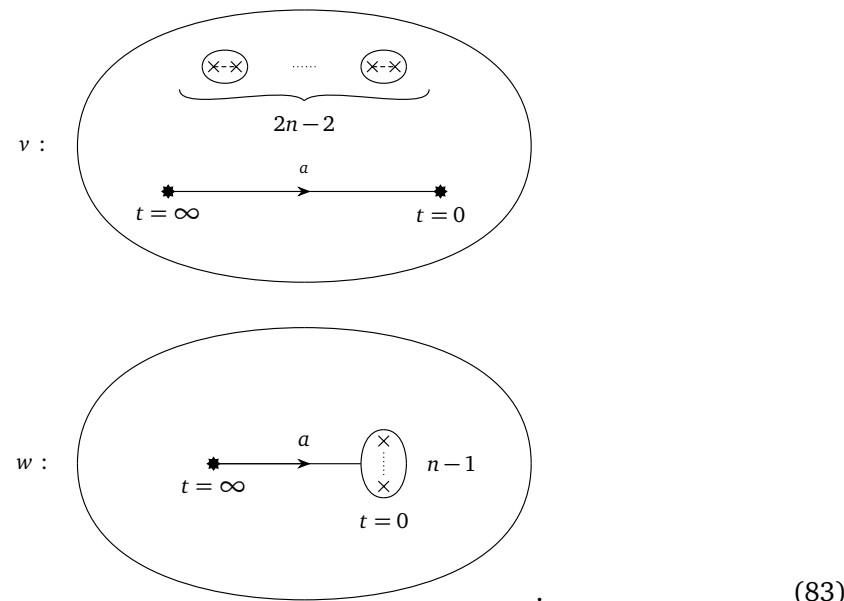

$$(83)$$

Here we have denoted the collision of branch points by a circle enclosing them.

## 4.4 Holomorphic Factorization

The topology (83) for $\mathcal{P}(v), \mathcal{Q}(w)$ constrains the coefficients $u_k, c_k$ in (74) and (79). It is known [15, 53] that the degeneration for $v$ in (83) fixes the CB parameters such that $P_n^{(r)}(v) = 2\Lambda^n T_n^{(r)}(v/2\Lambda)$ with $T_n^{(r)}(x) = T_n(e^{2\pi i r/2n}x)$ where $T_n$ is the $n$-th Chebyshev polynomial of the first kind. Due to the Weyl invariance $v \to -v$ this gives in total $n$ physically

distinct solutions. On the other hand, the degeneration for $w$ in (83) fixes all $c_i = 0$, otherwise the $n$ sheets for $\mathcal{Q}(w)$ cannot come together at $t = 0$. With this we find $n$ distinct solutions for $\mathcal{P}(v), \mathcal{Q}(w)$ associated to the topological degeneration (83) at finite values of $\mu$ to be

$$P_n^{(r)}(v) - \Lambda^n \left( t + \frac{1}{t} \right) = 0, \qquad w^n = \mu^n \Lambda^n t, \tag{84}$$

parametrized by $r = 0, \ldots, n-1$. We solve these equations for $v$. First, for $r = 0$, we find $v = \Lambda \left( t^{1/n} + t^{-1/n} \right)$ which follows from the properties of the Chebyshev polynomials. For general $r$, we send $v \to v e^{2\pi i r/2n}$ followed by a coordinate transformation $e^{\pi i r} t \to t$. This gives

$$v = \Lambda \left( t^{1/n} + \omega^r t^{-1/n} \right), \tag{85}$$

where $\omega = \exp(2\pi i/n)$. We can pair the $n$ values of $v$ with $n$ values of $w$ by requiring

$$vw = \mu \Lambda^2 \left( t^{2/n} + \omega^r \right). \tag{86}$$

Overall we find

$$\begin{aligned} v &= \Lambda \left( t^{1/n} + \omega^r t^{-1/n} \right), \\ w &= \mu \Lambda t^{1/n}, \end{aligned} \tag{87}$$

where the $n$ different values of $t^{1/n}$ parametrize the $n$ sheets of the $\mathcal{N} = 1$ curve $\Sigma_r$. In total, we have $n$ different $\mathcal{N} = 1$ curves corresponding to the $n$ different vacua of the rotated $\mathcal{N} = 1$ theory. The curves of these $n$ vacua are related via Dehn twists. Consider the pairing condition (85) near $t = 0$ where it reads $vw = \mu \Lambda^2 \omega^r$. Going between the $r$-th and $(r+1)$-th vaccum the pairing between the sheets is cyclically permuted. From $w = \mu \Lambda t^{1/n}$ we see that this shift can be realized by circling once around origin. It follows that the branch cuts of curves associated with neighbouring vacua are related by a Dehn twist and we therefore depict the branch cut structure of the curve $\Sigma_r$ as:

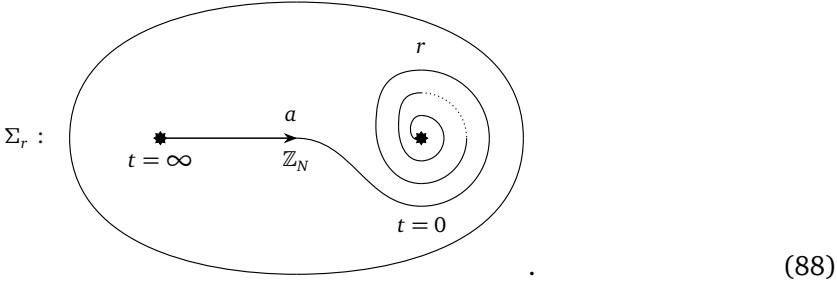

$$\tag{88}$$

The $v$-curve becomes singular and displays double points at those points in the CB that admit rotation (83). These singularities are removed in the $\mathcal{N} = 1$ curve as the double points are resolved to two points with distinct $w$-coordinates. The difference of the value for $w$ between these points encodes the vev of glueball superfield [10, 11, 13]. We redefine the coordinates to make this dimensionally manifest

$$v \to v/\Lambda, \qquad w \to w\Lambda, \tag{89}$$

and introduce the $\mathcal{N} = 1$ strong coupling scale $\Lambda_{\mathcal{N}=1}^3 = \mu \Lambda^2$. The new coordinates $(v, w)$ carry charges $(0, 2)$ under the $\mathbb{Z}_{2N}$ R-symmetry and are of mass dimension $(0, 3)$. The former now makes the cyclic rotation between the sheets paired in (88) clear, as the vacua are manifestly rotated into each other by the R-symmetry. Further we make the redefinition $v \to v - t^{1/n}$ and find

$$v^n = t, \qquad w^n = \Lambda_{\mathcal{N}=1}^{3n} t, \qquad vw = \Lambda_{\mathcal{N}=1}^3 \omega^r, \tag{90}$$

giving the $n$ curves described in [48]. Taking the limit $\mu \to \infty$ keeping $\Lambda_{\mathcal{N}=1}$ constant we are left with 4d $\mathcal{N} = 1$ $\mathfrak{su}(n)$ SYM [11]. Crucially the branch cut structure of the associated $\mathcal{N} = 1$ curves is not altered from (88) which correctly captures the topology of the $\mathcal{N} = 1$ curves associated with each SYM vacuum.

## 4.5 Line Operators and Confinement from the Curve

To discuss 1-form symmetry preserved in each of the $n$ vacua, we need to first choose a polarization $\Lambda \subset \mathcal{L}$ which determines the 1-form symmetry group $\widehat{\Lambda}$ of the absolute UV theory. Recall that for pure $\mathfrak{su}(n)$ SYM we have

$$\mathcal{L} \simeq \mathbb{Z}_n \times \mathbb{Z}_n. \tag{91}$$

The two factors are generated by $W$ and $H$, respectively, with the pairing

$$\langle W, W \rangle = \langle H, H \rangle = 0, \qquad \langle W, H \rangle = \frac{1}{n}. \tag{92}$$

Let us choose $\Lambda$ such that it contains Wilson line operators $ikW$ with $i \in \{0, 1, \cdots, l-1\}$ where $1 \le k, l \le n$ are integers such that $kl = n$. Then the gauge group of the corresponding absolute theory is

$$G = SU(n)/\mathbb{Z}_k, \tag{93}$$

where $\mathbb{Z}_k$ is the order $k$ subgroup of the center $\mathbb{Z}_n$ of the simply connected group $SU(n)$ associated to the gauge algebra $\mathfrak{g} = \mathfrak{su}(n)$. To make $\Lambda$ maximal, we can add to it the line operator $lH + mW$ where $m \in \{0, 1, \cdots, k-1\}$ is known as the discrete theta parameter associated to the absolute theory. Then

$$\Lambda = [kW, lH + mW] \subset \mathcal{L}, \tag{94}$$

is the subgroup of $\mathcal{L}$ generated by $kW$ and $lH + mW$.

The group structure of polarisation $\Lambda$ can be obtained by computing the Smith normal form of the following matrix associated to the generators of $\Lambda$

$$M_{klm} = \begin{pmatrix} k & 0 \\ m & l \end{pmatrix}. \tag{95}$$

A diagonal matrix $D$ is the Smith normal form of $M_{klm}$ if we find invertible integral matrices $S, T$ such that if $SM_{klm}T = D$. The form of $D$ is fixed to be $D = \text{diag}(d, N/d)$. We find that $d = \gcd(k, l, m)$ and correspondingly

$$\Lambda \cong \mathbb{Z}_d \times \mathbb{Z}_{n/d}, \tag{96}$$

is the group structure of $\Lambda$.

Now we compute the line operators $\mathcal{I}_r$ exhibiting perimeter law in vacuum $r$ from the topological structure (88) of the associated $\mathcal{N} = 1$ curve $\Sigma_r$. We can see that we need to encircle a puncture $n$ times to obtain a cycle on $\Sigma_r$ implying that $nW$ exhibits perimeter law, but $nW = 0$ in $\mathcal{L}$. On the other hand, following the branch cut, we see that $H + rW$ exhibits perimeter law, which is a non-trivial element of $\mathcal{L}$. Thus, $\mathcal{I}_r = [H + rW]$ is the subgroup of $\mathcal{L}$ generated by $H + rW$. The intersection $\Lambda_r = \mathcal{I}_r \cap \Lambda$ determines the line operators of the chosen absolute theory that exhibit perimeter law and we can compute it to be

$$\Lambda_r \cong \mathbb{Z}_{b_r}, \tag{97}$$

where $b_r = \gcd(k, m - lr)$. The 1-form symmetry group $\mathcal{O}_r$ preserved in $r$-th vacuum is now given by the Pontryagin dual of $\Lambda/\Lambda_r$ and crucially depends on the embedding $\iota_r : \Lambda_r \cong \mathbb{Z}_{b_r} \hookrightarrow \mathbb{Z}_d \times \mathbb{Z}_{n/d} \cong \Lambda$.

To compute $\mathcal{O}_r$ we denote the order $d, n/d$ generators of $\Lambda$ by $F, G$ respectively. Then we have the relation of bases $(F, G) = DT^{-1}(W, H)$. From (97) we find the generator of $\Lambda_r$ to be $B_r = (n/b_r)(H + rW)$. We expand this generator as $B_r = p_r F + q_r G$ with coefficients $(p_r, q_r) \in \mathbb{Z}_d \times \mathbb{Z}_{n/d}$. These coefficients follow in turn from $(p_r, q_r) = (n/b_r)(r, 1)TD^{-1}$. The quotient $\Lambda/\Lambda_r$ is computed using the Smith normal form (SNF) of the matrix

$$M_r = \begin{pmatrix} d & 0 \\ 0 & n/d \\ p_r & q_r \end{pmatrix} \xrightarrow{\text{SNF}} \begin{pmatrix} s_r & 0 \\ 0 & t_r \\ 0 & 0 \end{pmatrix}. \tag{98}$$

With this we find the 1-form symmetry group of the $r$-th vacuum to be

$$\mathcal{O}_r = \mathbb{Z}_{s_r} \times \mathbb{Z}_{t_r}, \tag{99}$$

where the integers $s_r, t_r$ can be computed form the minors of $M_r$ and are given by

$$s_r = \gcd\left(d, \frac{n}{d}, p_r, q_r\right), \qquad t_r = \gcd\left(n, \frac{np_r}{d}, dq_r\right). \tag{100}$$

Note that even if $s_r, t_r$ agree for two different vacua they are physically distinct if the associated embeddings $\iota_r : \mathbb{Z}_{b_r} \hookrightarrow \mathbb{Z}_d \times \mathbb{Z}_{n/d}$ differ. The embedding $\iota_r$ is determined by $B_r \in \Lambda$, that is the confining properties of two vacua $r_1, r_2$ differ if and only if $B_{r_1} \neq B_{r_2}$.

Before ending with two examples we give a simplification of the formula (99) for the case of $d = 1$ often encountered at low rank. In this case $s_r = 1$ and from $p_r = 0$ and $q_r = n/b_r$ it follows

$$d = 1: \qquad \mathcal{O}_r = \mathbb{Z}_{t_r} \tag{101}$$

with $t_r = \gcd(n, n/b_r) = n/b_r = n/\gcd(k, m - lr)$.

**Example**: Consider the gauge algebra $\mathfrak{su}(4)$. There are seven different choices for the spectrum of line operators [2]

$$
\begin{aligned}
SU(4): &\qquad \Lambda = \mathbb{Z}_4 = [W], \\
SU(4)/\mathbb{Z}_2: &\quad \begin{cases} SO(6)_+: & \Lambda = \mathbb{Z}_2^{(1)} \times \mathbb{Z}_2^{(2)} = [2W, 2H], \\ SO(6)_-: & \Lambda = \mathbb{Z}_4 = [W + 2H], \end{cases} \\
SU(4)/\mathbb{Z}_4: &\quad \begin{cases} \left(SU(4)/\mathbb{Z}_4\right)_0: & \Lambda = \mathbb{Z}_4 = [H], \\ \left(SU(4)/\mathbb{Z}_4\right)_1: & \Lambda = \mathbb{Z}_4 = [W + H], \\ \left(SU(4)/\mathbb{Z}_4\right)_2: & \Lambda = \mathbb{Z}_4 = [2W + H], \\ \left(SU(4)/\mathbb{Z}_4\right)_3: & \Lambda = \mathbb{Z}_4 = [3W + H]. \end{cases}
\end{aligned}
\tag{102}
$$

The line operators exhibiting perimeter law in each vacuum are given by the intersection $\mathcal{I}_r \cap \Lambda$ and computed mod 4 for $r = 0, 1, 2, 3$ respectively to be

$$
\begin{aligned}
SU(4): &\quad \mathcal{I}_r \cap \Lambda = 0, 0, 0, 0, \\
SO(6)_+: &\quad \mathcal{I}_r \cap \Lambda = [2H], [2W + 2H], [2H], [2W + 2H], \\
SO(6)_-: &\quad \mathcal{I}_r \cap \Lambda = 0, 0, 0, 0, \\
\left(SU(4)/\mathbb{Z}_4\right)_0: &\quad \mathcal{I}_r \cap \Lambda = [H], 0, [2H], 0, \\
\left(SU(4)/\mathbb{Z}_4\right)_1: &\quad \mathcal{I}_r \cap \Lambda = 0, [W + H], 0, [2W + 2H], \\
\left(SU(4)/\mathbb{Z}_4\right)_2: &\quad \mathcal{I}_r \cap \Lambda = [2H], 0, [2W + H], 0, \\
\left(SU(4)/\mathbb{Z}_4\right)_3: &\quad \mathcal{I}_r \cap \Lambda = 0, [2W + 2H], 0, [3W + H].
\end{aligned}
\tag{103}
$$

The 1-form symmetry preserved in each vacuum is given by the Pontryagin dual of the quotient $\Lambda/(\mathcal{I}_r \cap \Lambda)$

$$
\begin{aligned}
SU(4) : \quad & \mathcal{O}_r = \mathbb{Z}_4, \\
SO(6)_+ : \quad & \mathcal{O}_r = \mathbb{Z}_2^{(1)}, \mathbb{Z}_2^{(3)}, \mathbb{Z}_2^{(1)}, \mathbb{Z}_2^{(3)}, \\
SO(6)_- : \quad & \mathcal{O}_r = \mathbb{Z}_4, \\
\left(SU(4)/\mathbb{Z}_4\right)_0 : \quad & \mathcal{O}_r = 0, \mathbb{Z}_4, \mathbb{Z}_2, \mathbb{Z}_4, \\
\left(SU(4)/\mathbb{Z}_4\right)_1 : \quad & \mathcal{O}_r = \mathbb{Z}_4, 0, \mathbb{Z}_4, \mathbb{Z}_2, \\
\left(SU(4)/\mathbb{Z}_4\right)_2 : \quad & \mathcal{O}_r = \mathbb{Z}_2, \mathbb{Z}_4, 0, \mathbb{Z}_4, \\
\left(SU(4)/\mathbb{Z}_4\right)_3 : \quad & \mathcal{O}_r = \mathbb{Z}_4, \mathbb{Z}_2, \mathbb{Z}_4, 0.
\end{aligned}
\tag{104}
$$

Here $\mathbb{Z}_2^{(3)}$ is the diagonal subgroup of $\mathbb{Z}_2^{(1)} \times \mathbb{Z}_2^{(2)}$. The 1-form symmetries of $\left(SU(4)/\mathbb{Z}_4\right)_k$ are cyclic permutations of each other induced by shifts of the theta angle $\theta \to \theta + 2\pi$.

Note that the $SO(6)_+$ theory is obtained from the $SU(4)$ theory by gauging the $\mathbb{Z}_2$ subgroup of the $\mathbb{Z}_4$ 1-form symmetry of the $SU(4)$ theory. The resulting $\mathbb{Z}_2 \times \mathbb{Z}_2$ 1-form symmetry thus has a mixed anomaly [54], which is captured by the Bockstein of the extension $1 \to \mathbb{Z}_2 \to \mathbb{Z}_4 \to \mathbb{Z}_2 \to 1$. It would be interesting to see this anomaly from a direct reduction starting with the 6d anomaly polynomial.

**Example**: Consider the gauge algebra $\mathfrak{su}(12)$ with the polariztation $\Lambda = [6W, 4W+2H]$. Using (96) we find $\Lambda \cong \mathbb{Z}_2 \times \mathbb{Z}_6$. The generators of each factor are $(F, G) = (6H, 4H + 2W)$. There are 12 vacua labelled by $r = 0, \ldots, 11$ and with respect to the basis $F, G$ the generators of $\Lambda_r$ have the coordinates

$$
\left(
\begin{array}{c|cccccccccccc}
r & 0 & 1 & 2 & 3 & 4 & 5 & 6 & 7 & 8 & 9 & 10 & 11 \\
\hline
p_r & 1 & 1 & 1 & 1 & 1 & 1 & 1 & 1 & 1 & 1 & 1 & 1 \\
q_r & 0 & 3 & 2 & 3 & 0 & 5 & 0 & 3 & 2 & 3 & 0 & 2
\end{array}
\right).
\tag{105}
$$

The 1-form symmetry $\mathcal{O}_r$ preserved in each vacuum now follows from appending the columns of (105) as a row to the diagonal matrix $\mathrm{diag}(2, 6)$ and computing the diagonal entries of its Smith normal form. The 1-form symmetries preserved in each vacuum are isomorphic to:

$$
\begin{array}{c|cccccccccccc}
r & 0 & 1 & 2 & 3 & 4 & 5 & 6 & 7 & 8 & 9 & 10 & 11 \\
\hline
\mathcal{O}_r & \mathbb{Z}_6 & \mathbb{Z}_6 & \mathbb{Z}_2 & \mathbb{Z}_6 & \mathbb{Z}_6 & \mathbb{Z}_2 & \mathbb{Z}_6 & \mathbb{Z}_6 & \mathbb{Z}_2 & \mathbb{Z}_6 & \mathbb{Z}_6 & \mathbb{Z}_2
\end{array}.
\tag{106}
$$

The set of line operators displaying perimeter and area law differ for vacua with distinct $(p_r, q_r)$ even if their confinement indices agree. As in the previous example, there is a mixed 1-form symmetry anomaly associated to the extension $1 \to \mathbb{Z}_6 \to \mathbb{Z}_{12} \to \mathbb{Z}_2 \to 1$ [55].

# 5 Confinement Index for $\mathcal{N} = 1$ SYM with Adjoint Chiral

In this section we consider another rotation of $\mathcal{N} = 2$ SYM with gauge algebra $\mathfrak{g} = \mathfrak{su}(n)$. This rotation corresponds to turning on a generic tree-level cubic superpotential for the $\mathcal{N} = 1$ adjoint chiral multiplet living in the $\mathcal{N} = 2$ vector multiplet

$$
W(\Phi) = \frac{g}{3} \operatorname{Tr} \Phi^3 + \frac{\mu}{2} \operatorname{Tr} \Phi^2 .
\tag{107}
$$

We also briefly discuss rotations corresponding to generic superpotentials of higher order

$$
W(\Phi) = \sum_{i=2}^{k} \frac{g_i}{i} \operatorname{Tr} \Phi^i ,
\tag{108}
$$

for $k \leq n$, which are analyzed similarly. Note that the mass dimension of $g_i$ is negative for $i \geq 4$, so the higher order superpotentials are non-renormalizable and the resulting $\mathcal{N} = 1$ theory needs a UV cutoff to be well-defined (see the related discussion in section 3.3.3).

Field theoretically the confining properties of the theory after turning on these superpotentials was studied in [21], which we briefly review before turning to the main discussion deriving these properties from $\mathcal{N} = 1$ curves. Classical vacua are given by diagonal configurations of $\Phi$ with eigenvalues extremizing the superpotential. Classical vacua are therefore labelled by $k-1$ integers $(n_1, n_2, \cdots, n_{k-1})$ (such that $n_1 + n_2 + \cdots + n_{k-1} = n$) counting the number of eigenvalues fixed to the $k-1$ different critical points of the superpotential. In such a vacuum the gauge symmetry is broken as

$$\mathfrak{su}(n) \quad \rightarrow \quad \mathfrak{su}(n_1) \oplus \mathfrak{su}(n_2) \oplus \cdots \oplus \mathfrak{su}(n_{k-1}) \oplus \mathfrak{u}(1)^{k-2}. \tag{109}$$

At low energies the non-abelian factors decouple and individually confine, and the system settles in one of $n_1 n_2 \cdots n_{k-1}$ quantum vacua, leaving an abelian gauge theory. We label these vacua by integers $(r_1, r_2, \cdots, r_{k-1})$ where $r_i \in \{0, \ldots, n_i - 1\}$. The Wilson and 't Hooft lines $W_i, H_i$ of each non-abelian factor $\mathfrak{su}(n_i)$ are identified as Wilson and 't Hooft lines $W, H$ of the initial $\mathfrak{su}(n)$, which can be used to read the confining properties. We expect $n_i W_i, H_i + r_i W_i$ for each $i$ to exhibit perimeter law, implying perimeter laws for $n_i W, H + r_i W$ for each $i$. Thus the set of lines exhibiting perimeter law in vacuum $(r_1, r_2, \cdots, r_{k-1})$ can be written as

$$\mathcal{I}_{r_1, r_2, \cdots, r_{k-1}} = [H + r_1 W, \gcd(n_1, n_2, \cdots, n_{k-1}, r_1 - r_2, r_2 - r_3, \cdots, r_{k-2} - r_{k-1})W], \tag{110}$$

which is a subgroup of $\mathcal{L} \simeq \mathbb{Z}_n^W \times \mathbb{Z}_n^H$. The confining properties of each vacuum can now be determined once one chooses a polarization. For instance, if one chooses the purely electric polarization $\Lambda = \mathbb{Z}_n^W$, then the 1-form symmetry group preserved in vacuum $(r_1, r_2, \cdots, r_{k-1})$ is

$$\mathcal{O}_{r_1, r_2, \cdots, r_{k-1}} = \mathbb{Z}_t \; ; \qquad t = \gcd(n_1, n_2, \cdots, n_{k-1}, r_1 - r_2, r_2 - r_3, \cdots, r_{k-2} - r_{k-1}), \tag{111}$$

where $t$ is known as the confinement index of the vacuum.

## 5.1 Constraints from Rotation to $\mathcal{N} = 1$

The starting point is the Seiberg-Witten curve for $\mathfrak{g} = \mathfrak{su}(n)$ $\mathcal{N} = 2$ SYM

$$\mathcal{P}(v) = \det(v - \phi_\zeta) = P_n(v) - \Lambda^n \left( t + \frac{1}{t} \right) = 0. \tag{112}$$

Here $\phi_\zeta$ denotes the contraction of the Higgs field $\phi$ with the vector $\zeta = t \partial_t$ and $P_n(v) = v^n + \sum_{i=2}^n u_i v^{n-i}$ with the $u_i$ parametrizing the CB. Irregular punctures of type $\mathcal{P}_0$ are located at $t = 0, \infty$. The cubic superpotential (107) can be turned on by rotating the puncture at $t = \infty$ such that the Higgs field $\varphi$ is subject to the boundary conditions

$$t \to \infty : \qquad \varphi \to g \phi_\zeta^2 + \mu \phi_\zeta. \tag{113}$$

The asymptotics of the Higgs field $\phi$ at the puncture $t = \infty$ read

$$t \to \infty : \qquad \phi_\zeta = \Lambda t^{1/n} \text{diag}\left(1, \omega, \omega^2, \cdots, \omega^{n-1}\right) + \ldots, \tag{114}$$

at the puncture $t = \infty$, which follows from (112). Here $\omega = \exp(2\pi i/n)$. The eigenvalues of $\phi_\zeta$ grow as $\Lambda |t|^{1/n}$ whereby those of $\varphi$ grow as $g \Lambda^2 |t|^{2/n}$ as $t \to \infty$. This restricts the $w$-curve $\mathcal{Q}(w) = \det(w - \varphi) = 0$ to take the form

$$\mathcal{Q}(w) = w^n + bt^2 + t \sum_{k=0}^{\lfloor n/2 \rfloor} d_k w^k + \sum_{k=0}^{n-1} c_k w^k, \tag{115}$$

for some complex constants $b, c_k, d_k$. The boundary condition (113) fixes the terms of maximal growth $\mathcal{O}(t^2)$, i.e. the coefficients $b, d_{n/2}$ or $b$ when $n$ is even or odd respectively. This follows by substituting the asymptotics (114) into the boundary condition (113) and collecting all terms of the $w$-curve which do not receive contributions from the lower order terms. For even $n$ we find

$$\mathcal{Q}(w) = \left(w^{n/2} - g^{n/2}\Lambda^n t\right)^2 + t \sum_{k=0}^{n/2-1} d_k w^k + \sum_{k=0}^{n-1} c_k w^k \,, \tag{116}$$

while for odd $n$ we have

$$\mathcal{Q}(w) = w^n - g^n \Lambda^{2n} t^2 + t \sum_{k=0}^{(n-1)/2} d_k w^k + \sum_{k=0}^{n-1} c_k w^k \,. \tag{117}$$

This form of the $w$-curve follows purely from the prescribed behavior of $\varphi$ at the puncture $t = \infty$. We can improve on this ansatz for even $n$ with $n_1 = n_2$ and $r_1 = r_2$ where a spontaneously broken R-symmetry is restored. While select, these cases display interesting screening effects, as seen from the confinement index (111). We finish this section with a discussion of this symmetry enhancement before improving on the above Ansätze by prescribing branch points and cuts away from the punctures.

The superpotential (107) has two critical points associated to $\mathfrak{su}(n_1)$ and $\mathfrak{su}(n_2)$ gauge algebras arising at low energies. The exchange of these two critical points is an R-symmetry of the theory[17] which acts on the quantum vacua by $n_1 \leftrightarrow n_2$ and $r_1 \leftrightarrow r_2$. Thus, this symmetry is spontaneously preserved only in the vacua characterized by $n_1 = n_2$ and $r_1 = r_2$. The symmetry changes the sign of $v$ while leaving $w$ invariant and we therefore expect the eigenvalues of $\varphi$ to come in identical pairs. The $w$-curve must only have double roots and this improves the ansatz (116) to

$$\mathcal{Q}(w) = \left(w^{n/2} - g^{n/2}\Lambda^n t + \sum_{k=0}^{n/2-1} e_k w^k\right)^2 \,, \tag{118}$$

introducing the complex constants $e_k$. The sheets of the $v$-curve on the other hand must come in pairs related by the symmetry $v \leftrightarrow -v$, in particular there must be an involutive renumbering of the sheets which leaves the branch cuts structure invariant. The $v$-curve is an $n$-fold cover of the Gaiotto curve and the $w$-curve is an $n$-fold cover which degenerated to an $(n/2)$-fold cover as seen from the perfect square (118). We call latter as the reduced $w$-curve. By the above the branch cut structure of the reduced $w$-curve is the $\mathbb{Z}_2$ quotient of that of the $v$-curve. We discuss this very interesting R-symmetry for $\mathfrak{g} = \mathfrak{su}(4)$ at length in section C.2.

## 5.2 Topological Factorization

We begin by determining the ramification structure of the generic $v$- and $w$-curves from (112) and (116) or (117). The $v$-curve has two $\mathbb{Z}_n$ branch cuts emanating from the punctures at $t = 0, \infty$. Further there are $\deg \Delta(\mathcal{P}, v) = 2n - 2$ branch points terminating $\mathbb{Z}_2$ branch cuts. The $w$ curves (116), (117) have a single $\mathbb{Z}_n$ branch cut emanating from the rotated puncture at $t = \infty$. The discriminant $\Delta(\mathcal{Q}, w)$ has order

$$\deg \Delta(\mathcal{Q}, w) = \begin{cases} 2n - 3\,, & n \text{ even and } r_1 \neq r_2 \text{ whenever } n_1 = n_2 \\ 2n - 2\,, & n \text{ odd}, \end{cases} \tag{119}$$

---

[17]To see this, shift $v \to v - \mu/2g$ for which we have $W'(v) = gv^2 - \mu^2/4g$. The symmetry is now implemented by $v \to -v$ which changes the sign of the superpotential, and hence the symmetry is an R-symmetry.

which is equal to the number of branch points terminating $\mathbb{Z}_2$ branch cuts. The generic branch cut structure therefore takes the form[18]

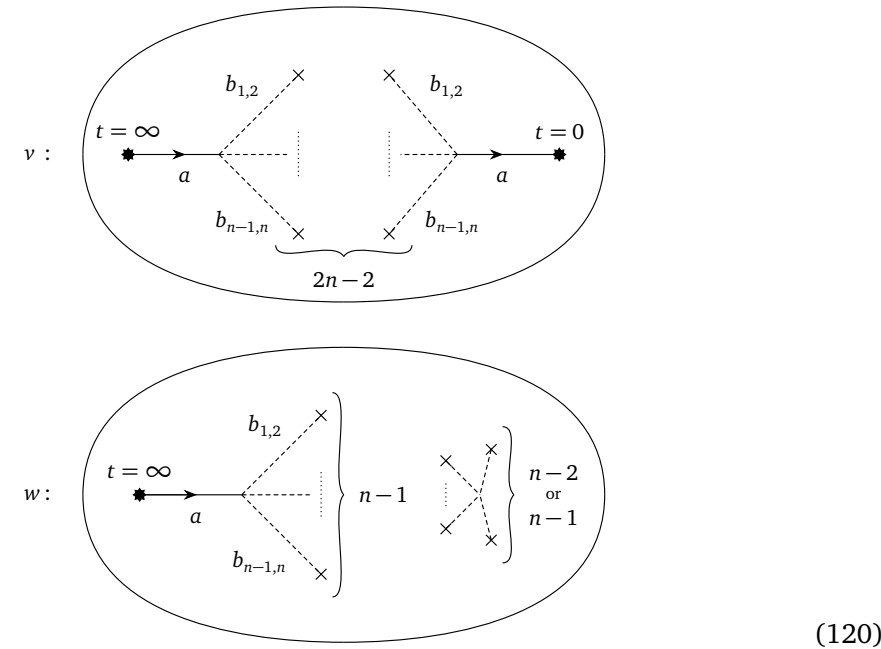

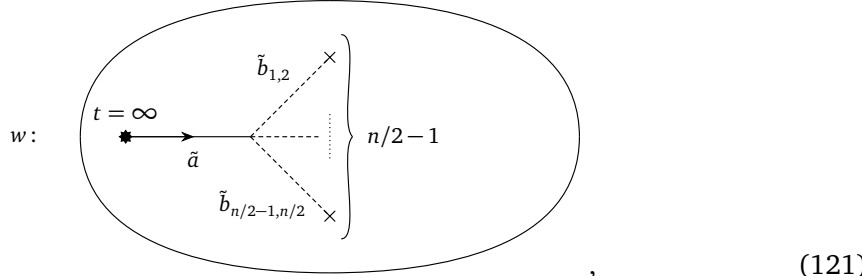

$$(120)$$

with $n-2$ and $n-1$ branch points not connected to $t=\infty$ in the $w$-curve for $n$ even and odd respectively. When $n$ is even and $n_1 = n_2$ and $r_1 = r_2$ the $w$-curve takes the form (118). We remove the doubling up of roots by considering the $(n/2)$-fold cover $\mathcal{Q}^{1/2}(w) = 0$ for which we find to have the branch cut structure:

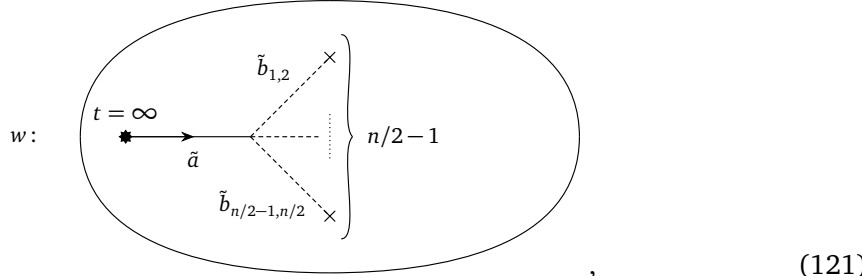

$$,\qquad(121)$$

where $\tilde{a}$ denotes the cyclic permutation of the $n/2$ sheets and $\tilde{b}_{ij}$ a transposition of the $i$-th and $j$-th sheet.

We now implement the factorization condition topologically. The gauge symmetry breaking to $\mathfrak{su}(n_1) \oplus \mathfrak{su}(n_2) \oplus \mathfrak{u}(1)$ demands the presence of $n_1 + n_2 - 2 = n - 2$ mutually local massless monopoles. We need to restrict to the CB sublocus on which the SW curve degenerates to genus one (here $\varphi$ can be turned on). Therefore this condition is addressed at the level of the $v$-curve and met by colliding all but one pair of branch points of the $v$-curve, which are related by the $t \leftrightarrow 1/t$, at $t = \pm 1$. When $n$ is even/odd there are three/two possibilities for the number of branch points colliding in at $t = \pm 1$. This follows from considering the limit $\Lambda \to 0$, which does not change the topology of the SW curve (or the $\mathcal{N} = 1$ curve after rotation). In this limit the dynamics of the $\mathfrak{su}(n_i)$ factors decouple and each subsector is described by its own SW curve $\mathcal{P}_{n_i} = P_{n_i}(v) - \Lambda^{n_i}(t + 1/t) = 0$. The polynomials $P_{n_i}(v)$ are the Chebyshev polynomials. The discriminant of these SW curves then takes the form

$$\Delta(\mathcal{P}_{n_i}, v) \sim (t+1)^{2k_{i,-}}(t-1)^{2k_{i,+}},\qquad(122)$$

___
[18]We remind the reader that a summary of our notation is in appendix A.

where $k_{i,-} = k_{i,+} + 1 = n_i/2$ and $k_{i,-} = k_{i,+} = (n_i - 1)/2$ for even and odd $n_i$, respectively
For example when $n_i = 2, 3, 4$ we have $k_{i,-} = 1, 1, 2$ and $k_{i,+} = 0, 1, 1$ respectively. Here $k_{i,\pm}$
denotes the number of branch points collided at $t = \pm 1$. Naively this suggests that a given
partition $n = n_1 + n_2$ fixes the number of branch points collided at $t = \pm 1$ to $k_\pm = k_{1,\pm} + k_{2,\pm}$.
However taking the chiral symmetry $t \to -t$ and $\Lambda^n \to -\Lambda^n$ of the $\mathfrak{su}(n)$ theory into account
we find that for every factorization of the discriminant $\Delta(\mathcal{P}, v)$ characterized by $(k_+, k_-)$ there
must also exist one with $(k_-, k_+)$ in the $\mathcal{N} = 2$ CB as the chiral symmetry maps $k_+ \leftrightarrow k_-$.
Therefore the discriminant of the rank $n$ SW curve takes one of the three possible forms

$$\Delta(\mathcal{P}, v) \sim \begin{cases} (t+1)^{n/2}(t-1)^{n/2-1} \\ (t+1)^{n/2}(t-1)^{n/2} \\ (t+1)^{n/2-1}(t-1)^{n/2} \end{cases}, \tag{123}$$

for even $n$, while taking one of the two forms

$$\Delta(\mathcal{P}, v) \sim \begin{cases} (t+1)^{(n-3)/2+1}(t-1)^{(n-3)/2} \\ (t+1)^{(n-3)/2}(t-1)^{(n-3)/2+1} \end{cases}, \tag{124}$$

for odd $n$ along loci of complex dimension one in the $\mathcal{N} = 2$ CB at which the gauge symmetry
enhances to $\mathfrak{su}(n_1) \oplus \mathfrak{su}(n_2) \oplus \mathfrak{u}(1)$. These loci are not irreducible in general. However, in
the low rank examples with gauge algebra $\mathfrak{g} = \mathfrak{su}(3), \mathfrak{su}(4)$ considered appendix C we have
precisely $2, 3$ such irreducible subloci characterized by (124), (123), respectively.

When the unbroken gauge symmetry is $\mathfrak{su}(n_1) \oplus \mathfrak{su}(n_2) \oplus \mathfrak{u}(1)$ the deformed $\mathcal{N} = 1$ gauge
theory has $n_1 n_2$ vacua. Each such vacuum is associated with an $\mathcal{N} = 1$ curve which in turn
follows from a rotation of the $v$-curve described above. Different $v$-curves are related by partial
Dehn twists, which result from movements of branch-points on the base curve. This follows
again from studying the topology of the SW curve in the limit $\Lambda \to 0$, where the dynamics of the
$\mathfrak{su}(n_i)$ factors decouple. There are $n_i - 1$ massless monopoles associated with each factor and
correspondingly $n_i - 1$ pairs of branch points colliding at $t = \pm 1$. These are in correspondence
with a subset of the branch points of the full $\mathfrak{su}(n)$ theory. The problem essentially factorizes
and we can determine for each $\mathfrak{su}(n_i)$ the set of branchpoint movements associated to different
Dehn twists, and then superpose them.

We therefore need to understand the movement of the branch points for the Chebyshev
polynomials when going between the monopole points of the $\mathfrak{su}(n_i)$ SW curve via the phase
rotation $v \to e^{\pi i/n_i} v$. These are precisely the Dehn twists discussed in section 4.

Consider the $v$-curve preparing the rotation to the vacuum labelled $(r_1, r_2) = (0, 0)$ with
the integer double $(n_1, n_2)$. We have the branch cut structure

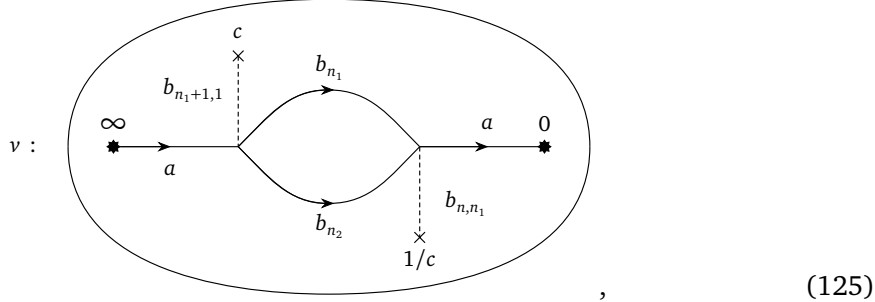

$$\tag{125}$$

where $b_{n_1}$ and $b_{n_2}$ act via cyclic permutation on the first $n_1$ and last $n_2$ sheets respectively,
i.e. the central branch lines are commuting and of monodromy type $\mathbb{Z}_{n_1}$ and $\mathbb{Z}_{n_2}$. The branch
lines connecting to $t = c, 1/c$ are $\mathbb{Z}_2$ branch lines. For clarity we depict (125) once more, now
labelling the branch cuts by their monodromy subgroups

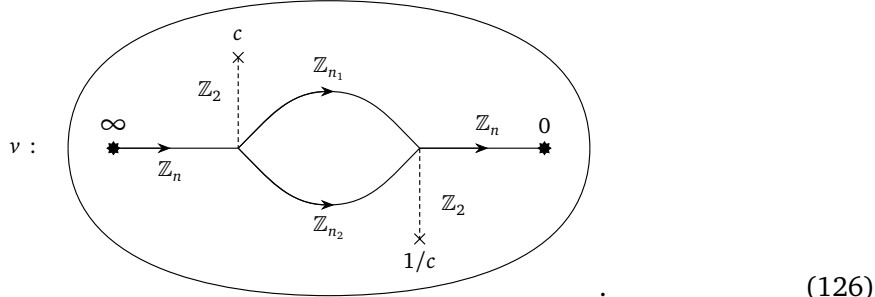

$$\text{(126)}$$

From (125) we generate a total of $n_1 n_2$ branch cut structures, labelled by integers $(r_1, r_2)$, by wrapping the $\mathbb{Z}_{n_1}$ and $\mathbb{Z}_{n_2}$ branch lines $r_1$ and $r_2$ times around the vertical equator of the Gaiotto curve respectively. We call the move individually increasing or decreasing $r_1, r_2$ by one a positive or negative partial Dehn twist respectively, while the move simultaneously increasing or decreasing $r_1, r_2$ by one is referred to as an overall positive or negative Dehn twist respectively. For instance, the $(1, 0)$ and $(0, 1)$ configurations respectively are

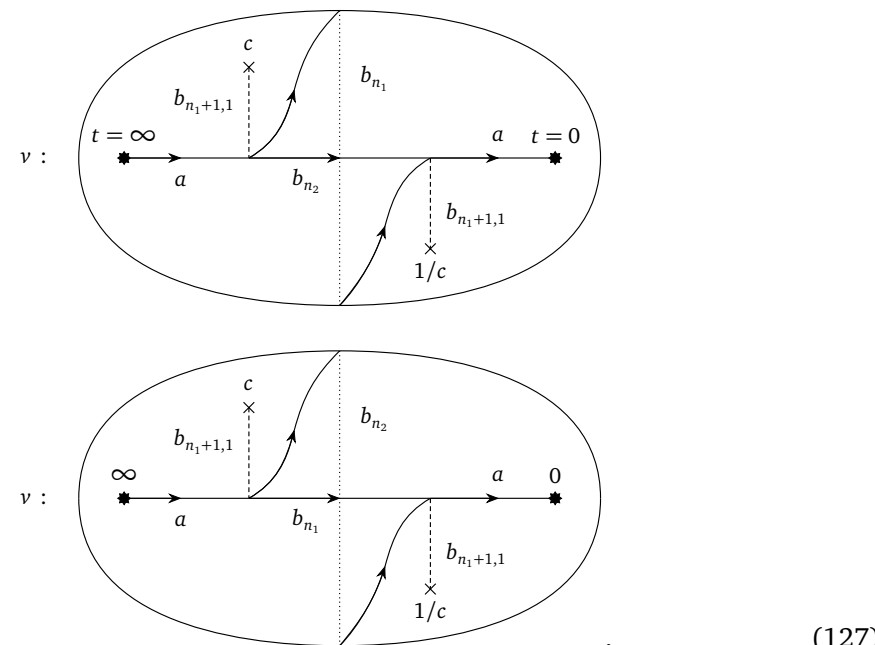

$$\text{(127)}$$

where the dotted lines depict a branch cut wrapping around the twice punctured sphere $C$. Wrapping the $\mathbb{Z}_{n_i}$ branch line $n_i$ times around the sphere the $n_i$ loops can be stacked and trivialize, i.e. we have the equivalence of configurations $(r_1, r_2) \sim (r_1 + n_1, r_2) \sim (r_1, r_2 + n_2)$ for a given partition $n = n_1 + n_2$. Therefore we restrict to the labels to run as $r_i = 0, \ldots, n_i - 1$ and for every partition $n = n_1 + n_2$ we therefore have $n_1 n_2$ branch cut structures of the type

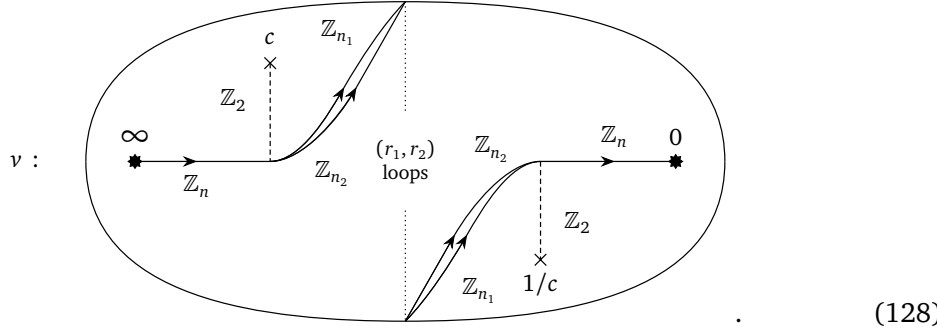

$$\text{(128)}$$

We turn to the $w$-curve. The branch points of the generic $w$-curve (120) must be moved to match the branch cut structure of the $v$-curve (128). The $v$-curve has a $\mathbb{Z}_n$ branch line terminating at $t = 0$. The $w$-curve is therefore required to have a branch point at $t = 0$ similarly terminating a $\mathbb{Z}_n$ branch line. This follows by colliding branch points on the lhs of (120) at $t = 0$. This move is realized by setting $c_k = 0$ in (116) and (117) improving the ansätze for the $w$-curve to

$$\mathcal{Q}(w) = w^n + g^n \Lambda^{2n} t^2 - 2g^{n/2} \Lambda^n w^{n/2} t + \sum_{k=0}^{n/2-1} d_k w^k t, \tag{129}$$

for even $n$ and to

$$\mathcal{Q}(w) = w^n - g^n \Lambda^{2n} t^2 + \sum_{k=0}^{\lceil n/2 \rceil - 1} d_k w^k t, \tag{130}$$

for odd $n$ as otherwise total ramification at $t = 0$ is not possible. For cases with $n_1 = n_2$ and $r_1 = r_2$ this move sets $e_k = 0$ in (118) which fully fixes the curve to

$$\mathcal{Q}(w) = \left( w^{n/2} - g^{n/2} \Lambda^n t \right)^2. \tag{131}$$

The remaining $\lceil n/2 \rceil$ complex constants $d_k$ in (129) and (130) are determined, up to discrete choices, by colliding all but a pair of the $n-2$ or $n-1$ branch points shown on the rhs in (120) and moving the two unpaired branch points to $t = c, 1/c$. The former fixes $\lceil n/2 \rceil - 2$ parameters and the latter the remaining 2. For a given $v$-curve with branch points at $t = c, 1/c$ we must pick the $w$-curve with the same branch cut structures of this discrete set of solutions. This finally determines the unique $w$-curve to a given $v$-curve, both share the branch cut structure shown in (128). In the cases with $n_1 = n_2$ and $r_1 = r_2$ the $w$-curve (121) however simply takes the form

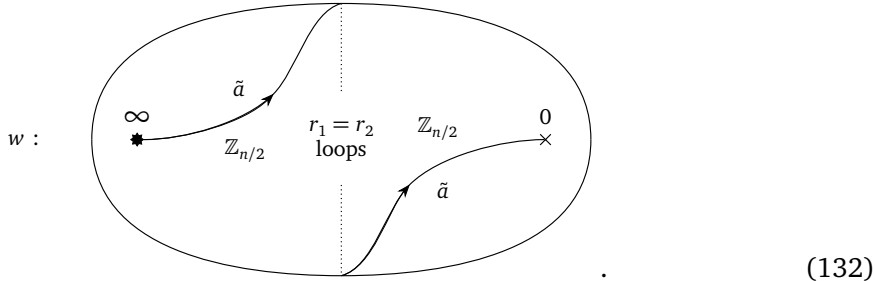

$$\tag{132}$$

This $w$-curve is the $\mathbb{Z}_2$ quotient of the corresponding $v$-curve (128) via identification of the sheets related through the unbroken $\mathbb{Z}_2$ R-symmetry.

The $\mathcal{N} = 1$ curve is now the diagonal of the $v, w$-curves. If both curves are given by (128) the $\mathcal{N} = 1$ curve is given by the $n$ pairs $(v_i, w_i)$ sweeping out its $i$-th sheet. For the cases with $n_1 = n_2$ and $r_1 = r_2$ with $v$-curve (128) and $w$-curve (132), the $\mathbb{Z}_2$ R-symmetry maps the $j$-th sheet of the $v$-curve to the $\tilde{j}$-th sheet where we number the sheets as $j = 1, \ldots, n/2$ and $\tilde{j} = n/2+1, \ldots, n$. The $\mathcal{N} = 1$ curve in these cases is given by the two sets of $n/2$ pairs $(v_j, w_j)$ and $(v_{\tilde{j}}, w_{\tilde{j}})$. In both cases the $\mathcal{N} = 1$ curve is an $n$-fold cover of the Gaiotto curve.

## 5.3 Line Operators, Confinement and Higher-order Superpotentials

Consider the CSW $\mathcal{N} = 1$ curve associated to the $(r_1, r_2)$ vacuum for partition $n = n_1 + n_2$ given by

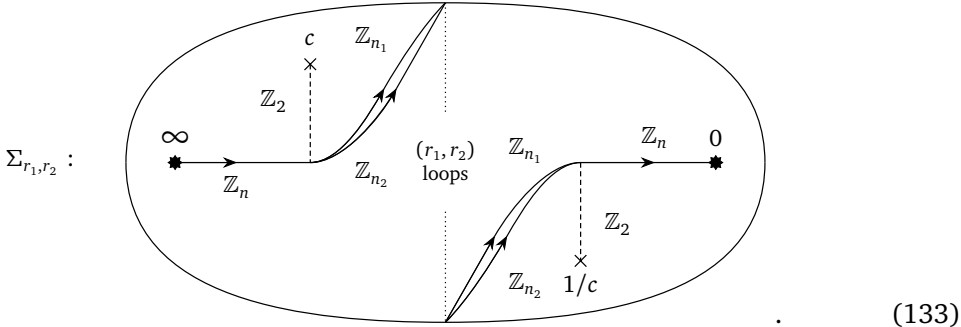

$$. \tag{133}$$

We now determine, using the curve (133) and following the procedure of section 3.5, the set $\mathcal{I}_{r_1,r_2}$ of line operators that would exhibit perimeter law in the $\mathcal{N}=1$ vacuum described by the curve (133). The central $\mathbb{Z}_{n_1}, \mathbb{Z}_{n_2}$ monodromy actions commute and act on a disjoint set of sheets. Starting from a point lying along the equator and on the sheet acted upon by the $\mathbb{Z}_{n_i}$ action, we return to the same sheet if we go around the equator $n_i$ times. Thus $n_i W \in \mathcal{I}_{r_1,r_2}$ for $i = 1, 2$. Now, start from the puncture at infinity from a sheet acted upon by the $\mathbb{Z}_{n_1}$ action, and follow the $\mathbb{Z}_n, \mathbb{Z}_{n_1}$ branch cuts to reach the puncture at $t = 0$. On this path, one is allowed to cross the $\mathbb{Z}_{n_2}$ branch cut, but not the $\mathbb{Z}_n, \mathbb{Z}_{n_1}, \mathbb{Z}_2$ branch cuts. Since $\mathbb{Z}_{n_2}$ does not act on this sheet, one remains on the same sheet and obtains a cycle on the $\mathcal{N}=1$ curve $\Sigma_{r_1,r_2}$ which projects to the cycle $H + r_1 W$ on the Gaiotto curve $\mathcal{C}$. Similarly, there is a cycle on $\Sigma_{r_1,r_2}$ which projects onto the cycle $H + r_2 W$ on $\mathcal{C}$. Thus, in total,

$$\mathcal{I}_{(r_1,r_2)} = [n_1 W, n_2 W, H + r_1 W, H + r_2 W] = [H + r_1 W, \gcd(n_1, n_2, r_1 - r_2) W], \tag{134}$$

matching the field theory expectation (110).

In a similar fashion, one can study the theory deformed by a general higher-order superpotential (108). This deformation can be achieved by performing the rotation

$$t \to \infty: \qquad \varphi \to \sum_{i=2}^{k} g_i \phi_\zeta^{i-1}. \tag{135}$$

Consider a vacuum specified by partition $n_1 + n_2 + \cdots + n_{k-1} = n$ and integers $(r_1, r_2, \cdots, r_{k-1})$ with $r_i \in \{0, \ldots, n_i - 1\}$. The $\mathcal{N}=1$ curve $\Sigma_{r_1,r_2,\cdots,r_{k-1}}$ has the following branch cut structure

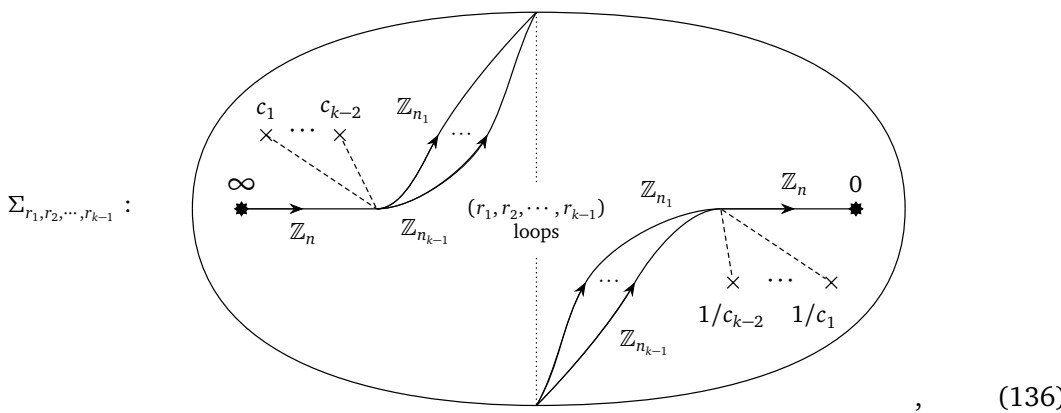

$$, \tag{136}$$

where dashed lines carry (in general, different) $\mathbb{Z}_2$ monodromies whose associated branch cuts combine with the $\mathbb{Z}_n$ branch line and split it into $\mathbb{Z}_{n_i}$ branch lines with each $\mathbb{Z}_{n_i}$ acting on mutually different $n_i$ number of sheets. The $\mathbb{Z}_{n_i}$ branch line wraps the sphere $r_i$ number of times along the equator. The branch points at $c_i$ and $1/c_i$ carry the same $\mathbb{Z}_2$ monodromy.

Let us study the cycles on this $\mathcal{N}=1$ curve. If we start at a point on equator and along a sheet acted upon by the $\mathbb{Z}_{n_i}$ line, then traversing the equator $n_i$ times brings us back to the

same sheet. Thus $n_i W$ exhibits perimeter law for each $i$. Moreover, starting at $t = \infty$ along a sheet acted upon by $\mathbb{Z}_{n_i}$ and running along the $\mathbb{Z}_n$ branch line and $\mathbb{Z}_{n_i}$ branch line to reach $t = 0$, we obtain the cycle $H + r_i W$, which implies that the associated line exhibits perimeter law. The full set of line operators exhibiting perimeter law can be written as

$$\mathcal{I}_{r_1, r_2, \cdots, r_{k-1}} = [H + r_1 W, \gcd(n_1, n_2, \cdots, n_{k-1}, r_1 - r_2, r_2 - r_3, \cdots, r_{k-2} - r_{k-1}) W], \quad (137)$$

matching the field theory expectation (110).

# 6 Confinement in Non-Lagrangian Theories

Because of the lack of available tools, confinement is typically studied only in Lagrangian theories. The main advantage of the method outlined in this paper is that it allows to study confinement for $\mathcal{N} = 1$ deformations of Class S theories, which are typically non-Lagrangian. Thus, in this section, we discuss a class of non-Lagrangian theories and apply our method to show that they contain vacua exhibiting confinement.

We discuss an infinite class of such theories. The simplest theory in this class is related to the famous $E_6$ Minahan-Nemeschansky theory. More precisely, it is an $\mathcal{N} = 1$ deformation of the asymptotically conformal 4d $\mathcal{N} = 2$ theory obtained by gauging an $\mathfrak{su}(3)^3$ subalgebra of the $\mathfrak{e}_6$ flavor algebra of the $E_6$ Minahan-Nemeschansky SCFT. Other examples in the class are $\mathcal{N} = 1$ deformations of the asymptotically conformal 4d $\mathcal{N} = 2$ theory obtained by gauging $\mathfrak{su}(n)^n$ flavor symmetry of the 4d $\mathcal{N} = 2$ SCFT obtained by gluing $n-2$ copies of $T_n$ trinions, or in other words, the 4d $\mathcal{N} = 2$ SCFT obtained by compactifying $A_{n-1}$ $(2,0)$ theory on a sphere with $n$ maximal regular untwisted punctures.

The first few subsections are devoted to the $n = 3$ example, while the last subsection discusses briefly the generalization to general $n$.

## 6.1 The 4d $\mathcal{N} = 2$ Setup: Sphere with three $\mathcal{P}_0$ Punctures

Consider compactification of 6d $\mathcal{N} = (2,0)$ theory of type $A_{n-1}$ on a sphere with three irregular punctures of type $\mathcal{P}_0$. We will refer to these theories by $\mathfrak{P}_{n,3}$, and more generally we define

$$\mathfrak{P}_{n,\alpha} = \text{6d } (2,0) \ A_{n-1} \text{ theory on a sphere with } \alpha \text{ irregular } \mathcal{P}_0 \text{ punctures}. \quad (138)$$

Though our interest is in the case $n = 3$, we keep $n$ general in this subsection. This constructs a 4d $\mathcal{N} = 2$ theory described by a quiver of the form

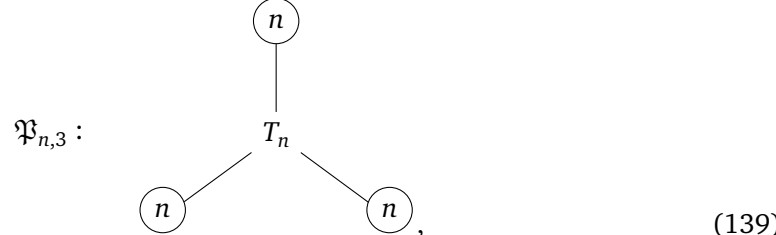

(139)

i.e. the $T_n$ theory with its $\mathfrak{su}(n)^3$ flavor symmetry gauged and no additional matter. For $n = 3$, the $T_3$ theory is the same as the $E_6$ Minahan-Nemeschansky theory.

Placing the punctures at $t = 0, 1, \infty$ the $v$-curve takes the form

$$\mathcal{P}(v) = v^n - \sum_{k=1}^{n-2} \frac{P_{n-k}(t) v^k}{(t-1)^{n-k}} - \frac{P_{n+3}(t)}{t(t-1)^{n+1}} = 0, \quad (140)$$

where the polynomials $P_k(t)$ are polynomial of degree $k$. These contain

$$3 + 3(n-1) + \frac{(n-2)(n-1)}{2},\tag{141}$$

complex parameters with $3(n-1)$ of them being the CB parameters associated to the three $\mathfrak{su}(n)$ gauge algebras, $(n-2)(n-1)/2$ of them being the CB parameters associated to the $T_n$ theory, and 3 of them being mass parameters associated to the strong-coupling scales of the three $\mathfrak{su}(n)$ gauge algebras which can be identified via the asymptotics

$$
\begin{aligned}
t \to 0 : \quad & \phi_\zeta \sim \Lambda_0 \, t^{-1/n} \mathrm{diag}\left(1, \omega, \omega^2, \cdots, \omega^{n-1}\right) + \cdots, \\
t \to 1 : \quad & \phi_\zeta \sim \Lambda_1 (t-1)^{-(n+1)/n} \mathrm{diag}\left(1, \omega, \omega^2, \cdots, \omega^{n-1}\right) + \cdots, \\
t \to \infty : \quad & \phi_\zeta \sim \Lambda_\infty \, t^{1/n} \mathrm{diag}\left(1, \omega, \omega^2, \cdots, \omega^{n-1}\right) + \cdots.
\end{aligned}
\tag{142}
$$

From this we learn

$$\Lambda_0^n = (-1)^{n+1} P_{n+3}(0), \qquad \Lambda_1^n = (-1)^{n+1} P_{n+3}(1), \qquad \Lambda_\infty^n = \frac{P_{n+3}(t)}{t^{n+3}}\bigg|_{t=\infty}.\tag{143}$$

Let us now turn to the study of the defect group of line operators in this 4d $\mathcal{N} = 2$ theory. Inserting the surface defect $f \in \widehat{Z} \simeq \mathbb{Z}_n$ along the cycles encircling the three punctures gives rise to three line operators $W_i$, and inserting $f$ along cycles stretching between the three punctures gives rise to line operators $H_{ij}$, as shown below

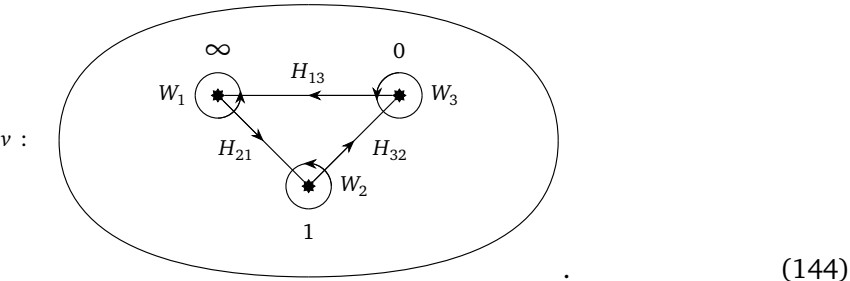

$$\tag{144}$$

$W_i$ can be identified with the Wilson line associated to $\mathfrak{su}(n)_i$ and $H_{ij}$ can be identifed with 't Hooft line $H_i - H_j$ where $H_i$ is the 't Hooft line associated to $\mathfrak{su}(n)_i$. These lines are not independent but sum to zero

$$W_1 + W_2 + W_3 = 0, \qquad H_{21} + H_{32} + H_{13} = 0.\tag{145}$$

Thus the defect group is

$$\mathcal{L} = \left\{W_i, H_{jk}\right\} / \left\{W_1 + W_2 + W_3 = 0, H_{21} + H_{32} + H_{13} = 0\right\}.\tag{146}$$

The pairing on these line operators is

$$\langle W_i, H_{ij}\rangle = 1/n, \qquad \langle W_j, H_{ij}\rangle = -1/n,\tag{147}$$

and zero otherwise. One can choose various polarizations, but we will be particularly concerned with the purely electric polarization $\Lambda = \{W_i\} / \{W_1 + W_2 + W_3 = 0\}$ which corresponds to choosing the global form of the gauge symmetry groups to be $SU(n)_i$ for each $i$. The 1-form symmetry group for this polarization is $\widehat{\Lambda} \simeq \mathbb{Z}_n \times \mathbb{Z}_n$.

## 6.2 Rotating to $\mathcal{N} = 1$: $\mathfrak{P}_{3,3}$

We now deform the $n = 3$ version of the above discussed $\mathcal{N} = 2$ theory to $\mathcal{N} = 1$, which is the theory $\mathfrak{P}_{3,3}$ defined in (138). We do this by rotating the two punctures at $t = 0, \infty$. The specific form of the rotation is discussed later. This rotation is only possible at certain points in the CB of $\mathcal{N} = 2$ vacua, whose branch cut structure we first discuss. At a generic point in the CB, the $v$-curve (148) has 12 branch points of $\mathbb{Z}_2$ monodromy. Thus, we can represent the branch cut structure as

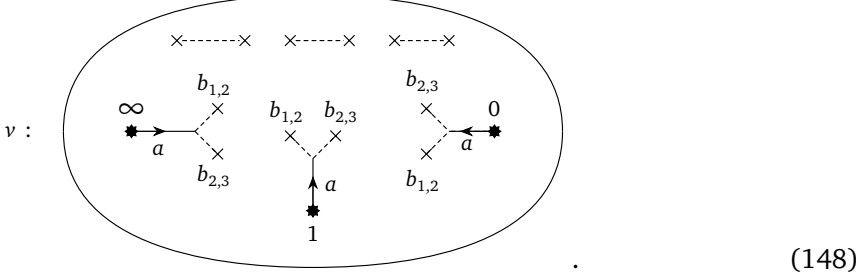

. (148)

The CB vacua that survive are obtained by colliding the two sets of 6 branch points together at $t = t_1, t_2$ with the $v$-curve being The $v$-curve after taking the above limit is written as

$$v^3 - \Lambda^3 \frac{(t - t_1)^3 (t - t_2)^3}{t(t-1)^4} = 0. \tag{149}$$

From this we see that all the CB parameters have been fixed, and the scale $\Lambda$ and the locations $t_1, t_2$ of collided branch points are determined in terms of $\Lambda_i$ computed in (143). There is no monodromy as one encircles the two points $t_1$ and $t_2$, while the three punctures still have a $\mathbb{Z}_3$ monodromy. Thus the branch cut structure has to be

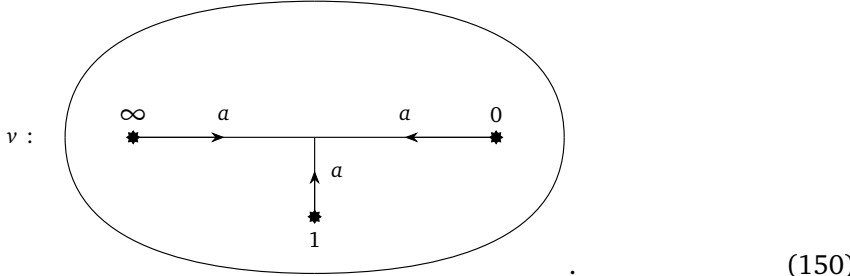

. (150)

The rotation that we perform is specified by the boundary conditions

$$
\begin{aligned}
t \to 0: & \quad \varphi \to \mu_0 \phi_\zeta, \\
t \to \infty: & \quad \varphi \to \mu_\infty \phi_\zeta.
\end{aligned}
\tag{151}
$$

The generic $w$-curve compatible with these asymptotics is

$$w^3 - \sum_{k=0}^{1} c_{3-k} w^k - \frac{d_0}{t} - d_\infty t = 0, \tag{152}$$

for some complex constants $c_2, c_3, d_0, d_\infty$. The associated branch cut structure can be displayed as

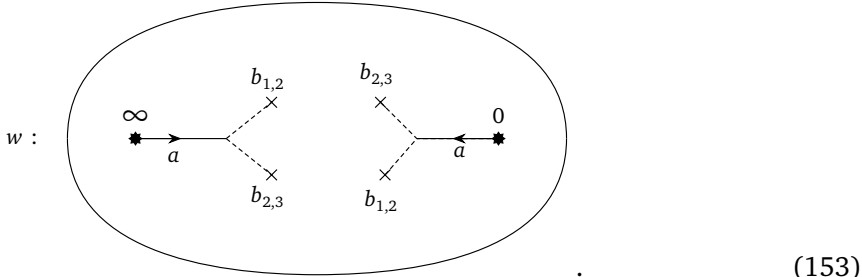

. (153)

The $w$-curve (189) has the symmetry

$$t \leftrightarrow d_0/t d_\infty \,, \tag{154}$$

which pairs branch points of identical $\mathbb{Z}_2$ monodromy. To match the $\mathbb{Z}_3$ monodromy of the $v$-curve about $t = 1$ we need to collide 2 branch points of the $w$-curve with $\mathbb{Z}_2$ monodromy at $t = 1$. Necessarily the remaining 2 branch points collide at $t = d_0/d_\infty \equiv d$. The point $t = 1$ is a branch point and necessarily fully ramified, i.e. all sheets must come together at $w = 0$. Thus we require that substitution of $t = 1$ into the $w$-curve (189) results in the equation $w^3 = 0$. This implies $c_3 = d_0 + d_\infty$ with $c_2$ vanishing. Further we have

$$\begin{aligned} , t \to 0: \quad & w^3 \to \mu_0^3 v^3 = \mu_0^3 \Lambda_0^3/t + \dots \\ t \to \infty: \quad & w^3 \to \mu_\infty^3 v^3 = \mu_\infty^3 \Lambda_\infty^3 t + \dots \,, \end{aligned} \tag{155}$$

from which $d_0 = \mu_0^3 \Lambda_0^3$ and $d_\infty = \mu_\infty^3 \Lambda_\infty^3$ follow. The $w$-curves potentially matched by the $v$-curve simplify to

$$w^3 - \mu_0^3 \Lambda_0^3 \left(1 - \frac{1}{t}\right) - \mu_\infty^3 \Lambda_\infty^3 (1 - t) = 0 \,. \tag{156}$$

The structure of this curve is determined by two disjoint branch cuts of $\mathbb{Z}_3$ monodromy and takes the form

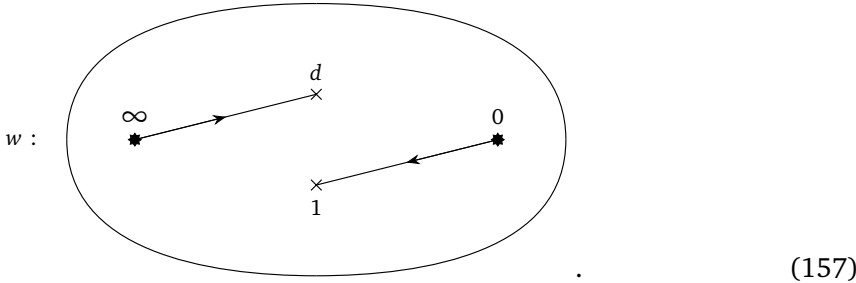

$$\tag{157}$$

We further take the limit $d = 1$, which constrains the two parameters $\mu_0, \mu_\infty$ in terms of each other (and $\Lambda_i$) and the final $w$-curve is then

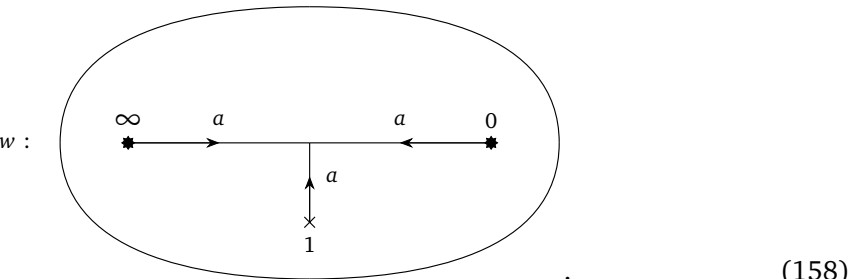

$$\tag{158}$$

In equations, the $w$-curve becomes

$$w^3 - \mu^3 \Lambda^3 \left(t + \frac{1}{t} - 2\right) = 0 \,, \tag{159}$$

where $\mu^3 = -\mu_\infty^3$.

The $\mathcal{N} = 1$ curve is specified by the diagonal combination of (150) and (158). The cycles on the $\mathcal{N} = 1$ curves project to $H_{ij}$ and $3W_i$. Thus $H_{ij}$ show perimeter law while $W_i$ show area law. Thus, in the electric polarization $\Lambda$, the preserved 1-form symmetry group in this vacuum is

$$\mathcal{O} = \widehat{\Lambda} \simeq \mathbb{Z}_3 \times \mathbb{Z}_3 \,. \tag{160}$$

That is, akin to the case of a vacuum of $\mathcal{N} = 1$ SYM in electric polarization, no element of the 1-form symmetry group is broken in this vacuum.

### 6.3 Generalizations: $\mathfrak{P}_{n,n}$

The above discussed example falls into an infinite class of theories which contain similar confining vacua. The $\mathcal{N} = 2$ theory $\mathfrak{P}_{n,\alpha=n}$, defined in (138), that we begin with is obtained by compactifying $A_{n-1}$ (2,0) theory on a sphere with $n$ irregular punctures of type $\mathcal{P}_0$. Notice that the number of punctures is correlated to the type of (2,0) theory. This $\mathcal{N} = 2$ theory $\mathfrak{P}_{n,n}$ can be understood as being obtained by gauging the $\mathfrak{su}(n)^n$ flavor symmetry algebra of the 4d $\mathcal{N} = 2$ SCFT $\mathfrak{S}_n$ obtained by compactifying $A_{n-1}$ (2,0) theory on a sphere with $n$ maximal regular punctures. Each $\mathfrak{su}(n)$ factor in the resulting $\mathfrak{su}(n)^n$ gauge algebra has non-vanishing beta function such that the associated gauge couplings asymptote to zero in the UV. Thus, the 4d $\mathcal{N} = 2$ theory $\mathfrak{P}_{n,n}$ under consideration asymptotes in the UV to the 4d $\mathcal{N} = 2$ SCFT $\mathfrak{S}_n$ described above.

If we label the punctures by $i \in \{1, 2, \cdots, n\}$, then the un-screened line operators can be written as $W_i, H_{ij}$, where $W_i$ arises by wrapping the surface defect $f$ along a loop encircling the $i$-th puncture, and $H_{ij}$ arises by wrapping $f$ along a 1-cycle going from puncture $j$ to puncture $i$. We can write the defect group as

$$\mathcal{L} = \mathcal{L}_W \times \mathcal{L}_H, \tag{161}$$

such that

$$\mathcal{L}_W = \{W_i\} / \left\{ \sum W_i = 0 \right\} \simeq \mathbb{Z}_n^{n-1} \tag{162}$$

and

$$\mathcal{L}_W = \left\{ H_{i+1,i}, H_{1n} \right\} / \left\{ \sum H_{i+1,i} + H_{1n} = 0 \right\} \simeq \mathbb{Z}_n^{n-1}. \tag{163}$$

The non-trivial pairings are

$$\langle W_i, H_{ij} \rangle = 1/n, \qquad \langle W_j, H_{ij} \rangle = -1/n. \tag{164}$$

We call the polarization $\Lambda = \mathcal{L}_W$ as the electric polarization.

Let the punctures be located at $t = 0, \infty, 1, p_1, p_2, \cdots, p_{n-3}$. We rotate all punctures except the one located at $t = 1$ such that the asymptotics of $\varphi$ are

$$
\begin{aligned}
t \to 0: &\quad \varphi \to \mu_0 \phi_\zeta, \\
t \to \infty: &\quad \varphi \to \mu_\infty \phi_\zeta, \\
t \to p_i: &\quad \varphi \to \mu_i \phi_\zeta.
\end{aligned}
\tag{165}
$$

We take specific limits of the $\nu$ and $w$ curves such that they become

$$
\begin{aligned}
\nu^n - \frac{\Lambda^n (t - t_{n-2}^\nu)^n (t - t_{n-1}^\nu)^n}{t(t-1)^{n+1}} \prod_{i=1}^{n-3} \frac{(t - t_i^\nu)^n}{(t - p_i)^{n+1}} = 0, \\
w^n - \frac{d^n (t-1)^{n-1}}{t} \prod_{i=1}^{n-3} \frac{(t - t_i^w)^n}{(t - p_i)^{n+1}} = 0,
\end{aligned}
\tag{166}
$$

where $\Lambda, t_i^\nu$ capture the strong coupling scales associated to $\mathfrak{su}(n)^n$ gauge algebra and $d, t_i^w$ capture the rotation parameters $\mu_i$. Since we have $n-1$ number of $\mu_i$ but only $n-2$ number of parameters $d, t_i^w$, there is a constraint the rotation parameters have to satisfy for the vacuum described by the above solution to exist. From the above expressions, we see that there is no monodromy around $t_i^\nu$ for the $\nu$-curve, and there is no monodromy around $t_i^w$ for the $w$-curve. On the other hand, there is a $\mathbb{Z}_n$ monodromy around $p_i, 1, 0, \infty$ for both $\nu$ and $w$ curves. Thus,

we can write the branch structure of the $\mathcal{N} = 1$ curve $\Sigma$ as

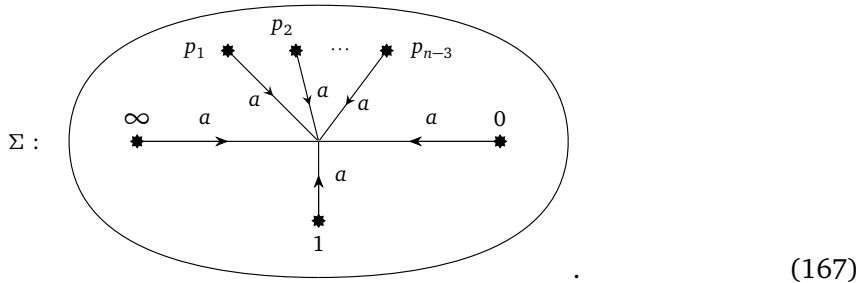

. (167)

From this curve we see that $H_{ij}$ show perimeter law, while $W_i$ show area law. It should be noted though that some combinations of $W_i$ also show perimeter law. For example, pick a positive integer $k$ that divides $n$. Choose any set $S_k$ comprising of $k$ punctures. Then the line operator

$$\frac{n}{k} \sum_{i \in S_k} W_i \,, \tag{168}$$

shows perimeter law. In any case, since $W_i$ show area law, this vacuum exhibits non-trivial confinement for the electric polarization $\Lambda = \mathcal{L}_W$. The preserved 1-form symmetry group depends on the divisors of $n$. If $n$ is prime, then the preserved 1-form symmetry group is easily computed to be

$$\mathcal{O} = \widehat{\Lambda} \simeq \mathbb{Z}_n^{n-1} \,. \tag{169}$$

Clearly this class of non-Lagrangian confining theories deserve further investigation. In a similar way, using the methods described in this paper, one can construct a large number of other classes of examples of $\mathcal{N} = 1$ theories that contain confining vacua.

# 7 Conclusion and Future Directions

In this paper, we studied confinement in 4d $\mathcal{N} = 1$ theories that can be obtained by deforming 4d $\mathcal{N} = 2$ theories of Class S. The line operators of the theory are encoded in 1-cycles on the Gaiotto curve used to compactify the 6d $(2,0)$ theory. The line operators exhibiting perimeter law in a vacuum $r$ are deduced as the subgroup of 1-cycles that arise as projections of 1-cycles living on the $\mathcal{N} = 1$ curve $\Sigma_r$ associated to the vacuum $r$, under the projection map provided by the fact that the $\mathcal{N} = 1$ curve is an $N$-fold cover of the Gaiotto curve. In this way, the 1-cycles on the $\mathcal{N} = 1$ curve $\Sigma_r$ encode confinement in the vacuum $r$. One point to be noted is that, in this paper, we have only considered situations in which the 4d $\mathcal{N} = 2$ Class S theory (before deformation) arises by an untwisted compactification of 6d $(2,0)$ theory of type $A_{n-1}$. The generalization to $(2,0)$ theories of other types and the inclusion of twists is an interesting future direction.

The $\mathcal{N} = 1$ curve is obtained as the spectral curve associated to a generalized Hitchin system comprising of two Higgs fields $\phi, \varphi$. $\phi$ is the standard $\mathcal{N} = 2$ Higgs field associated to the parent $\mathcal{N} = 2$ Class S theory. On the other hand, the asymptotic behavior of the other Higgs field $\varphi$ at the locations of punctures encodes the deformation parameters used to deform the parent $\mathcal{N} = 2$ theory to $\mathcal{N} = 1$ theory. However, the dictionary between the two has not been made explicit in this paper, and would be interesting to explore futhre. In this context, it would be interesting to understand the implications of turning on asymptotics of $\varphi$ that correspond to turning on irrelevant deformation parameters in the $\mathcal{N} = 2$ theory (see the discussion in section 3.3.3).

Another interesting direction to pursue following from this paper is to study 't Hooft anomalies of the global symmetries of these theories, in particular the 0- and 1-form symmetries. E.g. the SYM theories have a mixed anomaly between the chiral symmetry and 1-form symmetry, which in [56] was determined for $\mathcal{N} = 1$ SYM theory starting with the Little String anomaly polynomial. It would be very interesting to derive the anomalies, and the IR TQFTs describing various vacua, starting with the 6d $(2, 0)$ theory, and generalizing this to other theories that have a realization in terms of the class S inspired approach we have taken in this paper.

The class S theories can have in certain instances a description in terms of Type IIB compactified on a singular Calabi-Yau three-fold, constructed as an ALE-fibration over $\mathcal{C}$. The fibration structure is encoded in the $\mathcal{N} = 2$ Higgs field and it can be shown that the one-form symmetry from the class S description maps nicely to the IIB picture, where the 1-form symmetry is encoded in the boundary data of the Calabi-Yau [29]. It would be very interesting to determine the $\mathcal{N} = 1$ analog, i.e. how the pair of Higgs fields $\phi$ and $\varphi$ of the generalized Hitchin system, encode data of a IIB or F-theory background (one realization of 4d $\mathcal{N} = 1$ theories is in terms of elliptic Calabi-Yau four-folds).

The most exciting direction that this paper opens up is the study of confinement in non-Lagrangian theories. This is exemplified in section 6, where we provide construction of a family of non-Lagrangian $\mathcal{N} = 1$ theories that contain confining vacua. These confinement properties can be read off from the associated $\mathcal{N} = 1$ curves and the spectrum of line operators of the theory, that we also determine. In general, we hope that this work would act as a stepping stone to a more general study of phases of $\mathcal{N} = 1$ theories and possibly uncover interesting physical phenomena in these theories.

# Acknowledgements

We thank Fabio Apruzzi, Davide Gaiotto and Simone Giacomelli for discussions. This work is supported by ERC grants 682608 (LB and SSN) and 787185 (LB). SSN also acknowledges support through the Simons Foundation Collaboration on "Special Holonomy in Geometry, Analysis, and Physics", Award ID: 724073, Schafer-Nameki.

# A   Summary of Notation

- Gaiotto Curve: $\mathcal{C}$

- Higgs fields: $\phi, \varphi$

- Relative vector field: $\zeta$

- $\mathcal{N} = 1$ Curve: $\Sigma, \Sigma_r$

- $v$-curve: $\det(v - \phi_\zeta) = 0$

- $w$-curve: $\det(w - \varphi) = 0$

**Summary of notation in figures of $v$- and $w$-curves.**

- The picture shows the Gaiotto curve parametrized by $t$ and the branch cuts for $v, w$ as $n$-fold cover over it.

- Punctures and branch points are denoted by stars and crosses, respectively.

- Branch lines are labelled by a monodromy element in the symmetric group $S_n$.

- Cyclic permutation $(123\cdots n) \to (234\cdots n1)$ is denoted by $a$.

- Transposition of $ij$ is denoted by $b_{ij}$.

- Dashed lines are associated with $\mathbb{Z}_2$ branch cuts and solid oriented lines are associated with $\mathbb{Z}_{n>2}$ branch cuts.

- Branch lines are oriented such that the labelled monodromy takes action along a closed path $\gamma(\tau)$ when the cross product of the oriented line with the vector $\dot{\gamma}$ points out of the page.

- The circle $|t| = 1$ is located at the vertical equator with $t = 1$ on the front and $t = -1$ on the back.

## B  Rotation at Argyres-Douglas Points

In this appendix we discuss the rotation of 4d $\mathcal{N} = 2$ SYM theories with gauge algebra $\mathfrak{su}(n)$ by the superpotential $W(\phi) = (g/n)\phi^n$. This superpotential is a degenerate case of the higher order superpotentials for the CSW case discussed in section 5.3, as all critical points now coincide (at the origin). This case is of some interest because it provides an example showing the difference between considering a generic and a non-generic superpotential. Moreover, the $\mathcal{N} = 2$ CB vacua which are left unlifted by this deformation are the two Argyres-Douglas points, and so this case might have some interesting connections to $\mathcal{N} = 1$ deformations of the corresponding $\mathcal{N} = 2$ Argyres-Douglas theories.

We follow the steps outlined in section 3.4.2 to determine the $\mathcal{N} = 1$ curve. The topological manipulations are of the type already encountered in the CSW curves (194) and in the novel trinion-like curves (150) and (158) and are therefore worth isolated attention.

At the Argyres-Douglas points $n - 1$ mutually non-local dyons become massless. In the $\mathcal{N} = 2$ CB these points are characterized by the SW curve

$$v^n - \Lambda^n \left( t + \frac{1}{t} \pm 2 \right) = 0. \tag{170}$$

The discriminant $\Delta = \Lambda^{n(n-1)}(t \pm 1)^{2n-2}/t^{n-1}$ shows that $2n - 2$ branch points of $\mathbb{Z}_2$ monodromy have been collided at $t = \mp 1$. This collision is understood starting from the general $v$-curve:

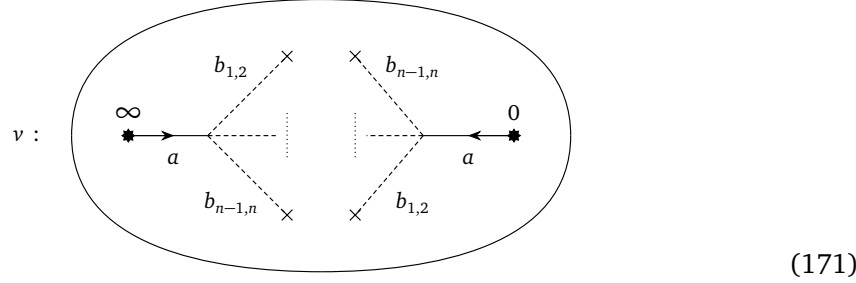

$$\tag{171}$$

First collide all branch points at $t = d, 1/d$ which now terminate $\mathbb{Z}_n$ branch cuts

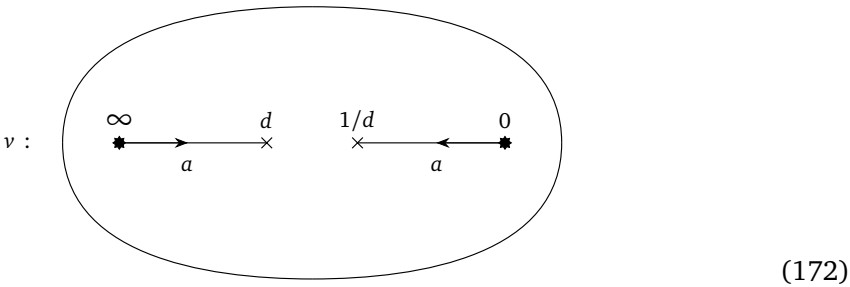

$$(172)$$

and subsequently we move both branch points to $t = \pm 1$. The branch cuts are oriented such that the $\mathbb{Z}_n$ monodromies add and we are left with a single non-singular branch point:

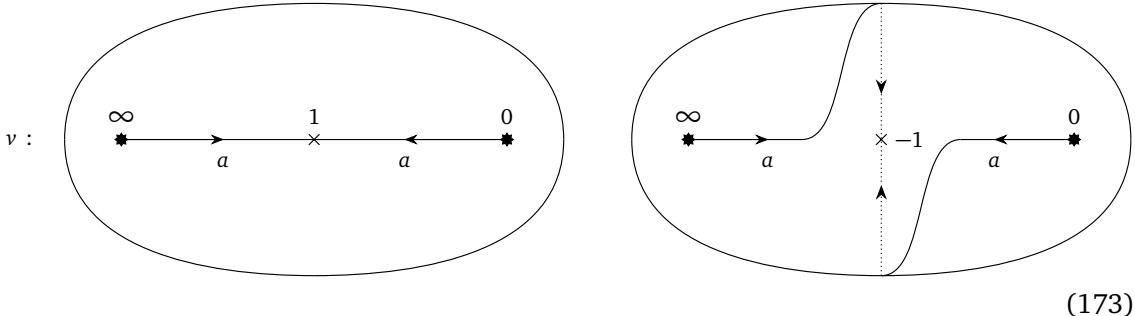

$$(173)$$

Solving (170) we find $v = \Lambda(t \pm 1)^{2/n} t^{-1/n}$ which matches the final picture in (173) having $\mathbb{Z}_n$ monodromy about $t = 0, \infty$ and twice the monodromy about $t = \mp 1$.

The $w$-curve follows from the rotation of the puncture $t = \infty$ via boundary conditions $\varphi \to g \phi_\zeta^{n-1}$ as $t \to \infty$. The Higgs field $\varphi$ is required to be regular elsewhere. This implies for the eigenvalues $|w| \sim |v|^{n-1} \sim |t|^{\frac{n-1}{n}}$ when approaching the rotated puncture at infinity. This bounds the growth of each summand in the spectral curve of $\varphi$ by $t^{n-1}$. With this we find the $w$-curve constrained to

$$w^n + p_1(t)w^{n-2} + p_2(t)w^{n-3} + \cdots + p_{n-1}(t) = 0, \qquad (174)$$

for some polynomials $p_k(t)$ of degree $k$ in $t$. This generic $w$-curve, prior to monodromy matching, has following branch cut structure

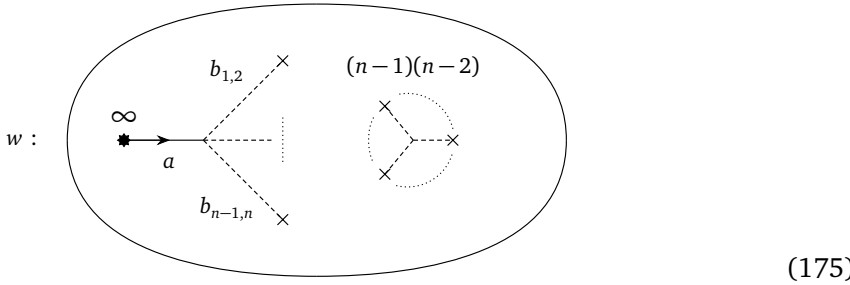

$$(175)$$

where $(n-1)(n-2)$ branch points are not connected to the puncture at $t = \infty$. We require $t = 0$ to be a branch point with $\mathbb{Z}_n$ monodromy in order to match the monodromy about the $\mathcal{P}_0$ puncture on the $v$-curve. This requires all sheets to ramify and at $t = 0$ the $w$-curve must reduce to $w^n = 0$. The constant terms in all polynomials $p_k(t)$ must therefore vanish. The terms of maximal growth are $w^n$ and the term in $p_{n-1}$ of degree $n-1$, the latter is therefore fixed by the boundary condition, we have $p_{n-1} = -g^n \Lambda^{n(n-1)} t^{n-1} + \mathcal{O}(t^{n-2})$. Next we match the branch points of the $v$-curve punctures. We require a branch point with an incoming $a^2$ monodromy at $t = \mp 1$ and again total ramification. The curve must take the form

$w^n - g^n \Lambda^{n(n-1)} t (t \pm 1)^{n-2} (\ldots) = 0$. The polynomials $p_k(t)$ vanish for $k < n-2$. The degree of $t(t \pm 1)^{n-2}$ is already $n-1$ and therefore the dotted term is simply a factor of 1. These manipulations correspond to colliding $n-1$ and $(n-1)(n-2)$ of the branch points displayed in (175) at $t = 0, \mp 1$ respectively.

In summary the $\mathcal{N} = 1$ curve is derived to be the diagonal curve of the following $v$- and $w$-curves

$$v^n = \Lambda^n \left( t + \frac{1}{t} \pm 2 \right), \qquad w^n = g^n \Lambda^{n(n-1)} t (t \pm 1)^{n-2}. \tag{176}$$

There is a unique $v$-curve for a given discriminant $\Delta = \Lambda^{n(n-1)} (t \pm 1)^{2n-2} / t^{n-1}$, i.e we have no Dehn twists around the punctures $t = 0, \infty$. Topologically this follows from the branch cuts running between punctures and branch points (rather than two punctures as for SYM). In (172) we have a choice of wrapping the branch cuts arbitrarily many times around the equator before colliding them at $t = d = \pm 1$. This results in a spiral centred at $t = \pm 1$ which can be unwound at the branch point. The sheets of the curve are paired by locally pairing an arbitrary pair $(v, w)$ and then extending this pairing over all sheets via the $\mathbb{Z}_n$ monodromy action. We can redefine $v, w$ by $n$-th roots of unity so any such pairing is equivalent. Thus, we find only two $\mathcal{N} = 1$ curves.

## C  CSW Examples

In this appendix we return to section 5 and consider the cases $\mathfrak{g} = \mathfrak{su}(3), \mathfrak{su}(4)$ rotated by cubic superpotentials in detail.

The example $\mathfrak{g} = \mathfrak{su}(3)$ shows clearly how partial Dehn twists arise as one moves between two degenerate SW curves in the CB which in turn are perturbed to two distinct $\mathcal{N} = 1$ vacua. Further, it serves as the simplest example for which we demonstrate how to match monodromies when pairing $v$- and $w$-curves, see appendix D for details. As $n = 3$ is prime there is no confinement and this example is only of technical interest.

The example $\mathfrak{g} = \mathfrak{su}(4)$ is physically more exciting. Here we find vacua where a broken R-symmetry is restored. This greatly simplifies the derivation of the $w$-curve which can now be derived from the $v$-curve by a quotient. Additionally, $\mathfrak{g} = \mathfrak{su}(4)$ is the lowest rank example at which the duality between two different gauge theories discussed in [21] occurs. We interpret this duality topologically and discuss further how the family of $\mathcal{N} = 1$ curves project onto the $\mathcal{N} = 2$ CB.

### C.1  Dehn Twists and Monodromies for $\mathfrak{g} = \mathfrak{su}(3)$

In this subsection, we illustrate the case of $n = 3$ in detail. For $n_1 = 2, n_2 = 1$, the $v$-curves are given by

$$0 = \left( v + \frac{\mu}{3g} \right)^2 \left( v - \frac{2\mu}{3g} \right) - \frac{\Lambda^3 (t \mp 1)^2}{t}, \tag{177}$$

while the $w$-curves are given by

$$0 = w^3 + \mu g \Lambda^3 t w - g^3 \Lambda^6 t (t \mp 1). \tag{178}$$

This corresponds to breaking $\mathfrak{su}(3) \to \mathfrak{su}(2) \oplus \mathfrak{u}(1)$. The two $\mathcal{N} = 1$ curves are associated to vacua arising from the $\mathfrak{su}(2)$ factor, while the fact that both curves are genus one reflects the presence of a low-energy $\mathfrak{u}(1)$ gauge field which corresponds to the $\mathfrak{u}(1)$ factor in the above breaking.

We begin by deriving the $v$-curve given in (177). The SW curve of the $\mathcal{N} = 2$ theory takes the form

$$\mathcal{P}(v) = v^3 + u_2 v + u_3 - \Lambda^3 \left( t + \frac{1}{t} \right) = 0.$$  (179)

The CB is complex two-dimensional, the SW curve has genus two. Along a codimension one subspace the curve degenerates to genus one and a single monopole becomes massless. Prior to this degeneration the $v$-curve has following branch cut structure:

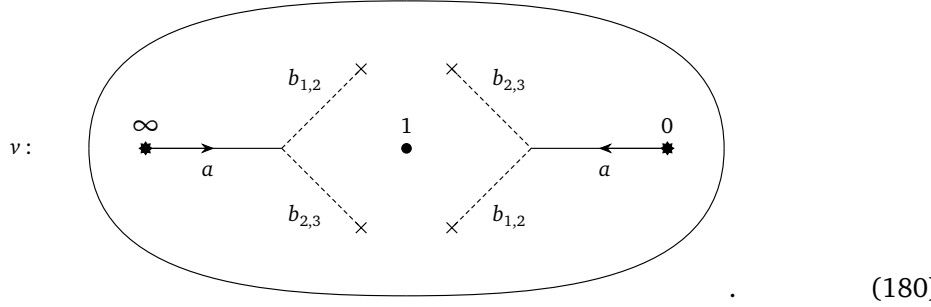

.  (180)

Pairs of branch points with identical monodromy are related via $t \to 1/t$ and can therefore only collide at $t = \pm 1$. Such degenerations result in the discriminant

$$\Delta(\mathcal{P}, v) = (t \mp 1)^2 p_2(t) \frac{1}{t^2},$$  (181)

where $p_2(t) = -27\Lambda^6(t-c)(t-1/c)$ is a polynomial of degree two in $t$ with zeros at $t = c, 1/c$. These values are the location of the unpaired $\mathbb{Z}_2$-valued branch points of the SW curve. There are seemingly four distinct ways to achieve this degeneration. We can collide either pair of branch points at $t = \pm 1$. This gives the four branch cut structures:

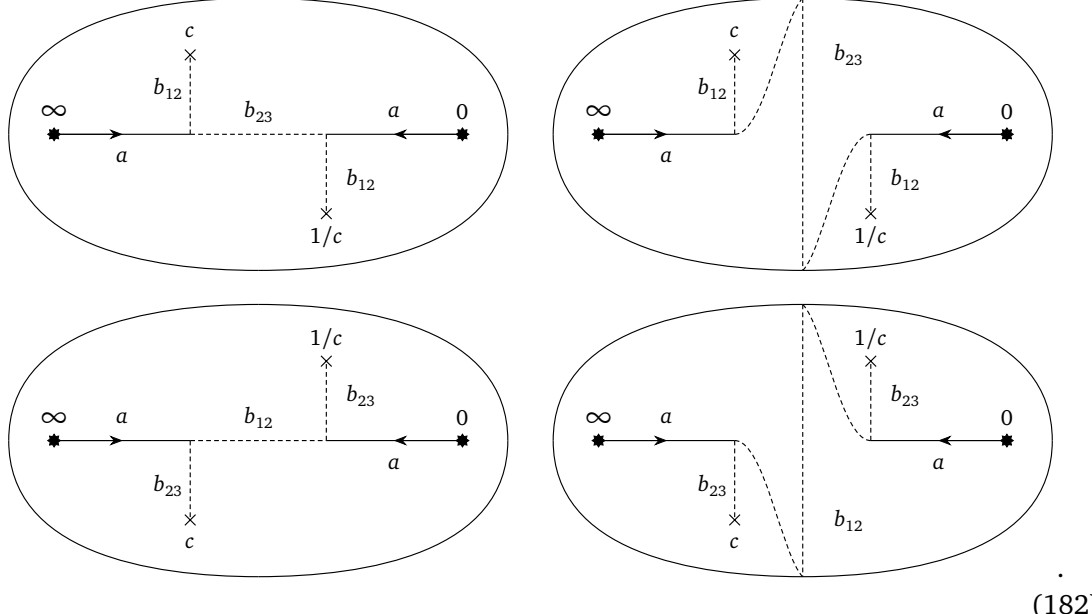

.  (182)

Here the central branch cuts in the right hand figures wrap around the back and equator of the Gaiotto curve $C \cong S^2$. Rows are related by a Dehn twist, columns describe identical configurations. For example consider the top left configurations and wrap the $b_{23}$ branch line around $c, 1/c$. Upon renumbering the sheets the bottom left configuration emerges. The relative Dehn twist is seen by say considering the degeneration for which branch points collide at $t = +1$, as shown in the left hand figures. Resolving this collision and moving the two branch

points to $t = -1$ along the equator $|t| = 1$ the symmetry $t \to 1/t$ moves the branch points through the upper and lower half-plane. The configurations therefore differ by a $\mathbb{Z}_2$ branch line along the equator. This configuration is topologically equivalent to the right hand pictures above. As representatives with distinct branch cut structure we consider the top row going forward. These result from colliding the brach points associated with $b_{23}$ at $t = \pm 1$.

We give the $v$-curves with these two branch cut configurations. The discriminant ansatz (181) determines the coefficients of the SW curve and fixes these to

$$0 = (v - \alpha)^2(v + 2\alpha) - \Lambda^3(t + 1/t \mp 2), \tag{183}$$

where $\alpha^3 = \Lambda^3(c \mp 1)^2/4c$. We therefore find two families of SW curves, parametrized by $\alpha$ and distinguished by whether pairs of branch points are collided at $t = \pm 1$ starting from (180). The branch cut structure of the curves (183) are shown on the left and right of (182) respectively. In the limit of vanishing scale $\Lambda \to 0$ the SW curve degenerates to a configuration with a double zero at $v = \alpha$ and a single zero at $v = -2\alpha$. This corresponds to classical vacua with two eigenvalues at the former and a single eigenvalue at the latter location. The remnant gauge symmetry is therefore $\mathfrak{su}(2) \oplus \mathfrak{u}(1)$.

Let us rotate the set-up by turning on the superpotential (107). The family of $\mathcal{N} = 2$ vacua described by the curves (183) are lifted up to points. These are determined in the limit $\Lambda \to 0$ in which the critical points of the superpotential determine the location of the eigenvalues, together with a shift of $v$ to the center of mass frame. The superpotential has two critical points at $v = 0, -\mu/g$. We therefore have two possible configurations given by two eigenvalues at $v = 0$ and one at $v = -\mu/g$ or vice versa. In the center of mass frame these configurations are given by $\alpha = \pm \mu/3g$. These are related by a $\mathbb{Z}_2$ R-symmetry which is spontaneously broken in this example. and the $\mathbb{Z}_2$ R-symmetry maps $\alpha \leftrightarrow -\alpha$. Without loss of generality we therefore consider the case $\alpha = -\mu/3g$ going forward, this gives (177).

The superpotential gives the boundary conditions (113) and the ansatz (117) becomes

$$0 = \mathcal{Q}(w) = w^3 - 3g^2\Lambda^3\beta tw - g^3\Lambda^6 t^2 - g^3\Lambda^6\gamma t. \tag{184}$$

The parameters $\beta, \gamma$ are now fixed by matching with the branch points of the $v$-curve. The discriminant of the $w$-curve takes the form

$$\Delta(\mathcal{Q}, w) = -27g^6\Lambda^{12}t^2\left(t^2 + (2\gamma - 4\beta^3)t + \gamma^2\right), \tag{185}$$

which we require to have zeros at $t = c, 1/c$. This gives $\gamma = \mp 1$ and $\beta^3 = (c \mp 1)^2/4c = \alpha^3$. The different roots of the latter equation, given $c \neq \pm 1$, are mapped by redefining $w \to e^{2\pi i/3}w$. We fix the choice of $\beta$ by requiring $\beta = \alpha = -\mu/3g$ and thus find two $w$-curves compatible with the discriminant (185), they read

$$0 = w^3 + \mu g\Lambda^3 tw - g^3\Lambda^6 t(t \mp 1). \tag{186}$$

These two $w$-curves differ in branch cut structure and computing the monodromies around all combinations of branch points $t = 0, \infty, c, 1/c$ we find

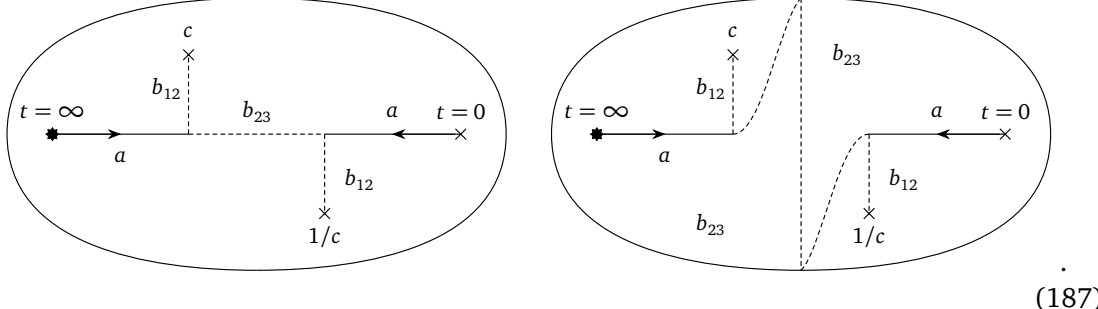

$$\tag{187}$$

where $\gamma = \mp 1$ gives the left, right configuration respectively. We give the monodromy computations in appendix D. Pairing the sheets as $(v_i, w_i)$ we obtain the $\mathcal{N} = 1$ curves.

## C.2 R-symmetry and CSW Duality for $\mathfrak{g} = \mathfrak{su}(4)$

We consider deformations of 4d $\mathcal{N} = 2$ SYM with $\mathfrak{g} = \mathfrak{su}(4)$ by turning on the superpotential (107). For the vacua $(r_1, r_2) = (0,0), (1,1)$ with $(n_1, n_2) = (2,2)$ in which the unbroken gauge symmetry is $\mathfrak{su}(2) \oplus \mathfrak{su}(2) \oplus \mathfrak{u}(1)$, we find the $v$-curves

$$0 = \left(v - \frac{\mu}{2g}\right)^2 \left(v + \frac{\mu}{2g}\right)^2 - \Lambda^4 \left(t + \frac{1}{t} \mp 2\right), \tag{188}$$

matched in monodromy by the $w$-curves

$$0 = (w^2 \mp g^2 \Lambda^4 t)^2. \tag{189}$$

These vacua have a confinement index of $t = 2$ and display interesting properties and degeneracies due to an unbroken R-symmetry acting as $(v, w) \to (-v, w)$ on the $\mathcal{N} = 1$ curve. The vacua $(r_1, r_2) = (1,0), (0,1)$ and those preserving $\mathfrak{su}(3) \oplus \mathfrak{u}(1) \subset \mathfrak{su}(4)$ do not confine and are described below.

We begin by deriving the $v$-curves for all vacua. First we study possible $v$-curves using topological considerations and how these relate via partial Dehn twists. This is essential to apply our proposal (66) as explained in section 5.3. Here redundancies in the description of the curves are given by trivial relabellings of the sheets, rotations in the $v$-plane and different choices of branch cuts. The latter can render curves labelled by distinct $(n_1, n_2)$ and $(r_1, r_2)$ as shown in (133) equivalent. Accounting for these we find in total $1 + 1 + 4 = 6$ curves for a given superpotential. The first two curves are those in (199) while the final $4 = 3 + 1$ curves belong to the non-confining cases.

The SW curve for the unperturbed $\mathcal{N} = 2$ theory takes the form

$$\mathcal{P}(v) = v^4 + u_2 v^2 + u_3 v + u_4 + \Lambda^4 \left(t + \frac{1}{t}\right) = 0. \tag{190}$$

The CB is complex three-dimensional and the SW curve has genus three. Along a codimension two subspace the curve degenerates to genus one and two monopoles become massless. These can be mutually local or not. Prior to this degeneration the $v$-curve has following branch cut structure:

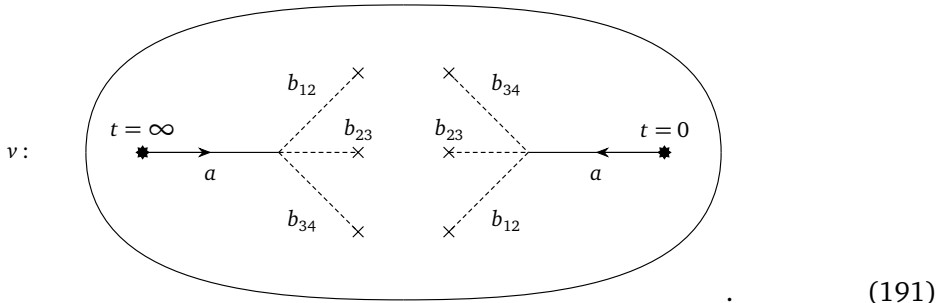

$$\tag{191}$$

We collide two pairs of branch points, related by $t \leftrightarrow 1/t$, at $t = \pm 1$. Different such collisions are related by partial Dehn twists. We begin by discussing the degenerations for which the branch point terminating $b_{23}$ is left unpaired and the pairs of branch points terminating $b_{12}, b_{34}$ are collided at $t = 1$. This yields the degeneration, referred to as the $(0,0)$ $v$-curve:

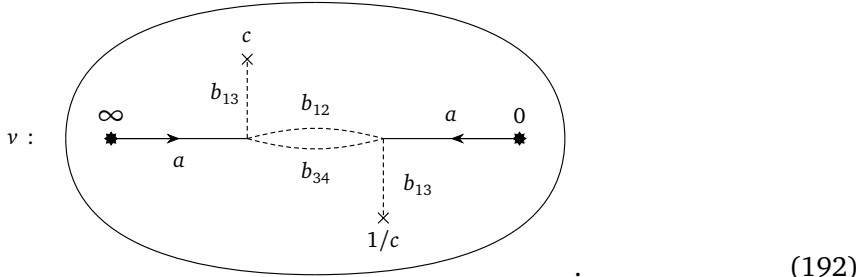

$$\tag{192}$$



Notice that the $b_{23}$ branch point has become a $b_{13}$ branch point due to conjugation by $b_{12}$, which occurs as the $b_{12}$ branch line slides over the $b_{23}$ branch point. The vacua $(1,0),(0,1)$ and $(1,1)$ follow from partial Dehn twists of the $b_{12}, b_{34}$ branch cuts and an overall Dehn twist, respectively. The former two will have branch points collided at both $t = \pm 1$ while the latter has both pairs of branch points collided at $t = -1$. Further we can wrap the branch lines $b_{12}, b_{34}$ around the puncture at $t = 0$ to yield the configuration

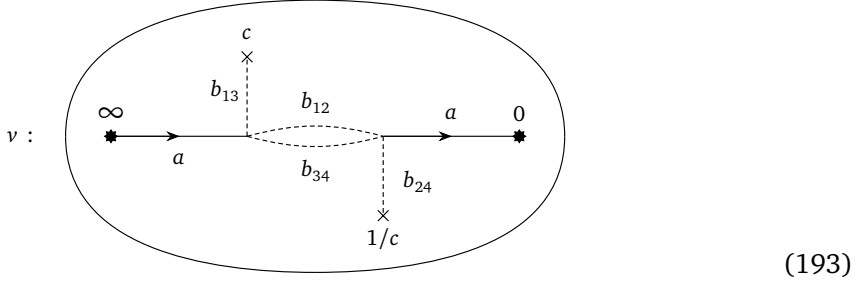

$$(193)$$

to directly connect with (125) and the discussion in section 5.2. Consider the relabelling of sheets with $1 \leftrightarrow 3$ and $2 \leftrightarrow 4$. This leaves the monodromy elements $a, b_{13}, b_{24} \in S_4$ invariant and interchanges $b_{12} \leftrightarrow b_{34}$. This relabelling of sheets maps the branch cut structure of the $(1,0)$ curve into that of the $(0,1)$ curve, they are topologically equivalent. Both the $(0,0)$ and $(1,1)$ curve are left invariant. This involutive relabelling realizes a $\mathbb{Z}_2$ R-symmetry acting on the $\nu$-curve via $\nu \to -\nu$ as we show below. The R-symmetry is spontaneously broken in the $(r_1, r_2) = (1,0),(0,1)$ vacua.

Next consider the degenerations for which $b_{12}$ are not collided. Consider the degenerations for which the remaining branch points are both collided at either $t = 1$ or $t = -1$:

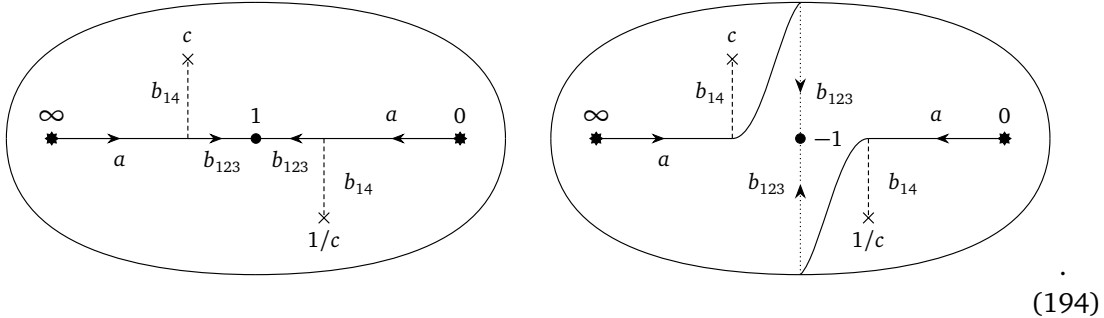

$$(194)$$

Here $b_{123}$ denotes the cyclic permutation $123 \to 231$ and we have marked the point $t = \pm 1$ on the front and back of the Gaiotto curve. These two degenerations are of AD type[19] as collisions give rise to monodromies about $t = \pm 1$. For the second kind of degenerations we collide pairs of branch points both at $t = 1$ and $t = -1$. This gives the $\nu$-curve

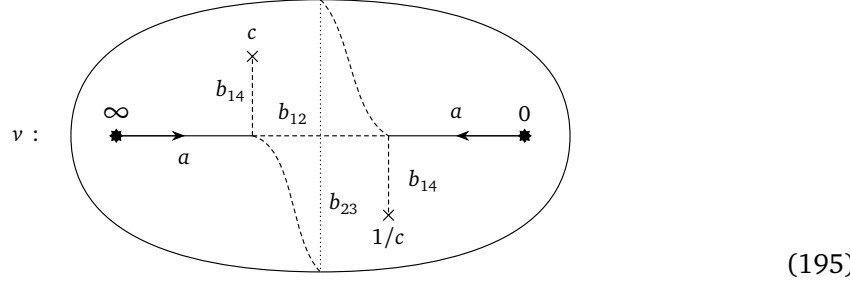

$$(195)$$

---

[19]See appendix B where the rotation results in AD points of the $\mathcal{N} = 2$ theory surviving as $\mathcal{N} = 1$ vacua. There all branch points are collided in a single point $t = \pm 1$ leading to a configuration similar to the one in (194). For such configurations, it is not possible to have multiple configurations related by Dehn twists, though this follows only after studying the equations. Here the discussion of Dehn twists is similarly redundant.

together with two more curves derived from positive overall Dehn twists. We refer to these curves as the $(0),(1),(2)$ vacua respectively. For the $v$-curve (195) we can loop the $b_{14}, b_{23}$ branch lines around the puncture $t = 0$ to give:

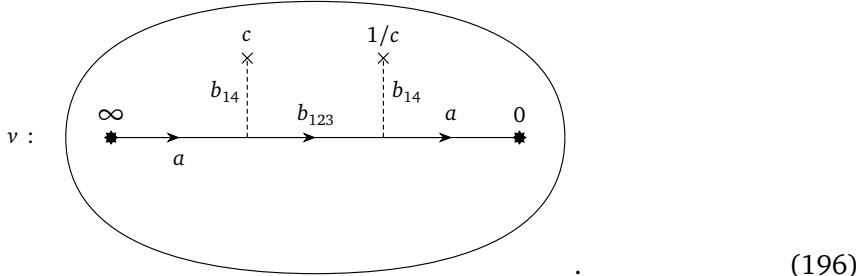

(196)

Finally we wrap the $b_{123}$ branch line around $1/c$ and again make direct contact with (125). The case of not colliding the branch cuts terminating the $b_{34}$ branch cuts is analyzed similarly and we refer to the resulting configurations as $(0)', (1)', (2)'$, respectively.

We discard the curves (194) and find a total of 8 curves for the collision of two branch points with non-commuting monodromy. These are the configurations $(0),(1),(2)$ and $(0)', (1)', (2)'$ and $(1,0),(0,1)$. Of these only 4 have distinct topology of the $v$-curve. The manoeuvres resulting in the curves $(0,1),(0)',(1)',(2)'$ are the same which give the curves $(1,0),(0),(1),(2)$ upon relabelling the sheets $1 \leftrightarrow 3$ and $2 \leftrightarrow 4$, these curves are therefore topologically identical.

Now we derive the equations describing the $v$-curves. The discriminants of the degenerate curves must take one of three forms

$$\Delta(\mathcal{P}, v) = \frac{(t-1)^{2+k}(t+1)^{2-k}}{t^3} p_2(t),$$ (197)

where $k = -2, 0, 2$. Here the polynomial $p_2(t) = -256\Lambda^{12}(t-c)(t-1/c)$ is of degree 2 in $t$ with roots at $t = c, 1/c$. The sublocus of the CB along which the SW curve degenerates to genus one has three irreducible components labelled by $k$. The discriminant (197) takes the form (123).

Let us first consider the cases $k = \pm 2$. The discriminant is reducible and contains a factor of $(t \mp 1)^4$. This degeneration is realized by either two pairs of sheets colliding at distinct points or three sheets colliding in one point above $t = \pm 1$. The latter configurations lead to AD-like singularities and are described by the curves (194), we discard these and focus on the former. The ansatz (197) then determines the $v$-curve

$$0 = (v - \alpha)^2 (v + \alpha)^2 - \Lambda^4 \left( t + \frac{1}{t} \mp 2 \right),$$ (198)

with $\alpha^4 = \Lambda^4 (c \mp 1)^2 / c$. The $-, +$ sign correspond to the configurations previously referred to as the $(0,0),(1,1)$ vacua respectively. In the limit $\Lambda \to 0$ the curve becomes reducible with two double zeros at $\pm\alpha$. The curves (199) therefore describe vacua of a gauge theory with gauge algebra $\mathfrak{su}(2) \oplus \mathfrak{su}(2) \oplus \mathfrak{u}(1)$.

We turn on the superpotential. The critical points $v = 0, -\mu/g$ of the superpotential fix $\alpha = -\mu/2g$ in the center of mass frame. Of the two one-dimensional families (199) only

$$0 = \left( v - \frac{\mu}{2g} \right)^2 \left( v + \frac{\mu}{2g} \right)^2 - \Lambda^4 \left( t + \frac{1}{t} \mp 2 \right),$$ (199)

are not lifted by this perturbation to $\mathcal{N} = 1$.

Now we consider the case $k = 0$. The discriminant ansatz (197) determines the one-parameter family of $v$-curves

$$0 = (v - \alpha)^2(v + \alpha)^2 - \Lambda^4 \left( t + \frac{1}{t} - \frac{5\Lambda^4}{8\alpha^4} + \frac{3\Lambda^{12}}{256\alpha^{12}} + \frac{2v}{\alpha} - \frac{\Lambda^8 v}{8\alpha^9} + \frac{3\Lambda^4 v^2}{8\alpha^6} \right), \tag{200}$$

which can, introducing $\lambda = \Lambda^4/\alpha^3$, also be parametrized as

$$0 = \left( v - \frac{\lambda}{4} \right)^3 \left( v + \frac{3\lambda}{4} \right) - \alpha^2 \left( \alpha\lambda t + \frac{\alpha\lambda}{t} - \alpha^2 - \frac{5\lambda^2}{8} + 2\lambda v + 2v^2 \right). \tag{201}$$

These expressions depend on $v/\alpha$ and $\alpha^4$ only. For a given branch point location $t = c$ there are $16 = 4 \times 4$ solutions $\alpha(c)$, where solutions occur in groups of four grouped by the action $\alpha \to i\alpha$. These rotations can be undone by similar coordinate transformations for $v$, which are a permutation of the sheets near the punctures. This leaves 4 curves up to such redefinitions, which for instance can be taken to have $\mathrm{Re}\,\alpha_i, \mathrm{Im}\,\alpha_i > 0$ where $i = 1, \ldots 4$ labels the curves. These four solutions are continuously connected varying $\alpha \in \mathbb{C}^\times$. The values of $\alpha$ with $c = \pm 1$ give in total 16 branch points in $\mathbb{C}^\times$ around which the branch points $t = c, 1/c$ are permuted. Two paths in $\mathbb{C}^\times$ from $\alpha_i$ to $\alpha_j$ therefore describe distinct topological moves (dragging of the branch point $t = c$) mapping the $i$-th to the $j$-th curve when the difference of paths encloses a branch point.

The four curves described above split as $4 = 3 + 1$. These are most easily distinguished by their monodromy of sheets over the Gaiotto curve. We then find 3 curves with a $\mathbb{Z}_3$ monodromy around paths containing the pair $c, 1/c$ and not enclosing punctures and a single curve with no monodromy. The latter is precisely the curve $(0)$ of (196) derived from topological considerations. The former are the curves $(1,0), (1), (2)$.

There are two limits for these curves. We can take $\Lambda \to 0$ keeping $\alpha$ constant or take $\Lambda, \alpha \to 0$ keeping $\lambda$ constant. In these limits we find $(v - \alpha)^2(v + \alpha)^2$ and $(v - \lambda/4)(v + 3\lambda/4)$ respectively. In the center of mass frame we have $\alpha = -\mu/2g$ and $\lambda = -\mu/g$. In the latter case the relation $\lambda = \Lambda^4/\alpha^3$ fixes $\alpha$. The first limit is associated with the gauge algebra $\mathfrak{su}(2) \oplus \mathfrak{su}(2) \oplus \mathfrak{u}(1)$, in the second limit with $\mathfrak{su}(3) \oplus \mathfrak{u}(1)$. In particular we can not assign a gauge algebra to the $v$-curve (200). This is the statement of the duality derived in [21].

This also follows from following topological consideration. Consider the vacuum $(1)$ resulting from (196) by a positive Dehn twist:

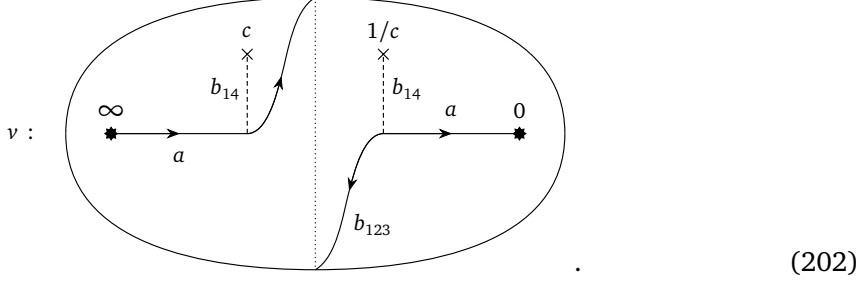

$$\tag{202}$$

We can rearrange branch cuts without changing the topology of the curve. First close the $b_{123}$ branch cut winding about the equator around the puncture $t = 0$, this gives:

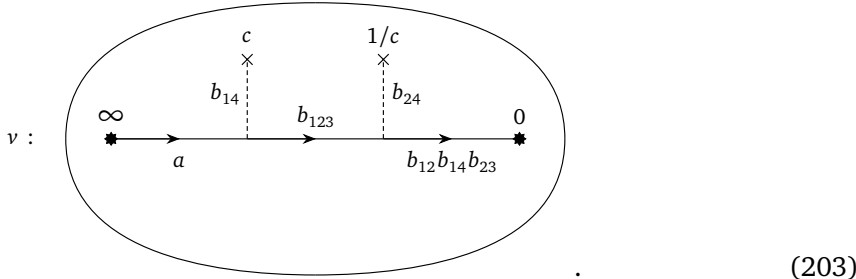

$$\tag{203}$$

Now contract the central branch cut labelled $b_{123}$ to a point creating a four-point junction between two $\mathbb{Z}_4$ and two $\mathbb{Z}_2$ branch lines. Next wrap the $\mathbb{Z}_2$ branch line connecting to $c$ around $1/c$ and resolve the four-point junction. This gives the configuration:

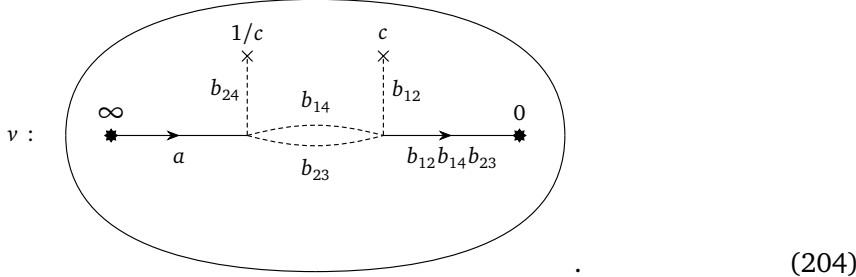

$$\tag{204}$$

Finally, wrap the $b_{12}, b_{14}$ branch line about $t = 0$. This gives a configuration with a partial Dehn twist:

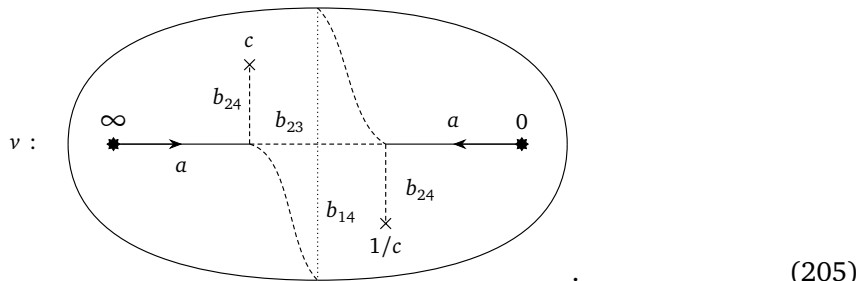

$$\tag{205}$$

This is equivalent to the $(1,0)$ or $(0,1)$ branch cut configuration upon cyclic relabellings, they have identical branch cut structure.

Now we discuss the $w$-curve. We again impose the boundary conditions (113). Diagonalizing these asymptotics we find the maximal growth of $|w| \sim g\Lambda^2 |t|^{1/2}$ when $t \to \infty$. The most general $w$-curve consistent with this bound is

$$0 = w^4 + c_1 w^3 + (c_2 + \delta t)w^2 + (c_3 + \beta t)w + c_4 + \gamma t + \epsilon t^2, \tag{206}$$

where $c_i, \beta, \gamma, \delta, \epsilon$ are complex constants. The $w$-curve is constrained to have $\mathbb{Z}_4$ monodromy at $t = 0, \infty$. The former sets the constants $c_i$ to zero to allow for total ramification while the latter requirement relates the coefficients of the maximals terms with growth $\mathcal{O}(t^2)$ when $t \to \infty$. We find

$$0 = \mathcal{Q}(w) = w^4 - 2g^2\Lambda^4 tw^2 + g^4\Lambda^8 t^2 + g^3\Lambda^6 \beta tw - g^4\Lambda^8 \gamma t, \tag{207}$$

where we have rescaled $\beta, \gamma$ to make them dimensionless. We fix these final two coefficients by requiring that the branch points are located at $c, 1/c$. The discriminant of the candidate curve (207) is

$$\Delta(\mathcal{Q}, w) = -g^{12}\Lambda^{24} t^3 \left(256t^2\beta^2 + (27\beta^4 - 288\beta^2\gamma - 256\gamma^2)t + 256\gamma^3\right). \tag{208}$$

It follows that $\beta^2 = \gamma^3$ and $-(c + 1/c) = (27\beta^4 - 288\beta^2\gamma - 256\gamma^2)/256\beta^2$. This gives

$$c + 1/c = -\frac{27}{256}\gamma^3 + \frac{288}{256}\gamma + \frac{1}{\gamma}. \tag{209}$$

For each value of $\beta$ we have four values for $\gamma$. The discriminant (208) only depends on $\beta^2$. We can redefine $w \leftrightarrow -w$ which maps $\beta \leftrightarrow -\beta$ as seen from (207), i.e. the values $\pm\beta$ are physical equivalent. We therefore find four $w$-curves in total, they are

$$0 = w^4 - 2g^2\Lambda^4 t w^2 + g^4\Lambda^8 t^2 + g^3\Lambda^6\gamma^{3/2} t w - g^4\Lambda^8\gamma t. \tag{210}$$

These four $w$-curves are matched to the $v$-curves by branch cut matching. The roots of (209) have no natural numbering so we resort to pairing curves in the limit of large real $c$. In this limit we have a small real root, one large real root and two complex roots with negative and positive imaginary parts related by conjugation. We compute monodromies following the approach of appendix D. The $w$-curves with these roots inserted match the $(0), (1, 0)$ and the $(1), (2)$ $v$-curve respectively.

The ansatz (206) only yields $w$-curves whose discriminant factors with $(t-1)^2(t+1)^2$. To match the $(0, 0)$ curve (199) we note that its branch cut structure

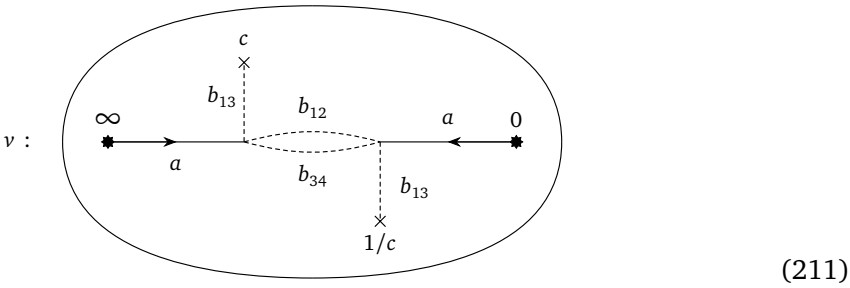

$$\tag{211}$$

is invariant under the R-symmetry $v \to -v$ which interchanges the sheets as $1 \leftrightarrow 3$ and $2 \leftrightarrow 4$. This allows for a $w$-curve which is a two-fold cover instead of a four-fold cover with each of its sheets matched to either the pair of sheets $1, 3$ or $2, 4$ of the $v$-curve. We fold the $v$-curve identifying the sheets related by the R-symmetry, this gives the very simple two-fold cover:

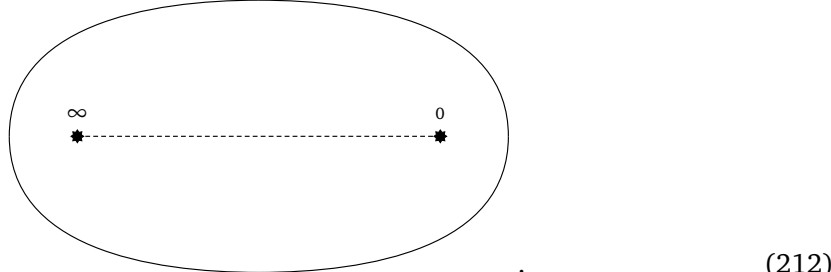

$$. \tag{212}$$

This is nothing but the branch cut configuration of SYM. We find the $w$-curve matching (211)

$$0 = (w^2 - g^2\Lambda^4 t)^2. \tag{213}$$

The $w$-curve matching the $(1, 1)$ vacuum follows from a Dehn twist and reads $0 = (w^2 + g^2\Lambda^4 t)^2$.

# D Monodromy Computations

In section 5 we considered rotations of 4d $\mathcal{N} = 2$ SYM with gauge algebras $\mathfrak{g} = \mathfrak{su}(n)$ by cubic superpotentials preserving $\mathcal{N} = 1$ supersymmetry. From first principles the rotated curve is

argued to be an $n$-fold cover of the Gaiotto curve and swept out by pairs of eigenvalues $v, w$ of the Higgs fields $\phi_\zeta, \varphi$ respectively. Consequently, the branch cut structures of $v, w$ must be consistent and allow for a pairing into $n$ pairs of eigenvalues across the Gaiotto curve. In our approach we determined the possible $v$-curves and $w$-curves independently and then paired them using this topological constraint. In this appendix we discuss the derivation of the relevant branch cut structures.

## D.1  The $v$-curve for CSW with $\mathfrak{g} = \mathfrak{su}(3)$

We discuss the derivation of the branch cuts for the $v$-curves of the low rank example $\mathfrak{g} = \mathfrak{su}(3)$ given in section C.1. The $v$-curves under consideration take the form (183) which we reproduce for convenience here

$$0 = (v - \alpha)^2 (v + 2\alpha) - \Lambda^3(t + 1/t \mp 2). \tag{214}$$

We proceed numerically in Mathematica and set the parameter $\Lambda = 1$. We plot three separate complex planes in the same numerical plane using the code

```
Manipulate[
n = 3; index = Table[i, {i, n}];
Poly = 2 - t[[1]] - 1/(t[[1]] + I t[[2]]) - I t[[2]] + (v + 2 (\[Alpha][[1]] + I \[Alpha][[2]]))(v - \[Alpha][[1]] - I \[Alpha][[2]])^2;
branchpts = {1 - 2 Sqrt[(\[Alpha][[1]] + I \[Alpha][[2]])^3 + (\[Alpha][[1]] + I \[Alpha][[2]])^6 + 2 (\[Alpha][[1]] + I \[Alpha][[2]])^3,
1 + 2 Sqrt[(\[Alpha][[1]] + I \[Alpha][[2]])^3 + (\[Alpha][[1]] + I \[Alpha][[2]])^6 + 2 (\[Alpha][[1]] + I \[Alpha][[2]])^3, 0};
scale = 1;
roots = NSolve[Poly == 0, v];
rts = v /. roots[[index]];
Graphics[{Green, Disk[{0., 0}, .03], Black, Disk[t, .04], Yellow, Disk[\[Alpha], .04], Blue,
Table[Disk[{scale*Re[branchpts[[i]]], scale*Im[branchpts[[i]]]}, .03], {i, 3}], Red,
Table[Disk[{Re[rts[[i]]], Im[rts[[i]]]}, .02], {i, 3}],
Purple, {Disk[{1, 0}, .03], Disk[{-1, 0}, .03], Disk[{0, 1}, .03], Disk[{0, -1}, .03]}},
PlotRange -> 2.5, ImageSize -> 1000],
{{t, {1/2, 0}}, {-3, -3}, {3, 3}, Locator, Appearance -> None}, {{\[Alpha], {-1/2, 0}}, {-3, -3}, {3, 3}, Locator, Appearance -> None},
TrackedSymbols -> True]
```

and

```
Manipulate[
n = 3; index = Table[i, {i, n}];
Poly = -2 - t[[1]] - 1/(t[[1]] + I t[[2]]) - I t[[2]] + (v + 2 (\[Alpha][[1]] + I \[Alpha][[2]])) (v - \[Alpha][[1]] - I \[Alpha][[2]])^2;
branchpts = {-1 -
2 Sqrt[-(\[Alpha][[1]] + I \[Alpha][[2]])^3 + (\[Alpha][[1]] + I \[Alpha][[2]])^6 + 2 (\[Alpha][[1]] + I \[Alpha][[2]])^3,
-1 + 2 Sqrt[-(\[Alpha][[1]] + I \[Alpha][[2]])^3 + (\[Alpha][[1]] + I \[Alpha][[2]])^6 + 2 (\[Alpha][[1]] + I \[Alpha][[2]])^3, 0};
scale = 1;
roots = NSolve[Poly == 0, v];
rts = v /. roots[[index]];
Graphics[{Green, Disk[{0., 0}, .03], Black, Disk[t, .04], Yellow, Disk[\[Alpha], .04], Blue,
Table[Disk[{scale*Re[branchpts[[i]]], scale*Im[branchpts[[i]]]}, .03], {i, 3}], Red,
Table[Disk[{Re[rts[[i]]], Im[rts[[i]]]}, .02], {i, 3}],
Purple, {Disk[{1, 0}, .03], Disk[{-1, 0}, .03], Disk[{0, 1}, .03], Disk[{0, -1}, .03]}},
PlotRange -> 2.5, ImageSize -> 1000],
{{t, {1/2, 0}}, {-3, -3}, {3, 3}, Locator,
Appearance -> None}, {{\[Alpha], {0.9485864075433501, 0}}, {-3, -3}, {3, 3}, Locator, Appearance -> None},
TrackedSymbols -> True]
```

for the $v$-curves with discriminant factors $(t \mp 1)^2$ or equivalently the $v$-curves with two branch points collided at $t = \pm 1$ respectively. This generates as out-put the manipulate environments shown and explained in figure 18. From this we derive the branch cut structures. Consider three loops $\gamma_i$ in the $t$-plane with start and end point at a marked point on the Gaiotto curve (black) and are oriented positively, i.e. counter-clockwise. We choose $\gamma_1$ to enclose the branch point at $t = 1/c$ (blue), $\gamma_2$ to enclose the central branch point (blue) at $t = 0$ and $\gamma_3$ to enclose the branch point at $t = c$ (blue) in figure 18. Along these paths we find, respectively for the curve (214), the two sets of monodromies:

$$
\begin{aligned}
\gamma_1 : & \quad b_{31}, b_{31}, \\
\gamma_2 : & \quad a, a, \\
\gamma_3 : & \quad b_{12}, b_{31}, \\
\gamma_1 + \gamma_2 : & \quad b_{23}, b_{23}, \\
\gamma_2 + \gamma_3 : & \quad b_{23}, b_{12}, \\
\gamma_3 + \gamma_1 : & \quad a, \mathrm{Id}, \\
\gamma_1 + \gamma_2 + \gamma_3 : & \quad a, a^{-1}.
\end{aligned}
\tag{215}
$$

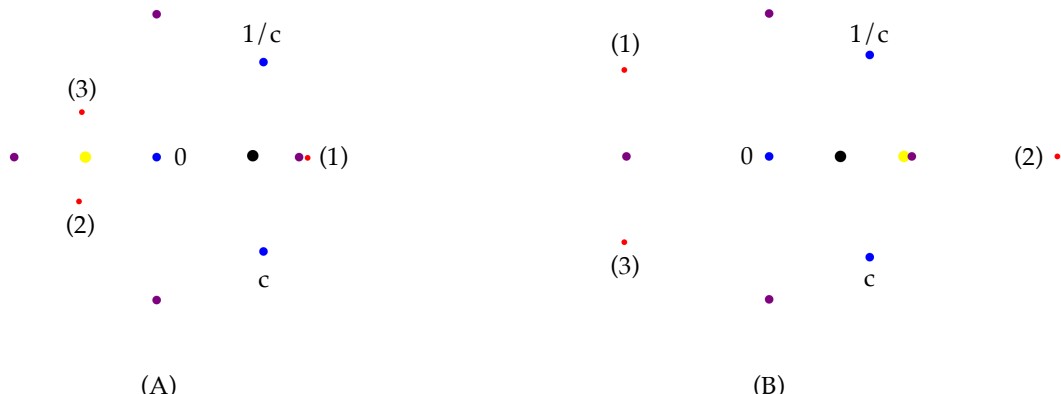

Figure 18: `Manipulate` environment in `Mathematica`. The purple dots are added by hand, fixed and give a frame of reference. The yellow dot sets the value of $\alpha$ and moves in a parameter plane. The blue dots give the location of the branch points in dependence of $\alpha$ and live on the $t$-plane. The black dot determines a value for $t$ in the $t$-plane. The red dots are the three sheets $v(t)$ for given $t$ and move in the cotangent plane of the Gaiotto curve. The yellow and black dot can be dragged using the cursor. The black dot is to be dragged along the paths $\gamma_i$. The subfigure labelled (A) and (B) give the $v$-curve with discriminant factors $(t-1)^2$ and $(t+1)^2$ respectively. The numbers in brackets are attached to the red dots and give a labelling of the three sheets in a neighbourhood of $t = 1/2$. The labels $c, 1/c$ mark the location of the branch points. The singularities are at $t = 0, \infty$ of which the first is labelled.

Here $b_{ij}$ denotes the transposition interchanging the sheets labelled $ij$ and $a$ denotes the cyclic permutation $(123) \to (231)$. These monodromies match those displayed in

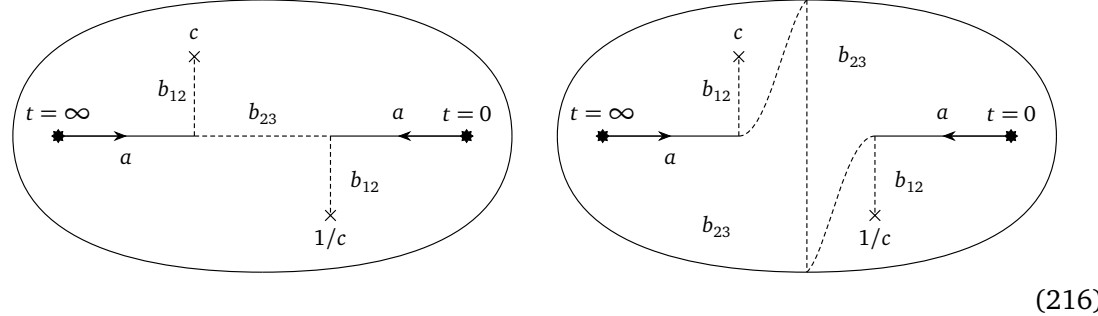

$$(216)$$

respectively. This identification is dependent on the contour $\gamma_3 + \gamma_1$ which in the analysis yielding (215) was taken to be contractible to $t = 1$ without crossing the punctures at $t = 0, \infty$ and therefore runs in (216) as

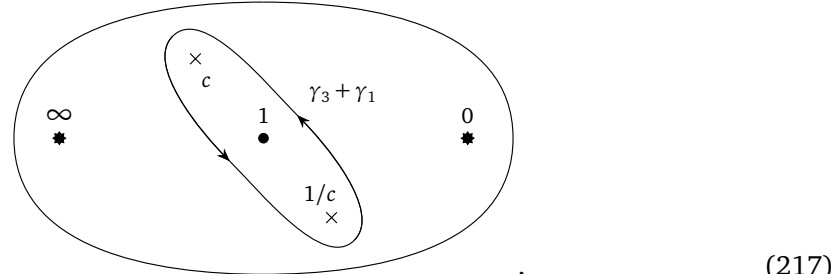

$$(217)$$

Here we add $t = 1$ as a marked point. The results (216) are checked by considering the limits $c \to \pm 1$. We discuss $c \to 1$. In this limit the first curve in (216) simply becomes the

final configuration shown in (173) giving the $v$-curve which can be rotated to the AD theory. Equivalently, the discriminant enhances to include the factor $(t-1)^4$ from colliding two pairs of branch points at $t = 1$ which is the hall mark of the CB points permitting rotation to the AD theory. In the same limit the discriminant of the second curve in turn factors with $(t-1)^2(t+1)^2$ which is the hall mark of the CB points allowing for rotations to SYM. The difference of these limits can be read off from the monodromies about the $\gamma_3 + \gamma_1$ cycle. For example, considering the second configuration of branch cuts in (216) there is no monodromy about $\gamma_3 + \gamma_1$. Therefore the same pair of sheets comes together at $t = c, 1/c$ and moving to $c \to 1$ describes precisely the degeneration discussed in section 4. The occurrence of the SYM and AD points is interchanged when considering the degeneration $c \to -1$.

## D.2 The $w$-curve for CSW with $\mathfrak{g} = \mathfrak{su}(3)$

We discuss the derivation of the branch cuts for the $w$-curves of the low rank example $\mathfrak{g} = \mathfrak{su}(3)$ given in section C.1. The $w$-curves under consideration take the form (186) which we reproduce for convenience here

$$0 = w^3 - 3g^2\Lambda^3\alpha tw - g^3\Lambda^6 t(t \mp 1). \tag{218}$$

We proceed numerically in `Mathematica` and set the parameter $g, \Lambda = 1$. We plot three separate complex planes in the same numerical plane using the code

```
Manipulate[
n = 3; index = Table[i, {i, n}];
Poly = w^3 + (1 - t[[1]] - I t[[2]]) (t[[1]] + I t[[2]]) - 3 w (t[[1]] + I t[[2]]) (\[Alpha][[1]] + I \[Alpha][[2]]);
branchpts = {1 - 2 Sqrt[(\[Alpha][[1]] + I \[Alpha][[2]])^3 + (\[Alpha][[1]] + I \[Alpha][[2]])^6 + 2 (\[Alpha][[1]] + I \[Alpha][[2]])^3,
1 + 2 Sqrt[(\[Alpha][[1]] + I \[Alpha][[2]])^3 + (\[Alpha][[1]] + I \[Alpha][[2]])^6 + 2 (\[Alpha][[1]] + I \[Alpha][[2]])^3, 0};
scale = 1;
roots = NSolve[Poly == 0, w];
rts = w /. roots[[index]];
Graphics[{Green, Disk[{0., 0}, .03], Black, Disk[t, .04], Yellow, Disk[\[Alpha], .04], Blue,
Table[Disk[{scale*Re[branchpts[[i]]], scale*Im[branchpts[[i]]]}, .03], {i, 3}], Red, Table[Disk[{Re[rts[[i]]], Im[rts[[i]]]}, .02], {i, 3}],
Purple, {Disk[{1, 0}, .03], Disk[{-1, 0}, .03], Disk[{0, 1}, .03], Disk[{0, -1}, .03]}}, PlotRange -> 2.5, ImageSize -> 1000],
{{t, {1/2, 0}}, {-3, -3}, {3, 3}, Locator, Appearance -> None},
{{\[Alpha], {-1/2, 0}}, {-3, -3}, {3, 3}, Locator, Appearance -> None}, TrackedSymbols -> True]
```

and

```
Manipulate[
n = 3; index = Table[i, {i, n}];
Poly = w^3 - (t[[1]] + I t[[2]]) (1 + t[[1]] + I t[[2]]) - 3 w (t[[1]] + I t[[2]]) (\[Alpha][[1]] + I \[Alpha][[2]]);
branchpts = {-1 - 2 Sqrt[-(\[Alpha][[1]] + I \[Alpha][[2]])^3 + (\[Alpha][[1]] + I \[Alpha][[2]])^6 + 2 (\[Alpha][[1]] + I \[Alpha][[2]])^3,
-1 + 2 Sqrt[-(\[Alpha][[1]] + I \[Alpha][[2]])^3 + (\[Alpha][[1]] + I \[Alpha][[2]])^6 + 2 (\[Alpha][[1]] + I \[Alpha][[2]])^3, 0};
scale = 1;
roots = NSolve[Poly == 0, w];
rts = w /. roots[[index]];
Graphics[{Green, Disk[{0., 0}, .03], Black, Disk[t, .04], Yellow, Disk[\[Alpha], .04], Blue,
Table[Disk[{scale*Re[branchpts[[i]]], scale*Im[branchpts[[i]]]}, .03], {i, 3}], Red, Table[Disk[{Re[rts[[i]]], Im[rts[[i]]]}, .02], {i, 3}],
Purple, {Disk[{1, 0}, .03], Disk[{-1, 0}, .03], Disk[{0, 1}, .03], Disk[{0, -1}, .03]}},
PlotRange -> 2.5, ImageSize -> 1000],
{{t, {1/2, 0}}, {-3, -3}, {3, 3}, Locator, Appearance -> None},
{{\[Alpha], {0.9485864075433501, 0}}, {-3, -3}, {3, 3}, Locator, Appearance -> None}, TrackedSymbols -> True]
```

for the $w$-curves with ($\mp$) in (218). This generates as out-put the `Manipulate` environments shown and explained in figure 19. From this we derive the branch cut structures by considering the monodromy about the same loops $\gamma_i$ as in appendix D.1. We find exactly the monodromies (215) and conclude that the branch cut structures of the $w$-curves (218) are respectively:

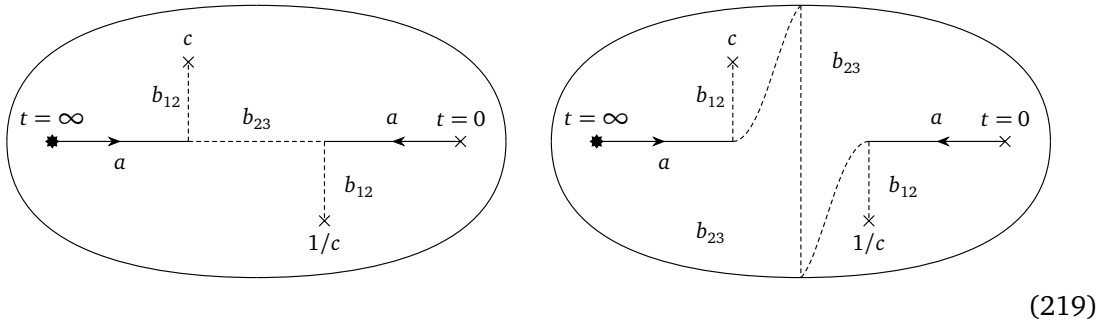

$$\tag{219}$$

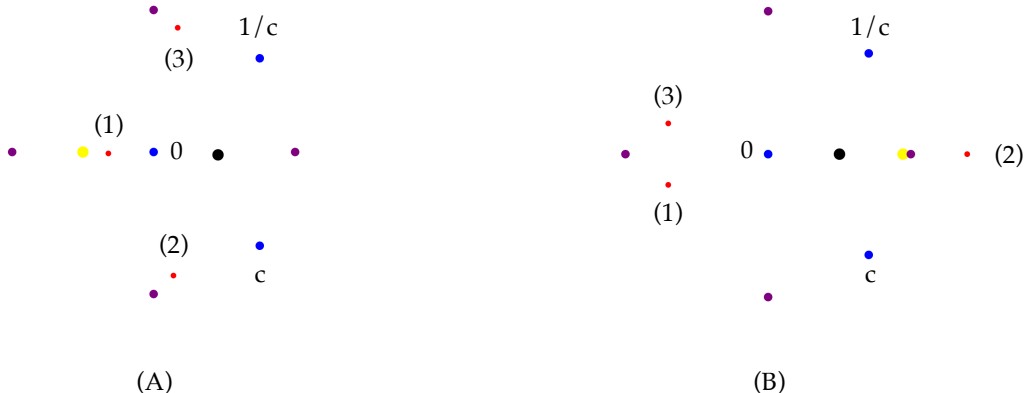

Figure 19: `Manipulate` environment in `Mathematica`. The purple dots are added by hand, fixed and give a frame of reference. The yellow dot sets the value of $\alpha$ and moves in a parameter plane. The blue dots give the location of the branch points in dependence of $\alpha$ and live on the $t$-plane. The black dot determines a value for $t$ in the $t$-plane. The red dots are the three sheets $w(t)$ for given $t$ and move in the cotangent plane of the Gaiotto curve. The yellow and black dot can be dragged using the cursor. The black dot is to be dragged along the paths $\gamma_i$. The subfigure labelled (A) and (B) give the $w$-curve with ($\mp$) in (218) respectively. The numbers in brackets are attached to the red dots and give a labelling of the three sheets in a neighbourhood of $t = 1/2$. The labels $c, 1/c$ mark the location of the branch points. The singularities are at $t = 0, \infty$ of which the first is labelled.

# E    Rotations via the Dijkgraaf-Vafa Curve

Here we present the derivation of some $\mathcal{N} = 1$ curves using the Dijkgraaf-Vafa (DV) curve [15, 21]. For the SYM curves these are known results [8, 10, 11]. The aim of this section is to show that both this approach and our approach agree on non-trivial examples, we discuss the CSW set-up for $\mathfrak{g} = \mathfrak{su}(n)$ with $n = 3$ explicitly. Further, at higher rank the spectral curves of the Higgs field $\varphi$ is increasingly difficult to derive from the DV curve and our approach offers a computational alternative. More importantly the DV curve approach lends itself less straightforwardly to rotations of more general class S theories, such as those discussed in section 6, and is unsuitable to study the topology of the $\mathcal{N} = 1$ curve.

We begin by reviewing aspects of the Dijkgraaf-Vafa curve. The DV curve for 4d $\mathcal{N} = 2$ SYM with gauge algebra $\mathfrak{su}(n)$, deformed to $\mathcal{N} = 1$ by the tree level superpotential $W$ of degree $k + 1$, reads

$$w^2 - W'(v)w - f_{k-1}(v) = 0. \tag{220}$$

Here $f_{k-1}$ is polynomial of degree $k - 1$. The DV curve determines the eigenvalues of the Higgs field $\varphi$. The coordinate $v$ is a solution to the SW curve $P_n(v) - \Lambda^n(t + 1/t) = 0$. These two curves are expected to be equivalent to the $v$- and $w$-curve in our previous discussion.

The SW curve is necessarily tuned to a $(k - 1)$-dimensional subloci of the CB at which $n - k + 1$ mutually local monopoles become massless, otherwise a superpotential can not be turned on. Turning on a specific superpotential this locus is lifted to points, which are determined by the coefficients of $W(v)$. These $(k - 1)$-dimensional loci are most conveniently described using the coordinate $z = 2t - P_n(v)$ with respect to which the SW curve becomes

$$z^2 = (P_n(v) + 2\Lambda^n)(P_n(v) - 2\Lambda^n). \tag{221}$$

The presence of the massless monopoles is then equivalent to the factorization

$$P_n(v) + 2\Lambda^n = H_{s_+}^+(v)^2 R_{n-2s_+}^+(v),$$
$$P_n(v) - 2\Lambda^n = H_{s_-}^-(v)^2 R_{n-2s_-}^-(v),$$
(222)

where $n - k = s_+ + s_-$ and $H_{s_\pm}^\pm, R_{n-2s_\pm}^\pm$ are polynomials of degree $s_\pm, n-2s_\pm$ respectively. This factorization condition cuts out the $(k-1)$-dimensinoal sublocus on the CB. Of this sublocus the values extremizing the superpotential are not lifted, see section 5. Given a vacuum characterized by the polynomials $R_{n-2s_\pm}^\pm$ and $W'$ the polynomial $f_{k-1}$ is computed via the matrix model curve

$$y^2 = W'(x)^2 + 4f_{k-1}(x) = g_k^2 R_{n-2s_+}^+(x) R_{n-2s_-}^-(x),$$
(223)

where $g_k$ is the leading order coefficient in $W'(v)$.

## E.1 Rotation of SYM

The case of $\mathcal{N} = 1$ SYM with a massive adjoint chiral follows from rotations with $W'(v) = \mu v$. The SW curve is tuned to CB points at which $n-1$ mutually local monopoles become massless, see section 4. At these points the polynomial $P_n$ takes the form $P_n^{(r)}(v) = 2\Lambda^n T_n^{(r)}(v/2\Lambda)$ with $T_n^{(r)}(x) = T_n(e^{2\pi i r/2n} x)$ where $T_n$ is the $n$-th Chebyshev polynomial of the first kind and $r = 0, \ldots, n-1$. From (222) and (223) the constant polynomial $f_0$ is determined to $f_0 = -\mu^2 \Lambda^2 \omega^r$ with $\omega = \exp(2\pi i/n)$. The DV curve therefore takes the form

$$w^2 - \mu w v + \mu^2 \Lambda^2 \omega^r = 0.$$
(224)

We solve for $v$ and substitute the expression into the SW curve to find

$$\frac{\Lambda^{2n}\mu^{2n}}{w^n} + w^n - \frac{\Lambda^n \mu^n (t^2 + 1)}{t} = 0,$$
(225)

which we solve for $w^n$ for two solutions

$$w^n = \frac{\mu^n \Lambda^n}{t}, \qquad w^n = \mu^n \Lambda^n t,$$
(226)

of which the second matches our previous result in (87). Interestingly the DV curve gives expression (226) for rotations about either of the punctures at $t = 0, \infty$.

## E.2 Rotation of CSW with $\mathfrak{g} = \mathfrak{su}(3)$

Here we consider rotations to the CSW set-up for gauge algebra $\mathfrak{g} = \mathfrak{su}(3)$ by the superpotential $W'(v) = gv^2 + \mu v$. This rotation is possible along a one-dimensional sublocus of the CB at which a single monopole becomes massless. For a choice of constants $g, \mu$ we find the factorization condition (222) to be satisfied by two choices for $P_3(v)$, they are

$$P_3^{(1)}(v) = (v + \mu/g)^2 v + 2\Lambda^3,$$
$$P_3^{(2)}(v) = (v + \mu/g)^2 v - 2\Lambda^3.$$
(227)

From (222) and (223) the linear polynomial $f_1$ is computed respectively to

$$f_1^{(1)}(v) = g^2 \Lambda^3 v,$$
$$f_1^{(2)}(v) = -g^2 \Lambda^3 v.$$
(228)

We consider the case labelled (1). The DV curve takes the form

$$w^2 - (gv^2 + \mu v)w - g^2\Lambda^3 v = 0. \tag{229}$$

We solve for $v$ and substitute the result in the SW curve, square the relation to remove square roots, clear denominators and find the expression to factorize

$$\left(g\Lambda^3 tw\mu + g^3\Lambda^6(t-1) + t^2 w^3\right)\left(-g\Lambda^3 tw\mu + g^3\Lambda^6(t-1)t - w^3\right) = 0. \tag{230}$$

This gives the two $w$-curves

$$\begin{aligned}
0 &= t^2 w^3 + g\Lambda^3 tw\mu + g^3\Lambda^6(t-1), \\
0 &= w^3 + g\Lambda^3 tw\mu - g^3\Lambda^6(t-1)t.
\end{aligned} \tag{231}$$

They correspond to rotations about $t = 0, \infty$ respectively. We shift the $v$-curve eliminating the $v^2$ factor and pair it with the second curve in (231). This gives the $v$- and $w$-curve

$$\begin{aligned}
0 &= (v + \mu/3g)^2(v - 2\mu/3g) - \Lambda^3(t + 1/t - 2), \\
0 &= w^3 + \mu g\Lambda^3 tw - g^3\Lambda^6(t-1)t,
\end{aligned} \tag{232}$$

matching one set of the curves given in (177). The second curve follows from an identical analysis starting from the case labelled (2) above. In total we find two $\mathcal{N} = 1$ curves from rotation about each puncture.

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
