# Peer review of "Liberating Confinement from Lagrangians: 1-form Symmetries and Lines in 4d N=1 from 6d N=(2,0)"

_SciPost Physics, doi:SciPost Phys. 12, 040 (2022)_

## Round 2 · Referee Report · Anonymous (Referee 1) · 2021-10-3

Strengths

The paper introduces a method to detect confinement for some 4d N=1 theories obtained from class S theories. Since the method is based on the study of the geometry of the compactification, it could be applied also to non-Lagrangian theories.

Weaknesses

Very technical presentation throughout the paper.

Report

The paper addresses the problem of detecting confinement using a geometric method. More specifically, the authors consider four-dimensional N=1 deformations of N=2 class S theories, and argue that the behavior of vevs of the line operators of the resulting theory is encoded in the 1-cycles of the N=1 curve associated to the chosen vacuum. Most of the paper is devoted to checking that the geometric analysis leads to the same results as the field theory one. Following the formal definition of confinement for a gauge theory, they then conclude that this provides a method for detecting the "confining phase" of non-Lagrangian theories.

After minor revisions, I would recommend this paper for publication.

Requested changes

1) In Section 1, the authors state that their "considerations apply to general class S setup". Is it obvious that a more general setup would have the same behavior?

2) The authors propose a method for detecting confinement, and check in detail that it reproduces the results, known from a field-theoretical analysis, for the cases of $\mathfrak{su}(n)$ N=1 SYM and the theory studied by Cachazo-Seiberg-Witten. Both these theories have confining vacua. Would it be a good proof of concept for their method to consider a case where it is known from field theory that the N=1 deformation does not admit a confining vacuum?

3) Some minor additional points: - There are a few mathematical notations/quantities that are used but then defined only later on. For instance, $\Lambda_{\mathcal{N}=2}$ is first used in p. 8, and then defined at p. 36. Similarly, the Pontryagin dual is already used at p. 15, but only defined at p. 18. - The contents of footnote 3 p. 8 are not common knowledge for a non-expert reader. Could the authors provide a reference? - At p. 17: "we will only study theories for which $\mathcal{L}$ is an abelian group under OPE of line operators". Is there a case in which it is not? - The text around (3.5) p. 19 contains repetitions and colloquialisms: "is modded out by modding out... that are modded out..." - Minor typos (understandable in a 83-page paper): p. 9 "led", p. 18 "where $\hat{\Lambda}$ is the Pontryagin dual of $\Lambda$", p. 18 footnote 9 "The Pontryagin dual group..."

  • validity: high
  • significance: high
  • originality: high
  • clarity: high
  • formatting: perfect
  • grammar: excellent

Author:  Max Hubner  on 2021-11-02  [id 1904]

(in reply to Report 1 on 2021-10-03)
Category:
remark

We thank the referee for their valuable feedback and suggestions.

Regarding the first point raised in the report, we just wanted to highlight that one can also include regular punctures and our considerations would apply in the same fashion as for irregular punctures. We have modified the sentence to make it more clear.

The second remark suggests adding non-confining examples. The current version of the paper deals in multiple instances (eg. SO(3) SYM) with theories that have both confining and non-confining vacua. As for theories with exclusively non-confining vacua, a few examples like SU(N) with fundamental chirals come into mind, but such theories have a trivial 1-form symmetry to start with, and hence there is no scope for confinement. As far as theories having non-trivial 1-form symmetry and exclusively non-confining vacua are concerned, we are afraid we do not know any Lagrangian examples that 1. lie in the class of theories we consider, and 2. do not already arise by choosing a polarization in su(n) SYM.

Regarding footnote 3 p. 8, we added a reference. Regarding the sentence on p. 17, we will added a footnote 8. Finally, the referee kindly pointed out many typos that have been corrected.

We hope that the revisions will make the paper suitable for publication.

---

## Editorial Decision

published